# OGBench:
# Benchmarking Offline Goal-Conditioned RL

**Seohong Park**[1]    **Kevin Frans**[1]    **Benjamin Eysenbach**[2]    **Sergey Levine**[1]
[1]University of California, Berkeley    [2]Princeton University
seohong@berkeley.edu

## Abstract

Offline goal-conditioned reinforcement learning (GCRL) is a major problem in reinforcement learning (RL) because it provides a simple, unsupervised, and domain-agnostic way to acquire diverse behaviors and representations from unlabeled data without rewards. Despite the importance of this setting, we lack a standard benchmark that can systematically evaluate the capabilities of offline GCRL algorithms. In this work, we propose OGBench, a new, high-quality benchmark for algorithms research in offline goal-conditioned RL. OGBench consists of 8 types of environments, 85 datasets, and reference implementations of 6 representative offline GCRL algorithms. We have designed these challenging and realistic environments and datasets to directly probe different capabilities of algorithms, such as stitching, long-horizon reasoning, and the ability to handle high-dimensional inputs and stochasticity. While representative algorithms may rank similarly on prior benchmarks, our experiments reveal stark strengths and weaknesses in these different capabilities, providing a strong foundation for building new algorithms.

Project page: https://seohong.me/projects/ogbench

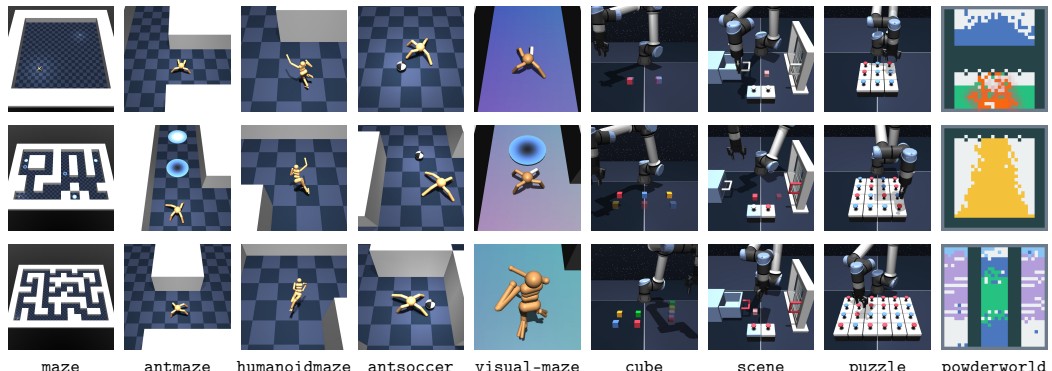

|       |         |            |          |             |      |       |        |            |
|-------|---------|------------|----------|-------------|------|-------|--------|------------|
| maze  | antmaze | humanoidmaze | antsoccer | visual-maze | cube | scene | puzzle | powderworld |

Figure 1: **OGBench Overview.** OGBench provides a variety of state- and pixel-based locomotion, manipulation, and drawing tasks that are designed to exercise diverse challenges in offline goal-conditioned RL, such as stitching, long-horizon reasoning, and stochastic control.

## 1 Motivation

**Why offline goal-conditioned reinforcement learning (RL)?** The enduring trend in modern machine learning is to simplify domain-specific assumptions and scale up the data. In computer vision and natural language processing, the strongest general-purpose models are trained via simple *unsupervised* objectives on raw, unlabeled data, such as next-token prediction, contrastive learning, and masked auto-encoding. What analogous paradigm could enable data-driven unsupervised learning for reinforcement learning? Ideally, such a framework should be able to produce from data a generalist policy that can be directly queried or adapted to solve a variety of downstream tasks, much like how generative language models trained via next-token prediction can be easily adapted to everyday tasks.

We posit that a natural analogy to data-driven unsupervised learning in RL is **offline goal-conditioned RL (GCRL)**. The objective of offline goal-conditioned RL is fully unsupervised, remarkably simple, and requires no domain knowledge: it merely aims to learn to reach any state from any other state in the dataset in the fewest number of steps. However, mastering this simple objective is exceptionally difficult: the agent needs to not only acquire diverse skills to efficiently navigate the state space, but

also have a deep, complete understanding of the underlying world and dataset. As a result, offline goal-conditioned RL yields a highly capable general-purpose multi-task policy as well as rich, useful representations that can be adapted to solve a variety of downstream tasks (Ghosh et al., 2023; Kim et al., 2024). Indeed, for such simplicity and generality, interest in (offline) goal-conditioned RL has recently surged to the extent that even a standalone workshop on goal-conditioned RL was held at a machine learning conference.[1]

**Why a new benchmark?** Despite the importance of and increasing interest in offline goal-conditioned RL, we currently lack a standard benchmark that can systematically assess the capabilities of offline GCRL algorithms, such as the ability to stitch, perform long-horizon reasoning, and handle stochasticity. Prior works in offline goal-conditioned RL (Eysenbach et al., 2022; Ma et al., 2022; Park et al., 2023; Myers et al., 2024) have mainly used either existing datasets for standard offline RL tasks without modification (*e.g.*, D4RL (Fu et al., 2020)), relatively simple goal-conditioned tasks (*e.g.*, Fetch (Plappert et al., 2018)), or tasks tailored to demonstrate the individual abilities of the proposed methods. This often results in limited evaluation. Prior works often evaluate their multi-task policies only on a single task (when using datasets not originally designed for offline GCRL), or learn relatively simple behaviors. While there exist some prior tasks tailored to evaluate *individual* properties of offline GCRL such as stitching or generalization (Yang et al., 2023; Ghugare et al., 2024), we lack a comprehensive, standardized benchmark that exhaustively assesses various properties of offline GCRL algorithms with diverse, challenging tasks.

Therefore, we introduce a benchmark named **Offline Goal-Conditioned RL Benchmark (OGBench)** in this work (Figure 1). The primary goals of this benchmark are to facilitate *algorithms research* in offline goal-conditioned RL and to provide a set of complex tasks that can unlock the potential of offline GCRL. Our benchmark introduces 8 types of environments and 85 datasets across robotic locomotion, robotic manipulation, and drawing, and provides well-tuned reference implementations of 6 representative offline GCRL methods. These datasets, tasks, and implementations are carefully designed. Tasks are designed in a way that complex behaviors can naturally emerge when successfully solved, and that they pose diverse algorithmic challenges in offline GCRL, such as goal stitching, stochastic control, long-horizon reasoning, and more. Dataset and task difficulties are carefully adjusted to highlight stark contrasts between algorithms across multiple criteria. The entire benchmark is designed to minimize unnecessary computational overhead and maximize usability such that any researcher can easily iterate and evaluate new ideas. We believe OGBench serves as a solid foundation for developing ideal algorithms for goal-conditioned and unsupervised RL from data.

## 2 PROBLEM SETTING

The offline goal-conditioned RL problem is defined by a controlled Markov process $\mathcal{M} = (\mathcal{S}, \mathcal{A}, \mu, p)$ (*i.e.*, a Markov decision process (MDP) without rewards) and an unlabeled dataset $\mathcal{D}$, where $\mathcal{S}$ denotes the state space, $\mathcal{A}$ denotes the action space, $\mu(s) \in \Delta(\mathcal{S})$[2] denotes the initial state distribution, and $p(s' \mid s, a) \colon \mathcal{S} \times \mathcal{A} \to \Delta(\mathcal{S})$ denotes the transition dynamics function. $\Delta(\mathcal{X})$ denotes the set of probability distributions defined on a set $\mathcal{X}$. The dataset $\mathcal{D} = \{\tau^{(n)}\}_{n \in \{1, 2, \ldots, N\}}$ consists of unlabeled trajectories $\tau^{(n)} = (s_0^{(n)}, a_0^{(n)}, s_1^{(n)}, \ldots, s_{T_n}^{(n)})$.

The objective of offline goal-conditioned RL is to learn to reach any state from any other state in the minimum number of time steps. Formally, offline GCRL aims to learn a goal-conditioned policy $\pi(a \mid s, g) \colon \mathcal{S} \times \mathcal{S} \to \Delta(\mathcal{A})$ that maximizes the objective $\mathbb{E}_{\tau \sim p(\tau \mid g)}[\sum_{t=0}^{T} \gamma^t \delta_g(s_t)]$ for all $g \in \mathcal{S}$, where $T \in \mathbb{N}$ denotes the episode horizon, $\gamma \in (0, 1)$ denotes the discount factor, $p(\tau \mid g)$ denotes the distribution given by $p(\tau \mid g) = \mu(s_0)\pi(a_0 \mid s_0, g)p(s_1 \mid s_0, a_0) \cdots p(s_T \mid s_{T-1}, a_{T-1})$, and $\delta_g(\cdot)$ denotes the Dirac delta "function"[3] at $g$. Note that we use the entire state space as the goal space (*i.e.*, a goal is simply a full state, not part of the state like only the $x$-$y$ position of the agent). This choice makes the objective fully unsupervised, making it suitable for domain-agnostic training from unlabeled data. Our goal in this paper is to propose a new benchmark in offline GCRL. That is, formally speaking, to specify the dynamics and dataset for each task we introduce.

---

[1] https://goal-conditioned-rl.github.io/2023/

[2] We denote placeholder variables in gray throughout the paper.

[3] In a discrete MDP, $\delta_g(s)$ is equal to an indicator function $\mathbf{1}_{\{g\}}(s)$. In a continuous MDP, it is technically not well-defined in its current form. It can be made precise using measure-theoretic notation or the distribution theory, but we choose to avoid them for simplicity.

Table 1: **Properties of benchmark tasks.** We summarize the properties of benchmark tasks commonly used in prior works (above the line) and our OGBench tasks (below the line).

| Benchmark Task | Type[1] | Longest Task[2] | # Subtasks[3] | Test Stitching?[4] | Have Stoch. Tasks?[5] | Support Pixels?[6] | Multi-Goal?[7] | Dependency[8] |
|---|---|---|---|---|---|---|---|---|
| D4RL AntMaze | Loco. | $\approx 400$ | - | ✗ | ✗ | ✗ | ✗ | MuJoCo |
| D4RL Kitchen | Manip. | $\approx 250$ | 4 | ○ | ✗ | ✗ | ✗ | MuJoCo |
| Roboverse | Manip. | $\approx 100$ | 1-2 | ○ | ✗ | ○ | ○ | PyBullet |
| Fetch | Manip. | $\approx 50$ | 1 | ✗ | ✗ | ✗ | ○ | MuJoCo |
| PointMaze (ours) | Loco. | $\approx 600$ | - | ○ | ○ | ✗ | ○ | MuJoCo |
| AntMaze (ours) | Loco. | $\approx 1000$ | - | ○ | ○ | ○ | ○ | MuJoCo |
| HumanoidMaze (ours) | Loco. | $\approx 3000$ | - | ○ | ✗ | ○ | ○ | MuJoCo |
| AntSoccer (ours) | Loco. | $\approx 1000$ | - | ○ | ✗ | ✗ | ○ | MuJoCo |
| Cube (ours) | Manip. | $\approx 400$ | 1-4 | ○ | ✗ | ○ | ○ | MuJoCo |
| Scene (ours) | Manip. | $\approx 400$ | 2-8 | ○ | ✗ | ○ | ○ | MuJoCo |
| Puzzle (ours) | Manip. | $\approx 800$ | 2-24 | ○ | ✗ | ○ | ○ | MuJoCo |
| Powderworld (ours) | Draw. | $\approx 100$ | - | ○ | ○ | ○ | ○ | NumPy |

[1] Environment type (locomotion, manipulation, or drawing).   [2] The (approximate) minimum number of environment steps to solve the longest task.
[3] The number of atomic behaviors (*e.g.*, pick-and-place) in each manipulation task.   [4] Does it contain tasks that require goal stitching?
[5] Does it contain tasks with stochastic dynamics?   [6] Does it support pixel-based observations?   [7] Does it use multiple goals for evaluation?
[8] The main dependency of the benchmark.

## 3   How Have Prior Works Benchmarked Offline GCRL?

While many excellent offline GCRL algorithms have been proposed so far, the community currently lacks a standardized way to evaluate their performance, unlike other fields in RL (Brockman et al., 2016; Tassa et al., 2018; Fu et al., 2020; Terry et al., 2021). Tasks used by prior works in offline GCRL are often limited for providing a proper evaluation for various reasons. For example, many works directly use tasks in existing offline RL benchmarks (not necessarily designed for offline *goal-conditioned* RL), such as D4RL AntMaze and Kitchen (Fu et al., 2020). However, since the tasks are designed for single-task offline RL, they often evaluate their multi-task policies only on the single, original goal when using these tasks (Eysenbach et al., 2022; Park et al., 2023; Zheng et al., 2024b; Myers et al., 2024), which results in limited evaluation. Some employ online GCRL tasks provided by Plappert et al. (2018) (*e.g.*, Fetch tasks) with policy-collected datasets (Ma et al., 2022; Yang et al., 2023), but these tasks are mostly atomic (*e.g.*, single pick-and-place) and do not sufficiently address various challenges in offline GCRL, such as long-horizon reasoning and goal stitching. While Roboverse (Fang et al., 2022; Zheng et al., 2024a) provides pixel-based manipulation tasks, the tasks are still relatively atomic and it requires installing multiple, fragmented dependencies, making it less approachable for researchers. Some prior works construct bespoke tasks and datasets to individually study the important and specific features of their algorithms (*e.g.*, stitching (Ma et al., 2022; Ghugare et al., 2024; Wang et al., 2024) and stochasticity (Myers et al., 2024)), but it is often not entirely clear how algorithms compare to one another or if the new capabilities of new methods (*e.g.*, stitching) come at the loss of other capabilities. These limitations of previous evaluation tasks have motivated us to create a new benchmark. In this work, we introduce a set of diverse tasks that cover various challenges in offline GCRL, enabling a much more thorough and multi-faceted evaluation than previous tasks. We summarize the properties of the previous tasks and our new tasks in Table 1 and refer to Appendix E for further discussion on related work.

## 4   Overview of OGBench

We now introduce our benchmark, **Offline Goal-Conditioned RL Benchmark (OGBench)**. The primary goal of this benchmark is to provide tasks and datasets to unlock the full potential of offline goal-conditioned RL. To this end, we pose diverse challenges in offline GCRL throughout the benchmark in such a way that researchers can easily test and iterate on algorithmic ideas, and that complex, intriguing behaviors can naturally emerge when successful. OGBench consists of 8 types of environments, 85 datasets, and reference implementations of 6 representative offline GCRL algorithms. In the following sections, we first describe the challenges in offline GCRL (Section 5) and then outline our core design philosophies (Section 6). We next introduce the tasks and datasets (Section 7) and present the benchmarking results of the current algorithms (Section 8).

## 5   Challenges in Offline Goal-Conditioned RL

Offline goal-conditioned RL, despite its simplicity, is a challenging problem. Here, we discuss the major challenges in offline GCRL, which will motivate the design choices in our benchmark tasks.

**(1) Learning from suboptimal, unstructured data:** An ideal offline GCRL algorithm should be able to learn an effective multi-task policy from diverse and suboptimal data. This is especially important, considering that suboptimal (yet diverse) data is much cheaper to collect than curated,

expert datasets (Lynch et al., 2019), and that the very use of large, diverse, unstructured data is one of the foundations for the success of modern machine learning. Reflecting this challenge, we provide datasets with high diversity and varying suboptimality in this benchmark to challenge the capabilities of offline GCRL algorithms.

**(2) Goal stitching:** Another important challenge is to stitch the initial and final states of different trajectories to learn more diverse behaviors. We call this "goal stitching." Goal stitching is different from "regular" stitching in offline RL, which applies only when the dataset is suboptimal. Unlike regular stitching, goal stitching applies *even when the dataset only consists of optimal, expert trajectories*, because we can often acquire more diverse goal-reaching behaviors by stitching multiple trajectories together, regardless of their optimality. For instance, an agent can stitch two atomic pick-and-place behaviors to sequentially move two objects in a single episode, even when the dataset does not contain any double pick-and-place behaviors. Goal stitching is crucial to learning diverse behaviors in many real-world applications with high behavioral diversity and large state spaces. In our benchmark, we introduce many tasks with large state spaces to assess the ability to stitch goals.

**(3) Long-horizon reasoning:** Long-horizon reasoning refers to the capability of navigating from a starting state to a goal state that is many steps apart. This challenge is important in many real-world tasks like autonomous driving or assembly, which may require several hours of continuous control or achieving dozens of subtasks. To substantially challenge the long-horizon reasoning ability of offline GCRL methods, we introduce tasks that are more than 5 times longer than previously used ones in terms of both the episode length and the number of subtasks (Table 1).

**(4) Handling stochasticity:** Another prominent challenge in offline GCRL is the ability to deal with stochastic environments. Correctly handling stochasticity is very important in practice, because virtually any real-world environment is stochastic due to partial observability. Yet, many works in offline GCRL assume *deterministic* dynamics to exploit the metric structure of temporal distances (Wang et al., 2023) or to enable hierarchical control (Park et al., 2023), at the cost of being optimistically biased in stochastic environments (Wang et al., 2023; Park et al., 2023). Correctly handling environment stochasticity while fully exploiting the recursive subgoal structure of GCRL remains an open problem. Since most previous tasks used to evaluate offline GCRL methods have deterministic dynamics (Table 1), we introduce several challenging tasks with stochastic dynamics in this benchmark.

## 6 Design Principles

Next, we discuss the design principles underlying our benchmark tasks. The tasks are intended to exercise the major challenges in offline GCRL in the previous section, while providing a set of high-quality tasks that provide not only a toolkit for algorithms research and evaluation, but also a platform for vividly illustrating the capabilities of offline GCRL with compelling and complex domains.

**(1) Realistic and exciting tasks:** The tasks should be realistic yet exciting enough while posing diverse challenges in offline goal-conditioned RL. Imagine a robot arm watching random movements of a puzzle and then solving it zero-shot at test time, a humanoid robot navigating through a labyrinth, or an agent painting cool pictures using different types of brushes. In this benchmark, we design new tasks such that these kinds of exciting behaviors can naturally emerge when an RL agent properly stitches different trajectory segments together (up to 24; see Table 1) or successfully generalizes. At the same time, we make sure our tasks exhaustively cover major challenges in offline goal-conditioned RL, such as long-horizon reasoning, stochastic control, and combinatorial generalization via goal stitching (Section 5).

**(2) Appropriate difficulty:** The tasks and datasets should have appropriate levels of difficulty to properly evaluate different algorithms. In other words, they should be of high quality *for benchmarking*. No matter how intricate or compelling a task is, it will fail to provide a useful signal for benchmarking if it is too easy, too hard, unsolvable from the given dataset, or does not clearly distinguish between more or less effective methods. In this work, we carefully curate and adjust the difficulty of each task and dataset such that they can provide effective guidance for algorithms research. For some tasks, we provide multiple versions with varying difficulty, all the way up to tasks that are difficult to solve with current methods, so that the same benchmark can continuously be used to develop new methods even in the future. Our rule of thumb is to have, for each type of tasks, at least one task where the current state-of-the-art offline GCRL method achieves a success rate of 20-30%. This ensures that the task is solvable from the dataset while leaving significant room for improvement.

**(3) Controllable datasets:** The benchmark should provide tools to control and adjust datasets for scientific research and ablation studies. Verifying the effectiveness of algorithms in real-world problems is surely important. However, for algorithms research, it is equally, if not more, important to provide analysis tools to enable a rigorous, scientific understanding of challenges and algorithms. Hence, instead of employing fixed, human-collected data, which does not always provide clear benchmarking signals for algorithms research, we choose to focus on simulated environments and synthetic datasets that we have full control of, and provide tools to reproduce and adjust them easily. We note that many algorithmic ideas in RL that have made a major impact, such as DQN (Mnih et al., 2013), PPO (Schulman et al., 2017), and CQL (Kumar et al., 2020), were originally developed in simulated environments. Even in natural language processing, studies on synthetic, controlled datasets have revealed the mechanisms and limitations of language models with scientific evidence (Allen-Zhu & Li, 2023a), and these insights have transferred to real scenarios (Allen-Zhu & Li, 2023b). We demonstrate how such controllability of datasets reveals challenges and design principles in offline GCRL in Section 8.2.

**(4) Minimal compute requirements:** The tasks should be designed to minimize *unnecessary* computational overhead so that as many researchers as possible, including those from small labs and underprivileged backgrounds, can quickly iterate on their new *algorithmic* ideas. This does not mean that the tasks should be easy (indeed, some of our tasks are very challenging (but solvable)!); it means the benchmark should focus mainly on algorithmic challenges (*e.g.*, not requiring high-resolution image processing). In our benchmark, we provide both state- and pixel-based observations whenever possible, and minimize the size of image observations (up to $64 \times 64 \times 3$) to reduce the computational burden. Moreover, we carefully adjust colors, transparency, and lighting for image-based tasks to enable pixel-based control without high-resolution images or multiple views.

**(5) High code quality:** The reference implementations should be very clean and well-tuned so that researchers can directly use our implementations to build their ideas, and the benchmark should be very easy to set up. Our benchmark environments only depend on MuJoCo (Todorov et al., 2012) and do not require any other dependencies (Table 1). For reference implementations, we minimize the number of file dependencies for each algorithm, largely following the spirit of the single-file implementations of recent RL libraries (Huang et al., 2022; Tarasov et al., 2023), while maintaining a minimal amount of additional modularity. We also extensively test and tune different design choices and hyperparameters for each offline GCRL algorithm, to the degree that several methods achieve even better performances than their original performances reported on previous benchmarks (Table 3).

# 7 ENVIRONMENTS, TASKS, AND DATASETS

We now introduce the environments, tasks, and datasets in our benchmark. They can be broadly categorized into three groups: locomotion, manipulation, and drawing. We provide a separate validation dataset for each dataset, and most tasks support *both* state- and pixel-based observations. Videos are available at https://seohong.me/projects/ogbench.

**Evaluation.** Each task in OGBench accompanies five pre-defined state-goal pairs for evaluation (Appendix H). Performance is measured by the average success rate across the five evaluation goals. For each pre-define state-goal pair, we perform multiple rollouts with slightly randomized initial and goal states. In each evaluation episode, a goal $g \in \mathcal{S}$ (which is simply another state; see Section 2) is given to the agent, and the episode immediately terminates when the agent reaches the goal. Each task has its own goal success criterion, which we describe in Appendix G.1.

## 7.1 LOCOMOTION TASKS

We provide four types of locomotion environments, **PointMaze**, **AntMaze**, **HumanoidMaze**, and **AntSoccer**, with diverse variants. These environments are designed to test the agent's long-horizon and hierarchical reasoning abilities. They are based on the MuJoCo simulator (Todorov et al., 2012).

**PointMaze (`pointmaze`), AntMaze (`antmaze`), and HumanoidMaze (`humanoidmaze`).** Maze navigation is one of the most widely used tasks for benchmarking offline GCRL algorithms. We provide three different types of maze navigation tasks: Point-Maze, which involves controlling a 2-D point mass, AntMaze, which involves controlling a quadrupedal

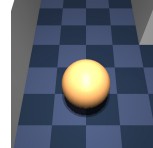 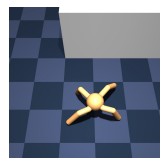 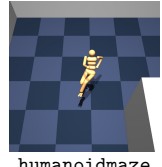

`pointmaze`    `antmaze`    `humanoidmaze`

Ant agent with 8 degrees of freedom (DoF), and HumanoidMaze, which involves controlling a much more complex 21-DoF Humanoid agent. The aim of these tasks is to control the agent to reach a goal location in the given maze. The agent must learn both the high-level maze navigation and low-level locomotion skills that involve high-dimensional control, purely from diverse offline trajectories.

In our benchmark, we substantially extend the original PointMaze and AntMaze tasks proposed by D4RL (Fu et al., 2020). Unlike the original D4RL tasks, which only support Point and Ant agents and do not challenge stitching[4], stochasticity, or pixel-based control, we support Humanoid control, pixel-based observations, and multi-goal evaluation, while providing more challenging and diverse types of mazes and datasets. The supported maze types are as follows:

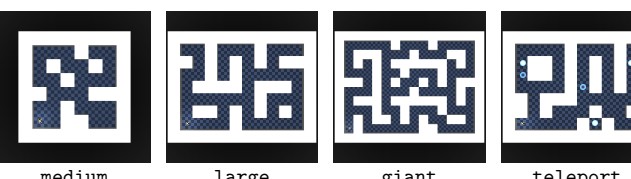

medium     large     giant     teleport

- `medium`: This is the smallest maze, with the same layout as the original `medium` maze in D4RL.
- `large`: This is a larger maze, with the same layout as the original `large` maze in D4RL.
- `giant`: This is the largest maze, twice the size of `large`. It has the same size as the previous `antmaze-ultra` maze by Jiang et al. (2023), but its layout is more challenging and contains longer paths that require up to 3000 environment steps (in the case of Humanoid). This maze is designed to substantially challenge the long-horizon reasoning capability of the agent.
- `teleport`: This maze is specially designed to challenge the agent's ability to handle environment stochasticity. It has the same size as `large`, but contains multiple *stochastic* teleporters. If the agent enters a black hole, it is immediately sent to a randomly chosen white hole. However, since one of the three white holes is a dead-end, there is always a risk in taking a teleporter. The agent therefore must learn to avoid the black holes, without being optimistically biased by "lucky" outcomes.

On these mazes, we collect datasets with a low-level directional policy trained via SAC (Haarnoja et al., 2018) and a high-level waypoint controller. For each maze type, we provide three types of datasets that pose different kinds of challenges (the figures below show example trajectories in these datasets):

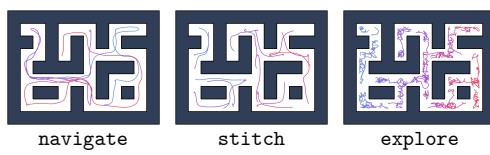

navigate     stitch     explore

- `navigate`: This is the standard dataset, collected by a noisy expert policy that navigates the maze by repeatedly reaching randomly sampled goals.
- `stitch`: This dataset is designed to challenge the agent's stitching ability. It consists of short goal-reaching trajectories, where the length of each trajectory is at most 4 cell units. Hence, the agent must be able to stitch multiple trajectories (up to 8) together to complete the tasks.
- `explore`: This dataset is designed to test whether the agent can learn navigation skills from extremely low-quality (yet high-coverage) data. It consists of random exploratory trajectories, collected by commanding the low-level policy with random directions re-sampled every 10 steps, with a large amount of action noise.

We provide two types of observation modalities:

- States: This is the default setting, where the agent has access to the full low-dimensional state representation, including its current $x$-$y$ position.
- Pixels (`visual`): This requires pure pixel-based control, where the agent only receives $64 \times 64 \times 3$ RGB images rendered from

colored maze     pixels

---

[4]See Ghugare et al. (2024) for the details about this point.

a third-person camera viewpoint. Following Park et al. (2023), we color the floor to enable the agent to infer its location from the images, obviating the need for a potentially expensive memory component. However, unlike Park et al. (2023), which additionally provides proprioceptive information, we do **not** provide any low-dimensional state information like joint angles; the agent must learn *purely* from image observations.

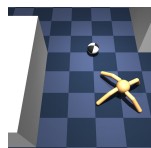

antsoccer

**AntSoccer** (`antsoccer`). To provide a more diverse type of locomotion task beyond simple maze navigation, we introduce a new locomotion task, AntSoccer. This task involves controlling an Ant agent to dribble a soccer ball. It is inspired by the `quadruped-fetch` task in the DeepMind Control suite (Tassa et al., 2018). AntSoccer is significantly harder than AntMaze because the agent must also carefully control the ball while navigating the environment. We provide two maze types: `arena`, which is an open space without walls, and `medium`, which is the same maze as the `medium` one in AntMaze. For datasets, we provide `navigate` and `stitch`. The `navigate` datasets consist of trajectories where the agent repeatedly approaches the ball and dribbles it to random locations. The `stitch` datasets consist of two different types of trajectories, maze navigation without the ball and dribbling with the ball near the agent, so that stitching is required to complete the full task. AntSoccer only supports state-based observations.

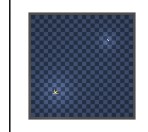
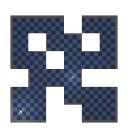

arena      medium

### 7.2 MANIPULATION TASKS

We provide a manipulation suite with three types of robotic manipulation tasks, **Cube**, **Scene**, and **Puzzle**, with diverse difficulties and complexities. They are designed to test the agent's object manipulation, sequential generalization, and combinatorial generalization abilities. These environments are based on the MuJoCo simulator (Todorov et al., 2012) and a 6-DoF UR5e robot arm (Zakka et al., 2022). On these tasks, we provide "play"-style datasets (`play`) (Lynch et al., 2019) collected by **non-Markovian** expert policies with temporally correlated noise. To support more diverse types of research (*e.g.*, dataset ablation studies), we additionally provide more noisy datasets (`noisy`) collected by Markovian expert policies with uncorrelated Gaussian noise, which we describe in Appendix D.

For all manipulation tasks, we support *both* state-based observations and pixel-based observations with $64 \times 64 \times 3$ RGB camera images. For pixel observations, we adjust colors and make the arm transparent to ensure full observability. The transparent arm in the figure might *appear* challenging, but the colors and transparency are carefully adjusted to minimize difficulties in visual perception, to the extent that some methods achieve even better performance with pixels (see Table 2).

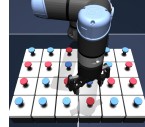
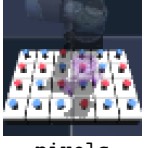

states      pixels

**Cube** (`cube`). This task involves pick-and-place manipulation of cube blocks, whose goal is to control a robot arm to arrange cubes into designated configurations. We provide four variants, `single`, `double`, `triple`, and `quadruple`, with different numbers (1-4) of cubes. We provide "play"-style datasets collected by a scripted policy that repeatedly picks a random block and places it in other random locations or on another block. At test time, the agent is given goal configurations that require moving, stacking, swapping, or permuting cube blocks. Hence, the agent must learn not only generalizable multi-object pick-and-place behaviors from unstructured random trajectories in the dataset, but also long-term plans to achieve the tasks (*e.g.*, permuting blocks requires non-trivial sequential and logical reasoning).

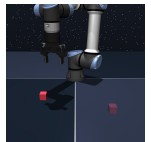
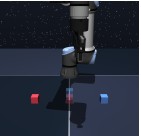

single      double

triple      quadruple

**Scene** (`scene`). This task is designed to challenge the sequential, long-horizon reasoning capabilities of the agent. It involves manipulating diverse everyday objects, such as a cube block, a window, a drawer, and two button locks, where pressing a button toggles the **lock** status of the corresponding object (the drawer or window). We provide "play"-style datasets collected by scripted policies that randomly interact with these objects. At test time, the agent is commanded to arrange the objects into a desired configuration. Evaluation tasks require a significant degree of sequential reasoning: for instance, some tasks require unlocking the drawer, opening it, putting the cube in the drawer, and closing it again (see the figure above), and the longest task involves eight atomic behaviors. Hence, the agent must be able to plan and sequentially combine learned manipulation skills.

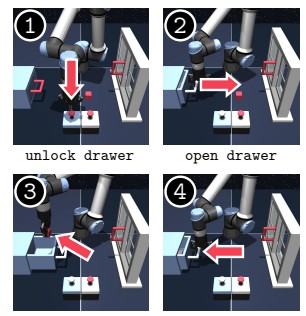

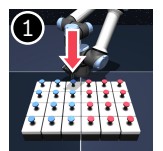 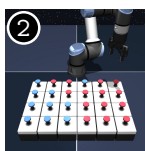

scene

**Puzzle** (`puzzle`). This task is designed to test the combinatorial generalization abilities of the agent. It requires solving the "Lights Out" puzzle[5] with a robot arm. The puzzle consists of a two-dimensional array of buttons (*e.g.*, a $4 \times 6$ grid), where pressing a button toggles the colors of the pressed button and the buttons adjacent to it (typically four, except on the edges and corners; see videos). The goal is to achieve a desired configuration of colors (*e.g.*, turning all the buttons blue) by pressing an appropriate combination of buttons. Since

puzzle

these buttons are implemented in the MuJoCo simulator, the agent must control a robot arm to *physically* press the buttons. We provide four levels of difficulty, 3x3, 4x4, 4x5, and 4x6, with different grid sizes. The datasets are collected by a scripted policy that randomly presses buttons in arbitrary sequences. Given the enormous state space of this task (with up to $2^{24} = 16{,}777{,}216$ distinct button states), the agent must achieve *combinatorial* generalization while mastering low-level continuous control. Some evaluation task in the hardest puzzle requires pressing more than 20 buttons, which also substantially challenges the long-horizon reasoning capabilities of the agent. This might sound very challenging (and it is!), but we provide different levels and enough data to ensure that they provide meaningful research signals and are solvable (see the results in Section 8).

## 7.3 DRAWING TASKS

**Powderworld** (`powderworld`). To provide more diverse tasks beyond robotic locomotion or manipulation, we introduce a *drawing* task, Powderworld (Frans & Isola, 2023)[6], which presents unique challenges with extremely high intrinsic dimensionality. The goal of Powderworld is to draw a target picture on a $32 \times 32$ grid using different types of "powder" brushes, where each powder brush has a distinct physical property corresponding to a unique element. For example, the "sand" brush falls down and piles up, and the "fire" brush burns combustible elements like "plant." We provide three versions of tasks, `easy`, `medium`, and `hard`, with different numbers of available elements (2, 5, and 8 elements, respectively). The datasets are collected by a random policy that keeps drawing arbitrary shapes with random brushes. This Powderworld task poses unique challenges that

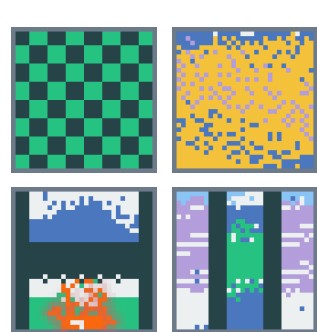

powderworld

are distinct from the other tasks in the benchmark. First, the agent must deal with the high *intrinsic* dimensionality of the states, which presents a substantial challenge in representation learning. Second, since the transitions of powder elements are mostly stochastic and unpredictable, the agent must be able to correctly handle environment stochasticity. Third, the agent must achieve a high degree of generalization and sequential reasoning through a deep understanding of the physics, in order to complete symmetrical, orderly test-time drawing tasks from random, chaotic data.

---

[5] https://en.wikipedia.org/wiki/Lights_Out_(game)
[6] Play here: https://kvfrans.com/powder/

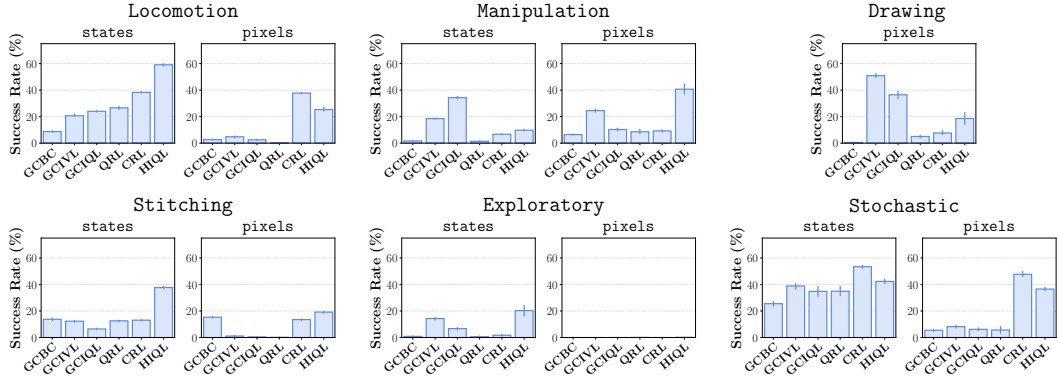

Figure 2: **Benchmarking offline GCRL methods.** We report the performances of six offline GCRL methods (GCBC, GCIVL, GCIQL, QRL, CRL, and HIQL), aggregated by different dataset categories (see Table 6 for the category list). The results are averaged over the tasks in each category, and then over 8 seeds (4 seeds for pixel-based tasks). Error bars denote 95% bootstrap confidence intervals. See Table 2 for the full results. In general, HIQL (a method that involves hierarchical policy extraction) tends to achieve strong performance across the board. Among the non-hierarchical methods, CRL tends to work best in locomotion tasks and GCIVL and GCIQL tend to work best in the others.

## 8 RESULTS

We now present and discuss the benchmarking results of existing offline goal-conditioned RL algorithms on OGBench.

### 8.1 ALGORITHMS

We benchmark six representative offline GCRL algorithms: goal-conditioned behavioral cloning (**GCBC**) (Lynch et al., 2019; Ghosh et al., 2021), goal-conditioned implicit {V, Q}-learning (**GCIVL** and **GCIQL**) (Kostrikov et al., 2022; Park et al., 2023), quasimetric RL (**QRL**) (Wang et al., 2023), contrastive RL (**CRL**) (Eysenbach et al., 2022), and hierarchical implicit Q-learning (**HIQL**) (Park et al., 2023). GCBC is the simplest goal-conditioned behavioral cloning method. GCIVL and GCIQL are offline GCRL algorithms that approximates the optimal value function using an expectile regression (Newey & Powell, 1987). QRL is a non-traditional GCRL method that fits a quasimetric value function with a dual objective. CRL is a "one-step" RL algorithm that fits a Monte Carlo value function via contrastive learning and performs one-step policy improvement. HIQL is a hierarchical RL method that extracts a two-level hierarchical policy from a single GCIVL value function. For benchmarking, we perform a similar amount of hyperparameter tuning for each method to ensure fair comparison. We refer the reader to Appendices F and G for the details.

### 8.2 BENCHMARKING RESULTS AND Q&AS

We present the full benchmarking results in Table 2 in Appendix B. Figure 2 summarizes the results by showing performances grouped by different task categories. Performances are measured by average (binary) success rates on the five test-time goals of each task. We train the agents for 1M gradient steps (500K for pixel-based tasks), and average the results over 8 seeds (4 seeds for pixel-based tasks). We discuss the results through Q&As (we refer to Appendix B for **more Q&As**).

**Q: Which method works best in general?**

**A:** While no single method dominates the others across all categories in Figure 2. HIQL (a method that involves hierarchical policy extraction) tends to achieve particularly strong performance among the benchmarked methods, especially in locomotion and visual manipulation tasks. Among the non-hierarchical methods, CRL tends to work best in locomotion tasks and GCIQL tends to work best in manipulation tasks. In the drawing tasks, GCIVL performs the best.

**Q: There seem to be a lot of datasets. What should I use for my research?**

**A:** For general offline GCRL algorithms research, we recommend starting with more "regular" datasets, such as `antmaze-{large, giant}-navigate`, `humanoidmaze-medium-navigate`, `cube-{single, double}-play`, `scene-play`, and `puzzle-3x3-play`. From there, depending on the performance on these tasks, try harder versions of them or more challenging tasks, such as `humanoidmaze-giant`, `antsoccer`, `puzzle-{4x4, 4x5, 4x6}`, and `powderworld`.

We also provide more specialized datasets that pose specific challenges in offline GCRL (Section 5). **For stitching,** try the `stitch` datasets in the locomotion suite as well as complex manipulation tasks that require stitching (*e.g.*, `puzzle`). **For long-horizon reasoning,** consider `humanoidmaze-giant`, which has the longest episode length, and `puzzle-4x6`, which has the most semantic steps. **For stochastic control,** try `antmaze-teleport`, which is specifically designed to challenge optimistically biased methods, and `powderworld`, which has unpredictable, stochastic dynamics. **For learning from highly suboptimal data,** consider `antmaze-explore` as well as the `noisy` datasets in the manipulation suite, which features high suboptimality and high coverage.

## 9 RESEARCH OPPORTUNITIES

In this section and Appendix C, we discuss potential research ideas and open questions.

**Be the first to solve unsolved tasks!** While all environments in OGBench have at least one variant that current methods can solve to some degree, there are still a number of challenging tasks on which no existing method achieves non-trivial performance, such as `humanoidmaze-giant`, `cube-triple`, `puzzle-4x5`, `powderworld-hard`, and more. We ensure that sufficient data is available for those tasks (which is estimated from the amount needed to solve their easier versions). We invite researchers to take on these challenges and push the limits of offline GCRL with better algorithms.

**How can we develop a policy that *generalizes* well at test time?** In our experiments, we found hierarchical RL methods (*e.g.*, HIQL) to work especially well in several tasks. Among several potential explanations, we hypothesize that this is mainly because hierarchical RL reduces learning complexity by having two policies specialized in different things, which makes both policies *generalize* better at evaluation time. After all, test-time generalization is known to be one of the major bottlenecks in offline RL (Park et al., 2024a). But, are hierarchies really necessary to achieve good test-time generalization? Can we develop a non-hierarchical method that enjoys the same benefit by exploiting the subgoal structure of offline GCRL? This would be especially beneficial, not just because it is simpler, but also because it can potentially yield better, unified *representations* that can potentially serve as a "foundation model" for fine-tuning.

**Can we develop a method that works well across all categories?** Our benchmarking results reveal that no method consistently performs best across the board. HIQL tends to achieve strong performance but struggles in pixel-based locomotion and state-based manipulation. GCIQL shows strong performance in state-based manipulation, but struggles in locomotion. CRL exhibits the opposite trend: it excels in locomotion but underperforms in manipulation. Is there a way to combine only the strengths of these methods to achieve the best performance across all types of tasks?

## 10 OUTLOOK

In this work, we introduced OGBench, a new benchmark designed to advance algorithms research in offline goal-conditioned RL. With the experimental results, we now revisit the very first question posed in this paper: "Why offline goal-conditioned RL?"

We hypothesize that offline goal-conditioned holds significant potential as a recipe for *general-purpose RL pre-training*, which yields richer and more diverse behaviors than (arguably more prevalent) generative pre-training, such as behavioral cloning and next-token prediction. Our experiments, albeit preliminary, show that even current offline GCRL algorithms can to some extent acquire effective policies for exceptionally long-horizon tasks entirely with sparse rewards, using data that is highly suboptimal. These results are not limited to toy example domains, but show up across a range of realistic simulated robotics and game-like settings in our benchmark. While generative objectives might capture the data distribution, offline GCRL can learn policies that actually achieve complex outcomes (such as beating a puzzle game) that could not be achieved simply by copying random data.

However, current offline GCRL algorithms also have limitations. As shown in our results, they often struggle with long-horizon, high-dimensional tasks and those that require stitching, and no single method consistently outperforms others across all tasks. This suggests that we have not yet found the ideal algorithm that can fully realize the promise of offline GCRL as general-purpose RL pre-training. The first step toward finding such an ideal algorithm is to set up a solid benchmark that sufficiently challenges the limits of offline GCRL from diverse perspectives. We believe OGBench provides this foundation, and will lead to the development of performant, scalable offline GCRL algorithms that enable building foundation models for general-purpose behaviors.

## ACKNOWLEDGMENTS

We thank Kevin Zakka for providing the initial codebase for the manipulation environments and helping with MuJoCo implementations, Vivek Myers for providing a JAX-based QRL implementation, and Colin Li, along with the members of the RAIL lab, for helpful discussions. This work was partly supported by the Korea Foundation for Advanced Studies (KFAS), National Science Foundation Graduate Research Fellowship Program under Grant No. DGE 2146752, ONR under N00014-20-1-2383 and N00014-22-1-2773, and Qualcomm. This research used the Savio computational cluster resource provided by the Berkeley Research Computing program at UC Berkeley.

## REPRODUCIBILITY STATEMENT

We provide the full implementation details in Appendix G. We provide the code as well as the exact command-line flags to reproduce the entire benchmark table, datasets, and expert policies at https://github.com/seohongpark/ogbench.

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

Table 2: **Full benchmark table.** We report each method's average (binary) success rate (%) across the five test-time goals on each task. The results are averaged over 8 seeds (4 seeds for pixel-based tasks), and we report the standard deviations after the $\pm$ sign. Numbers at or above 95% of the best value in the row are highlighted in bold.

| Environment | Dataset Type | Dataset | GCBC | GCIVL | GCIQL | QRL | CRL | HIQL |
|---|---|---|---|---|---|---|---|---|
| pointmaze | navigate | pointmaze-medium-navigate-v0 | $9_{\pm 6}$ | $63_{\pm 6}$ | $53_{\pm 8}$ | $\mathbf{82}_{\pm 5}$ | $29_{\pm 7}$ | $\mathbf{79}_{\pm 5}$ |
| | | pointmaze-large-navigate-v0 | $29_{\pm 6}$ | $45_{\pm 5}$ | $34_{\pm 3}$ | $\mathbf{86}_{\pm 9}$ | $39_{\pm 7}$ | $58_{\pm 5}$ |
| | | pointmaze-giant-navigate-v0 | $1_{\pm 2}$ | $0_{\pm 0}$ | $0_{\pm 0}$ | $\mathbf{68}_{\pm 7}$ | $27_{\pm 10}$ | $46_{\pm 9}$ |
| | | pointmaze-teleport-navigate-v0 | $25_{\pm 3}$ | $\mathbf{45}_{\pm 3}$ | $24_{\pm 7}$ | $4_{\pm 4}$ | $24_{\pm 6}$ | $18_{\pm 4}$ |
| | stitch | pointmaze-medium-stitch-v0 | $23_{\pm 18}$ | $70_{\pm 14}$ | $21_{\pm 9}$ | $\mathbf{80}_{\pm 12}$ | $0_{\pm 1}$ | $74_{\pm 6}$ |
| | | pointmaze-large-stitch-v0 | $7_{\pm 5}$ | $12_{\pm 6}$ | $31_{\pm 2}$ | $\mathbf{84}_{\pm 15}$ | $0_{\pm 0}$ | $13_{\pm 6}$ |
| | | pointmaze-giant-stitch-v0 | $0_{\pm 0}$ | $0_{\pm 0}$ | $0_{\pm 0}$ | $\mathbf{50}_{\pm 8}$ | $0_{\pm 0}$ | $0_{\pm 0}$ |
| | | pointmaze-teleport-stitch-v0 | $31_{\pm 9}$ | $\mathbf{44}_{\pm 2}$ | $25_{\pm 3}$ | $9_{\pm 5}$ | $4_{\pm 3}$ | $34_{\pm 4}$ |
| antmaze | navigate | antmaze-medium-navigate-v0 | $29_{\pm 4}$ | $72_{\pm 8}$ | $71_{\pm 4}$ | $88_{\pm 3}$ | $\mathbf{95}_{\pm 1}$ | $\mathbf{96}_{\pm 1}$ |
| | | antmaze-large-navigate-v0 | $24_{\pm 2}$ | $16_{\pm 5}$ | $34_{\pm 4}$ | $75_{\pm 6}$ | $83_{\pm 4}$ | $\mathbf{91}_{\pm 2}$ |
| | | antmaze-giant-navigate-v0 | $0_{\pm 0}$ | $0_{\pm 0}$ | $0_{\pm 0}$ | $14_{\pm 3}$ | $16_{\pm 3}$ | $\mathbf{65}_{\pm 5}$ |
| | | antmaze-teleport-navigate-v0 | $26_{\pm 3}$ | $39_{\pm 3}$ | $35_{\pm 5}$ | $35_{\pm 5}$ | $\mathbf{53}_{\pm 2}$ | $42_{\pm 3}$ |
| | stitch | antmaze-medium-stitch-v0 | $45_{\pm 11}$ | $44_{\pm 6}$ | $29_{\pm 6}$ | $59_{\pm 7}$ | $53_{\pm 6}$ | $\mathbf{94}_{\pm 1}$ |
| | | antmaze-large-stitch-v0 | $3_{\pm 3}$ | $18_{\pm 2}$ | $7_{\pm 2}$ | $18_{\pm 2}$ | $11_{\pm 2}$ | $\mathbf{67}_{\pm 5}$ |
| | | antmaze-giant-stitch-v0 | $0_{\pm 0}$ | $0_{\pm 0}$ | $0_{\pm 0}$ | $0_{\pm 0}$ | $0_{\pm 0}$ | $\mathbf{2}_{\pm 2}$ |
| | | antmaze-teleport-stitch-v0 | $31_{\pm 6}$ | $\mathbf{39}_{\pm 3}$ | $17_{\pm 2}$ | $24_{\pm 5}$ | $31_{\pm 4}$ | $36_{\pm 2}$ |
| | explore | antmaze-medium-explore-v0 | $2_{\pm 1}$ | $19_{\pm 3}$ | $13_{\pm 2}$ | $1_{\pm 1}$ | $3_{\pm 2}$ | $\mathbf{37}_{\pm 10}$ |
| | | antmaze-large-explore-v0 | $0_{\pm 0}$ | $\mathbf{10}_{\pm 3}$ | $0_{\pm 0}$ | $0_{\pm 0}$ | $0_{\pm 0}$ | $4_{\pm 5}$ |
| | | antmaze-teleport-explore-v0 | $2_{\pm 1}$ | $32_{\pm 2}$ | $7_{\pm 3}$ | $2_{\pm 2}$ | $20_{\pm 2}$ | $\mathbf{34}_{\pm 15}$ |
| humanoidmaze | navigate | humanoidmaze-medium-navigate-v0 | $8_{\pm 2}$ | $24_{\pm 2}$ | $27_{\pm 2}$ | $21_{\pm 8}$ | $60_{\pm 4}$ | $\mathbf{89}_{\pm 2}$ |
| | | humanoidmaze-large-navigate-v0 | $1_{\pm 0}$ | $2_{\pm 1}$ | $2_{\pm 1}$ | $5_{\pm 1}$ | $24_{\pm 4}$ | $\mathbf{49}_{\pm 4}$ |
| | | humanoidmaze-giant-navigate-v0 | $0_{\pm 0}$ | $0_{\pm 0}$ | $0_{\pm 0}$ | $1_{\pm 0}$ | $3_{\pm 2}$ | $\mathbf{12}_{\pm 4}$ |
| | stitch | humanoidmaze-medium-stitch-v0 | $29_{\pm 5}$ | $12_{\pm 2}$ | $12_{\pm 3}$ | $18_{\pm 2}$ | $36_{\pm 2}$ | $\mathbf{88}_{\pm 2}$ |
| | | humanoidmaze-large-stitch-v0 | $6_{\pm 3}$ | $1_{\pm 1}$ | $0_{\pm 0}$ | $3_{\pm 1}$ | $4_{\pm 1}$ | $\mathbf{28}_{\pm 3}$ |
| | | humanoidmaze-giant-stitch-v0 | $0_{\pm 0}$ | $0_{\pm 0}$ | $0_{\pm 0}$ | $0_{\pm 0}$ | $0_{\pm 0}$ | $\mathbf{3}_{\pm 2}$ |
| antsoccer | navigate | antsoccer-arena-navigate-v0 | $5_{\pm 1}$ | $47_{\pm 3}$ | $50_{\pm 2}$ | $8_{\pm 2}$ | $23_{\pm 2}$ | $\mathbf{58}_{\pm 2}$ |
| | | antsoccer-medium-navigate-v0 | $2_{\pm 0}$ | $4_{\pm 1}$ | $7_{\pm 1}$ | $2_{\pm 2}$ | $3_{\pm 1}$ | $\mathbf{13}_{\pm 2}$ |
| | stitch | antsoccer-arena-stitch-v0 | $\mathbf{24}_{\pm 8}$ | $21_{\pm 3}$ | $2_{\pm 0}$ | $1_{\pm 1}$ | $1_{\pm 0}$ | $15_{\pm 1}$ |
| | | antsoccer-medium-stitch-v0 | $2_{\pm 1}$ | $1_{\pm 0}$ | $0_{\pm 0}$ | $0_{\pm 0}$ | $0_{\pm 0}$ | $\mathbf{4}_{\pm 1}$ |
| visual-antmaze | navigate | visual-antmaze-medium-navigate-v0 | $11_{\pm 2}$ | $22_{\pm 2}$ | $11_{\pm 1}$ | $0_{\pm 0}$ | $\mathbf{94}_{\pm 1}$ | $\mathbf{93}_{\pm 4}$ |
| | | visual-antmaze-large-navigate-v0 | $4_{\pm 0}$ | $5_{\pm 1}$ | $4_{\pm 1}$ | $0_{\pm 0}$ | $\mathbf{84}_{\pm 1}$ | $53_{\pm 9}$ |
| | | visual-antmaze-giant-navigate-v0 | $0_{\pm 0}$ | $1_{\pm 1}$ | $0_{\pm 0}$ | $0_{\pm 0}$ | $\mathbf{47}_{\pm 2}$ | $6_{\pm 4}$ |
| | | visual-antmaze-teleport-navigate-v0 | $5_{\pm 1}$ | $8_{\pm 1}$ | $6_{\pm 1}$ | $6_{\pm 3}$ | $\mathbf{48}_{\pm 2}$ | $37_{\pm 2}$ |
| | stitch | visual-antmaze-medium-stitch-v0 | $67_{\pm 4}$ | $6_{\pm 2}$ | $2_{\pm 0}$ | $0_{\pm 0}$ | $69_{\pm 2}$ | $\mathbf{87}_{\pm 2}$ |
| | | visual-antmaze-large-stitch-v0 | $24_{\pm 3}$ | $1_{\pm 1}$ | $0_{\pm 0}$ | $1_{\pm 1}$ | $11_{\pm 3}$ | $\mathbf{28}_{\pm 2}$ |
| | | visual-antmaze-giant-stitch-v0 | $\mathbf{0}_{\pm 0}$ | $\mathbf{0}_{\pm 0}$ | $\mathbf{0}_{\pm 0}$ | $\mathbf{0}_{\pm 0}$ | $\mathbf{0}_{\pm 0}$ | $\mathbf{0}_{\pm 0}$ |
| | | visual-antmaze-teleport-stitch-v0 | $32_{\pm 3}$ | $1_{\pm 1}$ | $1_{\pm 0}$ | $1_{\pm 2}$ | $32_{\pm 6}$ | $\mathbf{37}_{\pm 4}$ |
| | explore | visual-antmaze-medium-explore-v0 | $\mathbf{0}_{\pm 0}$ | $\mathbf{0}_{\pm 0}$ | $\mathbf{0}_{\pm 0}$ | $\mathbf{0}_{\pm 0}$ | $\mathbf{0}_{\pm 0}$ | $\mathbf{0}_{\pm 0}$ |
| | | visual-antmaze-large-explore-v0 | $\mathbf{0}_{\pm 0}$ | $\mathbf{0}_{\pm 0}$ | $\mathbf{0}_{\pm 0}$ | $\mathbf{0}_{\pm 0}$ | $\mathbf{0}_{\pm 0}$ | $\mathbf{0}_{\pm 0}$ |
| | | visual-antmaze-teleport-explore-v0 | $0_{\pm 0}$ | $0_{\pm 0}$ | $0_{\pm 0}$ | $0_{\pm 0}$ | $1_{\pm 0}$ | $\mathbf{19}_{\pm 8}$ |
| visual-humanoidmaze | navigate | visual-humanoidmaze-medium-navigate-v0 | $0_{\pm 0}$ | $0_{\pm 0}$ | $0_{\pm 0}$ | $0_{\pm 0}$ | $\mathbf{1}_{\pm 0}$ | $0_{\pm 0}$ |
| | | visual-humanoidmaze-large-navigate-v0 | $\mathbf{0}_{\pm 0}$ | $\mathbf{0}_{\pm 0}$ | $\mathbf{0}_{\pm 0}$ | $\mathbf{0}_{\pm 0}$ | $\mathbf{0}_{\pm 0}$ | $\mathbf{0}_{\pm 0}$ |
| | | visual-humanoidmaze-giant-navigate-v0 | $\mathbf{0}_{\pm 0}$ | $\mathbf{0}_{\pm 0}$ | $\mathbf{0}_{\pm 0}$ | $\mathbf{0}_{\pm 0}$ | $\mathbf{0}_{\pm 0}$ | $\mathbf{0}_{\pm 0}$ |
| | stitch | visual-humanoidmaze-medium-stitch-v0 | $\mathbf{1}_{\pm 0}$ | $0_{\pm 0}$ | $0_{\pm 0}$ | $0_{\pm 0}$ | $\mathbf{1}_{\pm 0}$ | $0_{\pm 0}$ |
| | | visual-humanoidmaze-large-stitch-v0 | $\mathbf{0}_{\pm 0}$ | $\mathbf{0}_{\pm 0}$ | $\mathbf{0}_{\pm 0}$ | $\mathbf{0}_{\pm 0}$ | $\mathbf{0}_{\pm 0}$ | $\mathbf{0}_{\pm 0}$ |
| | | visual-humanoidmaze-giant-stitch-v0 | $\mathbf{0}_{\pm 0}$ | $\mathbf{0}_{\pm 0}$ | $\mathbf{0}_{\pm 0}$ | $\mathbf{0}_{\pm 0}$ | $\mathbf{0}_{\pm 0}$ | $\mathbf{0}_{\pm 0}$ |
| cube | play | cube-single-play-v0 | $6_{\pm 2}$ | $53_{\pm 4}$ | $\mathbf{68}_{\pm 6}$ | $5_{\pm 1}$ | $19_{\pm 2}$ | $15_{\pm 3}$ |
| | | cube-double-play-v0 | $1_{\pm 1}$ | $36_{\pm 3}$ | $\mathbf{40}_{\pm 5}$ | $1_{\pm 0}$ | $10_{\pm 2}$ | $6_{\pm 2}$ |
| | | cube-triple-play-v0 | $1_{\pm 1}$ | $1_{\pm 0}$ | $3_{\pm 1}$ | $0_{\pm 0}$ | $\mathbf{4}_{\pm 1}$ | $3_{\pm 1}$ |
| | | cube-quadruple-play-v0 | $\mathbf{0}_{\pm 0}$ | $\mathbf{0}_{\pm 0}$ | $\mathbf{0}_{\pm 0}$ | $\mathbf{0}_{\pm 0}$ | $\mathbf{0}_{\pm 0}$ | $\mathbf{0}_{\pm 0}$ |
| scene | play | scene-play-v0 | $5_{\pm 1}$ | $42_{\pm 4}$ | $\mathbf{51}_{\pm 4}$ | $5_{\pm 1}$ | $19_{\pm 2}$ | $38_{\pm 3}$ |
| puzzle | play | puzzle-3x3-play-v0 | $2_{\pm 0}$ | $6_{\pm 1}$ | $\mathbf{95}_{\pm 1}$ | $1_{\pm 0}$ | $3_{\pm 1}$ | $12_{\pm 2}$ |
| | | puzzle-4x4-play-v0 | $0_{\pm 0}$ | $13_{\pm 2}$ | $\mathbf{26}_{\pm 3}$ | $0_{\pm 0}$ | $0_{\pm 0}$ | $7_{\pm 2}$ |
| | | puzzle-4x5-play-v0 | $0_{\pm 0}$ | $7_{\pm 1}$ | $\mathbf{14}_{\pm 1}$ | $0_{\pm 0}$ | $1_{\pm 0}$ | $4_{\pm 1}$ |
| | | puzzle-4x6-play-v0 | $0_{\pm 0}$ | $10_{\pm 2}$ | $\mathbf{12}_{\pm 1}$ | $0_{\pm 0}$ | $4_{\pm 1}$ | $3_{\pm 1}$ |
| visual-cube | play | visual-cube-single-play-v0 | $5_{\pm 1}$ | $60_{\pm 5}$ | $30_{\pm 5}$ | $41_{\pm 15}$ | $31_{\pm 15}$ | $\mathbf{89}_{\pm 0}$ |
| | | visual-cube-double-play-v0 | $1_{\pm 1}$ | $10_{\pm 2}$ | $1_{\pm 1}$ | $5_{\pm 0}$ | $2_{\pm 1}$ | $\mathbf{39}_{\pm 2}$ |
| | | visual-cube-triple-play-v0 | $15_{\pm 2}$ | $14_{\pm 2}$ | $15_{\pm 1}$ | $16_{\pm 1}$ | $17_{\pm 2}$ | $\mathbf{21}_{\pm 0}$ |
| | | visual-cube-quadruple-play-v0 | $8_{\pm 1}$ | $0_{\pm 0}$ | $7_{\pm 1}$ | $5_{\pm 1}$ | $4_{\pm 1}$ | $\mathbf{14}_{\pm 1}$ |
| visual-scene | play | visual-scene-play-v0 | $12_{\pm 2}$ | $25_{\pm 3}$ | $12_{\pm 2}$ | $10_{\pm 1}$ | $11_{\pm 2}$ | $\mathbf{49}_{\pm 4}$ |
| visual-puzzle | play | visual-puzzle-3x3-play-v0 | $0_{\pm 0}$ | $21_{\pm 1}$ | $1_{\pm 2}$ | $0_{\pm 0}$ | $0_{\pm 0}$ | $\mathbf{73}_{\pm 8}$ |
| | | visual-puzzle-4x4-play-v0 | $10_{\pm 1}$ | $\mathbf{60}_{\pm 5}$ | $16_{\pm 4}$ | $0_{\pm 0}$ | $10_{\pm 6}$ | $\mathbf{60}_{\pm 41}$ |
| | | visual-puzzle-4x5-play-v0 | $5_{\pm 2}$ | $\mathbf{17}_{\pm 1}$ | $7_{\pm 2}$ | $0_{\pm 0}$ | $6_{\pm 1}$ | $13_{\pm 9}$ |
| | | visual-puzzle-4x6-play-v0 | $2_{\pm 1}$ | $\mathbf{15}_{\pm 1}$ | $2_{\pm 1}$ | $0_{\pm 0}$ | $3_{\pm 1}$ | $9_{\pm 6}$ |
| powderworld | play | powderworld-easy-play-v0 | $0_{\pm 0}$ | $\mathbf{99}_{\pm 1}$ | $93_{\pm 5}$ | $12_{\pm 2}$ | $22_{\pm 5}$ | $33_{\pm 9}$ |
| | | powderworld-medium-play-v0 | $1_{\pm 1}$ | $\mathbf{50}_{\pm 4}$ | $16_{\pm 5}$ | $3_{\pm 1}$ | $1_{\pm 1}$ | $22_{\pm 14}$ |
| | | powderworld-hard-play-v0 | $0_{\pm 0}$ | $\mathbf{4}_{\pm 3}$ | $0_{\pm 0}$ | $0_{\pm 0}$ | $0_{\pm 0}$ | $1_{\pm 1}$ |

# A    LIMITATIONS

While OGBench covers a number of challenges in offline goal-conditioned RL, such as long-horizon reasoning, goal stitching, and stochastic control, there exist other challenges that our benchmark does not address. For example, all OGBench tasks assume that the environment dynamics remain

Table 3: **How good are our reference implementations?** Our implementations generally achieve better performance than previously reported ones.

| | D4RL antmaze-large-diverse-v2 | | | D4RL antmaze-large-play-v2 | |
| --- | --- | --- | --- | --- | --- |
| **Method** | **Previously Reported Performance** | **Ours** | **Method** | **Previously Reported Performance** | **Ours** |
| GCBC | 20 (Park et al., 2023) | **41** $_{\pm 7}$ | GCBC | 23 (Park et al., 2023) | **39** $_{\pm 4}$ |
| GCIVL | 51 (Park et al., 2023) | **64** $_{\pm 8}$ | GCIVL | **57** (Park et al., 2023) | **58** $_{\pm 8}$ |
| GCIQL | 30 (Zeng et al., 2023) | **64** $_{\pm 10}$ | GCIQL | 40 (Zeng et al., 2023) | **55** $_{\pm 11}$ |
| QRL | **52**[7] (Zheng et al., 2024b) | 37 $_{\pm 14}$ | QRL | **53**[7] (Zheng et al., 2024b) | 38 $_{\pm 8}$ |
| CRL | 54 (Eysenbach et al., 2022) | **79** $_{\pm 6}$ | CRL | 49 (Eysenbach et al., 2022) | **74** $_{\pm 4}$ |
| HIQL | **88** (Park et al., 2023) | 87 $_{\pm 3}$ | HIQL | **86** (Park et al., 2023) | **87** $_{\pm 2}$ |

the same between the training and evaluation environments. Also, although several OGBench tasks (*e.g.*, Cube, Puzzle, and Powderworld) require unseen goal generalization to some degree, our tasks do not specifically test visual generalization to entirely new objects. Finally, we have made several trade-offs to reduce computational cost and to focus the benchmark on algorithms research at the expense of sacrificing realism to some degree (*e.g.*, the use of the transparent arm in manipulation environments, the use of synthetic (yet fully controllable) datasets, etc.). Nonetheless, we believe OGBench can spur the development of performant offline GCRL *algorithms*, which can then help researchers develop scalable data-driven unsupervised RL pre-training methods for real-world tasks.

## B    ADDITIONAL RESULTS AND Q&AS

We present the full benchmarking results in Table 2, and provide additional Q&As in this section.

**Q: Which methods are good at goal stitching?**

**A:** To see this, we can compare the performances on the `navigate` and `stitch` datasets from the same locomotion task in Table 2. The results suggest that, as expected, full RL-based methods like HIQL (*i.e.*, methods that fit the optimal value function $Q^*$) are better at stitching than one-step RL methods like CRL (*i.e.*, methods that fit the behavioral value function $Q^\beta$). For example, in visual locomotion tasks, the relative performance between HIQL and CRL is reversed on the `stitch` datasets (Figure 2).

**Q: Which methods are good at handling stochasticity?**

**A:** For this, we can compare the performances on the `large` and `teleport` mazes in Table 2, where both have the same maze size, but only the latter involves stochastic transitions that incur risk. Table 2 shows that, value-only methods like HIQL and QRL (*i.e.*, methods that do not have a separate Q function), which are optimistically biased in stochastic environments, struggle relatively more in stochastic `teleport` tasks. In contrast, CRL is generally robust to environment stochasticity, likely because it fits a Monte Carlo value function.

**Q: Which methods are good at handling pixel-based observations?**

**A:** Although state-based and pixel-based observations generally provide the same amount of information, several methods struggle to handle image observations due to additional representational challenges. We can understand how well a method addresses such representational challenges by comparing the performances of corresponding state- and pixel-based tasks. Table 2 shows that CRL is notably robust to the difference in input modalities, likely because it is based on a pure representation learning objective. HIQL also achieves strong performance in pixel-based tasks, especially in visual manipulation tasks. However, these methods are still not perfect at handling image observations; for example, HIQL achieves relatively weak performance on image drawing tasks. We suspect this is due to the difficulty of learning low-dimensional subgoal representations from states of high intrinsic dimensionality.

**Q: How good are our reference implementations?**

**A:** We compare the performance of our reference implementations with previously reported numbers on one of the most commonly used tasks in prior work, D4RL `antmaze-large` (Fu et al., 2020). Table 3 shows the comparison results, with the corresponding numbers taken from the prior works (Eysenbach et al., 2022; Park et al., 2023; Zeng et al., 2023; Zheng et al., 2024b). The results suggest that

---

[7]We note that Zheng et al. (2024b) use a different evaluation scheme based on the maximum performance over evaluation epochs. We report the average performance over the last three evaluation epochs (see Appendix G).

Table 4: **Do not use single-goal evaluation!** Only using a single state-goal pair (a common practice when using D4RL tasks for offline GCRL) can potentially lead to inaccurate conclusions about offline GCRL methods. OGBench always uses multi-goal evaluation. See how the rank between GCIQL and QRL is reversed with multi-goal evaluation on the same `antmaze-large` maze.

| Dataset | GCBC | GCIVL | GCIQL | QRL | CRL | HIQL |
|---|---|---|---|---|---|---|
| D4RL `antmaze-large-diverse-v2` (single-goal evaluation) | 41 ±7 | 64 ±8 | 64 ±10 | 37 ±14 | 79 ±6 | **87** ±3 |
| D4RL `antmaze-large-play-v2` (single-goal evaluation) | 39 ±4 | 58 ±8 | 55 ±11 | 38 ±8 | 74 ±4 | **87** ±2 |
| OGBench `antmaze-large-navigate-v0` (multi-goal evaluation, **ours**) | 24 ±2 | 16 ±5 | 34 ±4 | 75 ±6 | 83 ±4 | **91** ±2 |

our implementations generally achieve better performance than previously reported results, sometimes significantly surpassing them (*e.g.*, CRL).

### Q: Why can't I just use D4RL AntMaze instead of the OGBench one?

**A:** D4RL AntMaze is an excellent task for benchmarking offline RL algorithms. However, it is limited for benchmarking offline *goal-conditioned* RL algorithms because it only involves a single, fixed state-goal pair, and the datasets are tailored to this specific task (Fu et al., 2020). In contrast, OGBench supports multi-goal evaluation (and provides much more diverse types of tasks and datasets!). To empirically demonstrate this difference, we compare the benchmarking results on D4RL `antmaze-large-{diverse, play}` and OGBench `antmaze-large-navigate` in Table 4. The table suggests that single-goal evaluation is indeed limited, and is potentially prone to inaccurate conclusions: for example, see how the ranking between GCIQL and QRL is reversed with multi-goal evaluation on the same `antmaze-large` maze. Moreover, the performance differences between methods are more pronounced in OGBench AntMaze, showing that OGBench provides clearer research signals.

### Q: Have you found any insights on data collection for offline GCRL?

**A:** One of the main features of OGBench is that every task is accompanied by a reproducible and controllable data-generation script. Here, we show one example of how this controllability can lead to practical insights and raise open research questions. In Figure 3, we ablate the strength of Gaussian action noise $\sigma$ added to expert actions on two manipulation tasks (`cube-single-noisy` and `puzzle-3x3-noisy`), and measure how this affects performance.

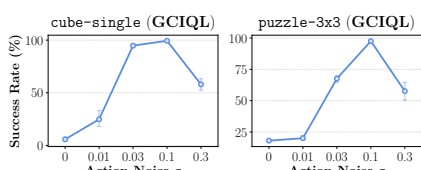

Figure 3: **Datasets *must* be noisy enough.**

The results are quite remarkable: they show that having the right amount of noise (*i.e.*, state coverage) is *very* important for achieving good performance. For example, the performance drops from 99% to 6% if there is no noise in expert actions, even on the most basic cube pick-and-place task. This suggests that, we may need to prioritize coverage much more than optimality when collecting datasets for offline GCRL in the real world as well, and failing to do so may lead to (surprising) failures in learning. Like action noise, we believe there are many other important properties of datasets that significantly affect performance. We believe that our fully transparent, controllable data-generation scripts can facilitate such scientific studies.

## C   MORE CONCRETE RESEARCH QUESTIONS

Here, we list additional, more concrete research questions that researchers may use as a starting point for research in offline GCRL:

- **Why is PointMaze so hard?** Table 2 shows that PointMaze is surprisingly hard, sometimes even harder than AntMaze for some methods. Why is this the case? Moreover, only in PointMaze does QRL significantly outperform the other methods. What causes this difference, and are there any insights we can take from these results?

- **How should we train subgoal representations?** Somewhat weirdly, HIQL struggles much more with state-based observations than pixel-based observations on manipulation tasks. We suspect this is related to subgoal representations, given that HIQL uses an additional learning signal

from the policy loss to further train subgoal representations *only* in pixel-based environments (which we found does not help in state-based environments). HIQL uses a value function-based subgoal representation (Appendix F), but is there a better, more stable way to learn subgoal representations for hierarchical RL and planning?

- **Do we really need the full power of RL?** While learning the optimal $Q^*$ function is in principle better than learning the behavioral $Q^\beta$ function, CRL (which fits $Q^\beta$) significantly outperforms GCIQL (which fits $Q^*$) on locomotion tasks. Why is this the case? Is it a problem with expectile regression in GCIQL or with temporal difference learning itself? In contrast, in manipulation environments, the result suggests the opposite: GCIQL is much better than CRL. Does this mean we do need $Q^*$ in these tasks? Or can it be solvable even with $Q^\beta$ if we use a better behavioral value learning technique than binary NCE in CRL?

- **Why can't we use random goals when training policies?** When training goal-conditioned policies, we found that it is usually better to sample (policy) goals *only* from the future state in the current trajectory (except on `stitch` or `explore` datasets; see Table 10). The fact that this works better even in Scene and Puzzle (which require goal stitching) is a bit surprising, because it means that the policy can still perform goal stitching to some degree even without being explicitly trained on the test-time state-goal pairs. At the same time, it is rather unsatisfying because this ability to stitch goals entirely depends on the seemingly "magical" generalization capabilities of neural networks. Is there a way to train a goal-conditioned policy with random goals while maintaining performance, so that it can perform goal stitching in a principled manner?

- **How can we combine expressive policies with GCRL?** In Appendix D, we show that current offline GCRL methods often struggle with datasets collected by non-Markovian policies in manipulation environments. Handling non-Markovian trajectory data is indeed one of the major challenges in behavioral cloning, for which many recent BC-based methods have been proposed (Zhao et al., 2023; Chi et al., 2023). How can we incorporate these recent advancements in behavioral cloning into offline GCRL?

## D  ADDITIONAL DATASETS

For manipulation tasks (Cube, Scene, and Puzzle), in addition to the main `play` datasets, we additionally provide `noisy` datasets that can potentially be useful for other types of research (*e.g.*, ablation studies on datasets, comparing performances on non-Markovian and Markovian datasets, etc.). The main difference is that the `play` datasets are collected by open-loop, non-Markovian expert policies with temporally correlated noise, while the `noisy` datasets are collected by closed-loop, Markovian expert policies with larger, uncorrelated Gaussian noise. Hence, the `play` datasets generally look more "natural" than the `noisy` datasets, but the latter has higher state coverage (videos).

Table 5 shows the full benchmark results on both the `play` and `noisy` datasets in the manipulation suite. The results suggest that the performances on these two datasets are mostly similar, but several methods struggle to handle narrower and non-Markovian trajectories in `play` datasets (*e.g.*, GCIQL almost perfectly solves `cube-single-noisy` but struggles on `cube-single-play`).

## E  PRIOR WORK IN GOAL-CONDITIONED RL

The problem of reaching any goal from any state has long been considered one of the central problems in reinforcement learning and sequential decision making (Kaelbling, 1993; Schaul et al., 2015; Andrychowicz et al., 2017), owing to its unsupervised nature, simplicity, and generality. There are many unique features of goal-conditioned RL that make it distinct from other (multi-task) RL problems, such as the presence of recursive subgoal structures, metric structures, and probabilistic interpretations. These intriguing properties have led to the development of diverse families of online and offline GCRL algorithms based on hindsight relabeling (Andrychowicz et al., 2017), hierarchical learning (Dayan & Hinton, 1992; Chane-Sane et al., 2021; Li et al., 2022; Park et al., 2023), planning (Savinov et al., 2018; Eysenbach et al., 2019; Nasiriany et al., 2019; Huang et al., 2019; Hoang et al., 2021; Kim et al., 2021; Wang et al., 2024), metric learning (Wang et al., 2023; Park et al., 2024b; Myers et al., 2024), dual optimization (Ma et al., 2022; 2023; Sikchi et al., 2024), weighted behavioral cloning (Yang et al., 2022; 2023; Hejna et al., 2023), generative modeling (Zeng et al., 2023; Hong et al., 2023), and contrastive learning (Eysenbach et al., 2022; Zheng et al.,

Table 5: **Full benchmarking results on additional noisy manipulation datasets.** The table shows the performances on both the `play` and `noisy` datasets in the manipulation suite. We report each method's average (binary) success rate (%) across the five test-time goals on each task. The results are averaged over 8 seeds (4 seeds for pixel-based tasks), and we report the standard deviations after the $\pm$ sign. Numbers at or above 95% of the best value in the row are highlighted in bold.

| Environment | Dataset Type | Dataset | GCBC | GCIVL | GCIQL | QRL | CRL | HIQL |
|---|---|---|---|---|---|---|---|---|
| cube | play | cube-single-play-v0 | $6_{\pm 2}$ | $53_{\pm 4}$ | $\mathbf{68}_{\pm 6}$ | $5_{\pm 1}$ | $19_{\pm 2}$ | $15_{\pm 3}$ |
| | | cube-double-play-v0 | $1_{\pm 1}$ | $36_{\pm 3}$ | $\mathbf{40}_{\pm 5}$ | $1_{\pm 0}$ | $10_{\pm 2}$ | $6_{\pm 2}$ |
| | | cube-triple-play-v0 | $1_{\pm 1}$ | $1_{\pm 0}$ | $3_{\pm 1}$ | $0_{\pm 0}$ | $4_{\pm 1}$ | $3_{\pm 1}$ |
| | | cube-quadruple-play-v0 | $\mathbf{0}_{\pm 0}$ | $\mathbf{0}_{\pm 0}$ | $\mathbf{0}_{\pm 0}$ | $\mathbf{0}_{\pm 0}$ | $\mathbf{0}_{\pm 0}$ | $\mathbf{0}_{\pm 0}$ |
| | noisy | cube-single-noisy-v0 | $8_{\pm 3}$ | $71_{\pm 9}$ | $\mathbf{99}_{\pm 1}$ | $25_{\pm 6}$ | $38_{\pm 2}$ | $41_{\pm 6}$ |
| | | cube-double-noisy-v0 | $1_{\pm 1}$ | $14_{\pm 3}$ | $\mathbf{23}_{\pm 3}$ | $3_{\pm 1}$ | $2_{\pm 1}$ | $2_{\pm 1}$ |
| | | cube-triple-noisy-v0 | $1_{\pm 1}$ | $\mathbf{9}_{\pm 1}$ | $2_{\pm 1}$ | $1_{\pm 0}$ | $3_{\pm 1}$ | $2_{\pm 1}$ |
| | | cube-quadruple-noisy-v0 | $\mathbf{0}_{\pm 0}$ | $\mathbf{0}_{\pm 0}$ | $\mathbf{0}_{\pm 0}$ | $\mathbf{0}_{\pm 0}$ | $\mathbf{0}_{\pm 0}$ | $\mathbf{0}_{\pm 0}$ |
| scene | play | scene-play-v0 | $5_{\pm 1}$ | $42_{\pm 4}$ | $\mathbf{51}_{\pm 4}$ | $5_{\pm 1}$ | $19_{\pm 2}$ | $38_{\pm 3}$ |
| | noisy | scene-noisy-v0 | $1_{\pm 1}$ | $\mathbf{26}_{\pm 5}$ | $\mathbf{26}_{\pm 2}$ | $9_{\pm 2}$ | $1_{\pm 1}$ | $\mathbf{25}_{\pm 4}$ |
| puzzle | play | puzzle-3x3-play-v0 | $2_{\pm 0}$ | $6_{\pm 1}$ | $\mathbf{95}_{\pm 1}$ | $1_{\pm 0}$ | $3_{\pm 1}$ | $12_{\pm 2}$ |
| | | puzzle-4x4-play-v0 | $0_{\pm 0}$ | $13_{\pm 2}$ | $\mathbf{26}_{\pm 3}$ | $0_{\pm 0}$ | $0_{\pm 0}$ | $7_{\pm 2}$ |
| | | puzzle-4x5-play-v0 | $0_{\pm 0}$ | $7_{\pm 1}$ | $\mathbf{14}_{\pm 1}$ | $0_{\pm 0}$ | $1_{\pm 0}$ | $4_{\pm 1}$ |
| | | puzzle-4x6-play-v0 | $0_{\pm 0}$ | $10_{\pm 2}$ | $\mathbf{12}_{\pm 1}$ | $0_{\pm 0}$ | $4_{\pm 1}$ | $3_{\pm 1}$ |
| | noisy | puzzle-3x3-noisy-v0 | $1_{\pm 0}$ | $42_{\pm 19}$ | $\mathbf{94}_{\pm 3}$ | $0_{\pm 0}$ | $30_{\pm 6}$ | $51_{\pm 11}$ |
| | | puzzle-4x4-noisy-v0 | $0_{\pm 0}$ | $20_{\pm 3}$ | $\mathbf{29}_{\pm 7}$ | $0_{\pm 0}$ | $0_{\pm 0}$ | $16_{\pm 4}$ |
| | | puzzle-4x5-noisy-v0 | $0_{\pm 0}$ | $\mathbf{19}_{\pm 0}$ | $\mathbf{19}_{\pm 0}$ | $0_{\pm 0}$ | $3_{\pm 2}$ | $5_{\pm 1}$ |
| | | puzzle-4x6-noisy-v0 | $0_{\pm 0}$ | $17_{\pm 2}$ | $\mathbf{18}_{\pm 2}$ | $0_{\pm 0}$ | $6_{\pm 3}$ | $2_{\pm 1}$ |
| visual-cube | play | visual-cube-single-play-v0 | $5_{\pm 1}$ | $60_{\pm 5}$ | $30_{\pm 5}$ | $41_{\pm 15}$ | $31_{\pm 15}$ | $\mathbf{89}_{\pm 0}$ |
| | | visual-cube-double-play-v0 | $1_{\pm 1}$ | $10_{\pm 2}$ | $1_{\pm 1}$ | $5_{\pm 0}$ | $2_{\pm 1}$ | $\mathbf{39}_{\pm 2}$ |
| | | visual-cube-triple-play-v0 | $15_{\pm 2}$ | $14_{\pm 2}$ | $15_{\pm 1}$ | $16_{\pm 1}$ | $17_{\pm 2}$ | $\mathbf{21}_{\pm 0}$ |
| | | visual-cube-quadruple-play-v0 | $8_{\pm 1}$ | $0_{\pm 0}$ | $7_{\pm 1}$ | $5_{\pm 1}$ | $4_{\pm 1}$ | $\mathbf{14}_{\pm 1}$ |
| | noisy | visual-cube-single-noisy-v0 | $14_{\pm 3}$ | $75_{\pm 3}$ | $48_{\pm 3}$ | $10_{\pm 5}$ | $39_{\pm 30}$ | $\mathbf{99}_{\pm 0}$ |
| | | visual-cube-double-noisy-v0 | $5_{\pm 1}$ | $17_{\pm 4}$ | $22_{\pm 2}$ | $6_{\pm 2}$ | $6_{\pm 3}$ | $\mathbf{59}_{\pm 3}$ |
| | | visual-cube-triple-noisy-v0 | $16_{\pm 1}$ | $18_{\pm 1}$ | $12_{\pm 1}$ | $9_{\pm 4}$ | $16_{\pm 1}$ | $\mathbf{23}_{\pm 2}$ |
| | | visual-cube-quadruple-noisy-v0 | $9_{\pm 0}$ | $0_{\pm 0}$ | $2_{\pm 2}$ | $0_{\pm 0}$ | $8_{\pm 2}$ | $\mathbf{12}_{\pm 8}$ |
| visual-scene | play | visual-scene-play-v0 | $12_{\pm 2}$ | $25_{\pm 3}$ | $12_{\pm 2}$ | $10_{\pm 1}$ | $11_{\pm 2}$ | $\mathbf{49}_{\pm 4}$ |
| | noisy | visual-scene-noisy-v0 | $13_{\pm 2}$ | $23_{\pm 2}$ | $12_{\pm 4}$ | $2_{\pm 0}$ | $15_{\pm 2}$ | $\mathbf{50}_{\pm 1}$ |
| visual-puzzle | play | visual-puzzle-3x3-play-v0 | $0_{\pm 0}$ | $21_{\pm 1}$ | $1_{\pm 2}$ | $1_{\pm 1}$ | $0_{\pm 0}$ | $\mathbf{73}_{\pm 8}$ |
| | | visual-puzzle-4x4-play-v0 | $10_{\pm 1}$ | $\mathbf{60}_{\pm 5}$ | $16_{\pm 4}$ | $0_{\pm 0}$ | $10_{\pm 6}$ | $\mathbf{60}_{\pm 41}$ |
| | | visual-puzzle-4x5-play-v0 | $5_{\pm 2}$ | $\mathbf{17}_{\pm 1}$ | $7_{\pm 2}$ | $0_{\pm 0}$ | $6_{\pm 1}$ | $13_{\pm 9}$ |
| | | visual-puzzle-4x6-play-v0 | $2_{\pm 1}$ | $\mathbf{15}_{\pm 1}$ | $2_{\pm 1}$ | $0_{\pm 0}$ | $3_{\pm 1}$ | $9_{\pm 6}$ |
| | noisy | visual-puzzle-3x3-noisy-v0 | $1_{\pm 1}$ | $20_{\pm 0}$ | $26_{\pm 4}$ | $0_{\pm 0}$ | $1_{\pm 1}$ | $\mathbf{70}_{\pm 6}$ |
| | | visual-puzzle-4x4-noisy-v0 | $7_{\pm 3}$ | $47_{\pm 3}$ | $49_{\pm 7}$ | $0_{\pm 0}$ | $6_{\pm 2}$ | $\mathbf{84}_{\pm 4}$ |
| | | visual-puzzle-4x5-noisy-v0 | $6_{\pm 1}$ | $14_{\pm 10}$ | $\mathbf{19}_{\pm 0}$ | $0_{\pm 0}$ | $7_{\pm 1}$ | $14_{\pm 10}$ |
| | | visual-puzzle-4x6-noisy-v0 | $2_{\pm 1}$ | $12_{\pm 8}$ | $\mathbf{17}_{\pm 1}$ | $0_{\pm 0}$ | $2_{\pm 1}$ | $14_{\pm 2}$ |

2024a;b). In this work, we also consider the problem of offline goal-conditioned RL; however, instead of proposing a new algorithm, we introduce a new benchmark and reference implementations to facilitate and advance algorithms research in offline GCRL.

## F  OFFLINE GCRL ALGORITHMS

In this section, we describe the six offline GCRL methods used for benchmarking in detail. We first define four goal-sampling distributions that correspond to the current state, uniform future states, geometric future states, and random states, respectively:

- $p^{\mathcal{D}}_{\mathrm{cur}}(g \mid s)$ denotes the Dirac delta distribution at $s$ (*i.e.*, $g$ is always set to $s$).
- $p^{\mathcal{D}}_{\mathrm{traj}}(g \mid s)$ denotes the uniform future state distribution defined as follows: assuming $s = s_t$ in a trajectory $\tau = (s_0, a_0, s_1, \ldots, s_T)$[8], we sample an index $k$ from the uniform distribution $\mathrm{Unif}(\min(t+1, T-1), T-1)$ (inclusive), and set $g = s_k$.
- $p^{\mathcal{D}}_{\mathrm{geom}}(g \mid s)$ denotes the truncated geometric future state distribution defined as follows: assuming $s = s_t$ in a trajectory $\tau = (s_0, a_0, s_1, \ldots, s_T)$, we sample an index $k$ from the geometric distribution $\mathrm{Geom}(1 - \gamma)$ (whose support starts from 1), and set $g = s_{\min(s+k, T-1)}$.
- $p^{\mathcal{D}}_{\mathrm{rand}}(g)$ denotes the uniform state distribution over the dataset $\mathcal{D}$.

Additionally, $p^{\mathcal{D}}_{\mathrm{mixed}}(g \mid s)$ denotes a mixture of these four goal-sampling distributions with a mixture ratio defined by hyperparameters, $p^{\mathcal{D}}(\cdot)$ simply denotes the uniform distribution over the dataset, and

---

[8]If there are multiple such $(\tau, t)$ tuples in the dataset, consider the uniform mixture of them.

we sometimes use $p^{\mathcal{D}}(\cdot \mid s, a)$ instead of $p^{\mathcal{D}}(\cdot \mid s)$ to denote the distribution corresponding to the state-action pair.

**Goal-conditioned behavioral cloning (GCBC).** GCBC (Lynch et al., 2019; Ghosh et al., 2021) simply performs behavioral cloning using future states in the same trajectory as goals. GCBC maximizes the following objective to train a goal-conditioned policy $\pi(a \mid s, g)$.

$$J_{\text{GCBC}}(\pi) = \mathbb{E}_{(s,a)\sim p^{\mathcal{D}}(s,a), g\sim p^{\mathcal{D}}_{\text{traj}}(g|s)}[\log \pi(a \mid s, g)]. \tag{1}$$

**Goal-conditioned implicit {V, Q}-learning (GCIVL and GCIQL).** GCIVL and GCIQL are goal-conditioned variants of implicit Q-learning (IQL) (Kostrikov et al., 2022), which is an offline RL algorithm that fits the optimal value functions ($V^*$ or $Q^*$) using an expectile regression (Newey & Powell, 1987). GCIQL is a straightforward goal-conditioned variant of IQL, and GCIVL is the $V$-only variant introduced by Park et al. (2023). GCIVL fits a value function $V(s, g)$ by minimizing the following loss:

$$\mathcal{L}_{\text{GCIVL}}(V) = \mathbb{E}_{s\sim p^{\mathcal{D}}(s), g\sim p^{\mathcal{D}}_{\text{mixed}}(g|s)} \left[ \ell^2_\kappa \left( r(s, g) + \gamma \bar{V}(s', g) - V(s, g) \right) \right], \tag{2}$$

where $r(s, g) = \mathbf{1}_{\{g\}}(s) - 1$ denotes the $(-1, 0)$-sparse goal-conditioned reward function, $\bar{V}$ denotes the target value function (Mnih et al., 2013), and $\ell^2_\kappa(x) = |\kappa - \mathbf{1}_{\{x : x<0\}}(x)|x^2$ denotes the expectile loss with an expectile $\kappa$.

GCIQL fits both $V(s, g)$ and $Q(s, a, g)$ by jointly minimizing the following losses:

$$\mathcal{L}^V_{\text{GCIQL}}(V) = \mathbb{E}_{(s,a)\sim p^{\mathcal{D}}(s,a), g\sim p^{\mathcal{D}}_{\text{mixed}}(g|s)} \left[ \ell^2_\kappa \left( \bar{Q}(s, a, g) - V(s, g) \right) \right], \tag{3}$$

$$\mathcal{L}^Q_{\text{GCIQL}}(Q) = \mathbb{E}_{(s,a,s')\sim p^{\mathcal{D}}(s,a,s'), g\sim p^{\mathcal{D}}_{\text{mixed}}(g|s)} \left[ \left( r(s, g) + \gamma V(s', g) - Q(s, a, g) \right)^2 \right], \tag{4}$$

where $\bar{Q}$ denotes the target Q function (Mnih et al., 2013). We note that GCIVL is optimistically biased in stochastic environments, but GCIQL is unbiased (Kostrikov et al., 2022; Park et al., 2023).

To extract a policy from the learned value functions, we can use either advantage-weighted regression (**AWR**) (Peters & Schaal, 2007; Peng et al., 2019) or behavior-constrained deep deterministic policy gradient (**DDPG+BC**) (Fujimoto & Gu, 2021). GCIVL uses the following value-only variant of the AWR objective (Park et al., 2023):

$$J^V_{\text{AWR}}(\pi) = \mathbb{E}_{(s,a,s')\sim p^{\mathcal{D}}(s,a,s'), g\sim p^{\mathcal{D}}_{\text{mixed}}(g|s)} \left[ e^{\alpha(V(s',g)-V(s,g))} \log \pi(a \mid s, g) \right], \tag{5}$$

where $\alpha$ is the temperature hyperparameter. In our experiments, GCIQL mainly uses the following DDPG+BC objective (which is known to be better than AWR (Park et al., 2024a)):

$$J_{\text{DDPG+BC}}(\pi) = \mathbb{E}_{(s,a)\sim p^{\mathcal{D}}(s,a), g\sim p^{\mathcal{D}}_{\text{mixed}}(g|s)} [Q(s, \pi^\mu(s, g), g) + \alpha \log \pi(a \mid s, g)], \tag{6}$$

where $\pi^\mu(s, g) = \mathbb{E}_{a\sim\pi(a|s,g)}[a]$. In discrete-action environments, GCIQL uses the following Q version of AWR:

$$J^Q_{\text{AWR}}(\pi) = \mathbb{E}_{(s,a,s')\sim p^{\mathcal{D}}(s,a,s'), g\sim p^{\mathcal{D}}_{\text{mixed}}(g|s)} \left[ e^{\alpha(Q(s,a,g)-V(s,g))} \log \pi(a \mid s, g) \right]. \tag{7}$$

In practice, we use standard double-value learning for GCIVL and GCIQL (Park et al., 2023), and Q normalization for DDPG+BC (Fujimoto & Gu, 2021).

**Quasimetric RL (QRL).** QRL (Wang et al., 2023) is a goal-conditioned value learning algorithm based on quasimetric learning, where a quasimetric means an asymmetric metric. In deterministic environments, the shortest path length between two states $d^*(s, g)$ is equivalent to the optimal undiscounted goal-conditioned value function $V^*(s, g)$ under the $(-1, 0)$-sparse reward function (Wang et al., 2023): $V^*(s, g) = -d^*(s, g)$. The main idea of QRL is to explicitly leverage the quasimetric property (*i.e.*, triangle inequality) of shortest path lengths, namely $d^*(s, w) + d^*(w, g) \geq d^*(s, g)$ for any $s, w, g \in \mathcal{S}$, by modeling it with a quasimetric network architecture like MRN (Liu et al., 2023) or IQE (Wang & Isola, 2022). Concretely, QRL maximizes the following constrained optimization objective:

$$\text{maximize} \quad \mathbb{E}_{s\sim p^{\mathcal{D}}(s), g\sim p^{\mathcal{D}}_{\text{rand}}(g)}[d(s, g)] \tag{8}$$

$$\text{s.t.} \quad \mathbb{E}_{(s,s')\sim p^{\mathcal{D}}(s,s')} \left[ (d(s, s') - 1)^2 \right] \leq \varepsilon^2, \tag{9}$$

where $d(s, g)$ is a quasimetric distance function (*e.g.*, an IQE network) and $\varepsilon$ is a hyperparameter that controls the strength of the constraint.

To extract a policy from the value function $V(s, g) = -d(s, g)$, QRL uses the value-only AWR loss Equation (5) in discrete-action MDPs. In continuous-action MDPs, QRL additionally fits a dynamics model $f(\phi(s), a) \colon \mathcal{Z} \times \mathcal{A} \to \mathcal{Z}$ (Wang et al., 2023), where $\mathcal{Z}$ denotes a latent space and $\phi(s) \colon \mathcal{S} \to \mathcal{Z}$ denotes the representation function used in the quasimetric distance function: namely, $d(s, g) = \tilde{d}(\phi(s), \phi(g))$ (*e.g.*, $\phi$ is the interval representation function in IQE). The dynamics loss is as follows:

$$\mathcal{L}_{\mathrm{dyn}}(f) = \mathbb{E}_{(s,a,s') \sim p^{\mathcal{D}}(s,a,s')} \left[ \tilde{d}\left(\phi(s'), f(\phi(s), a)\right) + \tilde{d}\left(f(\phi(s), a), \phi(s')\right) \right], \qquad (10)$$

where QRL jointly trains both $d$ and $f$ without stop-gradients. Based on the dynamics model $f$, QRL maximizes the following DDPG+BC-like loss:

$$J_{\mathrm{DDPG+BC}}(\pi) = \mathbb{E}_{(s,a) \sim p^{\mathcal{D}}(s,a), g \sim p^{\mathcal{D}}_{\mathrm{mixed}}(g|s)} \left[ -\tilde{d}\left(f(\phi(s), \pi^{\mu}(s, g)), \phi(g)\right) + \alpha \log \pi(a \mid s, g) \right], \tag{11}$$

where we use the same notation as Equation (6).

In practice, we employ a $\mathrm{softplus}$ loss shaping for the quasimetric loss and delta prediction for the dynamics model, as in Wang et al. (2023).

**Contrastive RL (CRL).** CRL (Eysenbach et al., 2022) is a "one-step" GCRL algorithm that first trains a Monte Carlo goal-conditioned value function using contrastive learning and performs a one-step policy improvement. CRL maximizes the following binary NCE objective (Ma & Collins, 2018) with respect to $f(s, a, g) \colon \mathcal{S} \times \mathcal{A} \times \mathcal{S} \to \mathbb{R}$:

$$J_{\mathrm{CRL}}(f) = \mathbb{E}_{(s,a) \sim p^{\mathcal{D}}(s,a), g \sim p^{\mathcal{D}}_{\mathrm{geom}}(g|s,a), g^- \sim p^{\mathcal{D}}_{\mathrm{rand}}(g)} [\log \sigma(f(s, a, g)) + \log(1 - \sigma(f(s, a, g^-)))], \tag{12}$$

where $\sigma \colon \mathbb{R} \to (0, 1)$ denotes the sigmoid function. The optimal solution to the above objective is given as $f(s, a, g) = \log(p^{\mathcal{D}}_{\mathrm{geom}}(g \mid s, a)/p^{\mathcal{D}}_{\mathrm{rand}}(g))$. Given the equivalence between the geometric future goal distribution ($p^{\mathcal{D}}_{\mathrm{geom}}$) and the Monte Carlo goal-conditioned Q function ($Q^{\mathrm{MC}}$) under the $(0, 1)$-sparse reward function $r(s, g) = \mathbf{1}_{\{g\}}(s)$ (Eysenbach et al., 2022), we get the following relation: $f(s, a, g) = \log Q^{\mathrm{MC}}(s, a, g) + C(g)$, where $C$ is a function that only depends on $g$. In practice, $f$ is parameterized as $f(s, a, g) = \phi(s, a)^{\top} \psi(g)/\sqrt{d}$ with $\phi \colon \mathcal{S} \times \mathcal{A} \to \mathcal{Z} = \mathbb{R}^d$ and $\psi \colon \mathcal{S} \to \mathcal{Z} = \mathbb{R}^d$ (note that this inner-product parameterization is universal (Park et al., 2024c)), and we use the future goals from the other states in the same batch as $g^-$. We also employ the double-value learning technique for $f$ (Eysenbach et al., 2022).

For policy extraction, in continuous-action MDPs, we employ DDPG+BC (Equation (6)) using $f$ instead of $Q$. In discrete-action MDPs, we use AWR (Equation (5)) using $(f, f^V)$ instead of $(Q, V)$, where we additionally train a contrastive value function $f^V(s, g) \colon \mathcal{S} \times \mathcal{S} \to \mathbb{R}$ using

$$J_{\mathrm{CRL-V}}(f^V) = \mathbb{E}_{s \sim p^{\mathcal{D}}(s), g \sim p^{\mathcal{D}}_{\mathrm{geom}}(g|s), g^- \sim p^{\mathcal{D}}_{\mathrm{rand}}(g)} [\log \sigma(f^V(s, g)) + \log(1 - \sigma(f^V(s, g^-)))], \tag{13}$$

with a similar inner product parameterization for $f^V$.

**Hierarchical implicit Q-learning (HIQL).** HIQL (Park et al., 2023) is an offline GCRL algorithm that extracts two policies from a single goal-conditioned value function. HIQL first trains GCIVL (Equation (2)) with a parameterized value function defined as $V(s, g) = \tilde{V}(s, \phi(s, g))$, where $\phi \colon \mathcal{S} \times \mathcal{S} \to \mathcal{Z}$ serves as a (state-dependent) subgoal representation function. Based on the GCIVL value function, HIQL extracts a high-level policy $\pi^h \colon \mathcal{S} \times \mathcal{S} \to \Delta(\mathcal{Z})$ and a low-level policy $\pi^{\ell} \colon \mathcal{S} \times \mathcal{Z} \to \Delta(\mathcal{A})$ with the following AWR-like objectives:

$$J^h_{\mathrm{HIQL}}(\pi^h) = \mathbb{E}_{(s_t, s_{t+k}) \sim p^{\mathcal{D}}, g \sim p^{\mathcal{D}}_{\mathrm{mixed}}(g|s_t)} \left[ e^{\alpha(V(s_{t+k}, g) - V(s_t, g))} \log \pi^h(\phi(s_t, s_{t+k}) \mid s_t, g) \right], \tag{14}$$

$$J^{\ell}_{\mathrm{HIQL}}(\pi^{\ell}) = \mathbb{E}_{(s_t, a_t, s_{t+1}, s_{t+k}) \sim p^{\mathcal{D}}} \left[ e^{\alpha(V(s_{t+1}, s_{t+k}) - V(s_t, s_{t+k}))} \log \pi^{\ell}(a_t \mid s_t, \phi(s_t, s_{t+k})) \right], \tag{15}$$

where we omit the arguments in $p^{\mathcal{D}}$, and $k$ denotes a hyperparameter corresponding to the subgoal step. For simplicity, we ignore some edge cases in the objectives above (*e.g.*, when $t + k$ exceeds the trajectory boundary, in which case we truncate); we refer to Park et al. (2023) or our code for the full details. Intuitively, the high-level policy predicts the representation of the optimal $k$-step subgoal, and the low-level policy predicts the optimal action based on the predicted subgoal.

In practice, following Park et al. (2023), we use the double-value learning technique, normalize the output of $\phi$, and allow gradient flows from the low-level AWR loss into $\phi$ (only) in pixel-based environments.

## G    IMPLEMENTATION DETAILS

We provide the full implementation details in this section. We release the code as well as the exact command-line flags to reproduce the entire benchmark table, datasets, and expert policies at https://github.com/seohongpark/ogbench.

### G.1    TASKS AND DATASETS

In this section, we provide further information about our tasks and datasets. We provide the basic specifications about the environments and datasets in Tables 7 and 8.

**Locomotion tasks.** For OGBench AntMaze and AntSoccer, we adopt the Ant model from D4RL AntMaze (Fu et al., 2020) (which is based on the Ant in OpenAI Gym (Brockman et al., 2016; Towers et al., 2024), but with a more restricted joint range) and the soccer ball model from the DeepMind Control suite (Tassa et al., 2018). For HumanoidMaze, we adopt the Humanoid model from the DeepMind Control suite (Tassa et al., 2018).

To collect datasets, we train expert low-level directional (AntMaze and HumanoidMaze) or goal-reaching (AntSoccer) policies using SAC with dense reward functions for 400K (AntMaze), 40M (HumanoidMaze), or 12M (AntSoccer) steps. For PointMaze, we use a scripted directional expert policy. When collecting datasets, we add Gaussian noise with a standard deviation of $0.5$ (`pointmaze`), $1.0$ (`explore`), or $0.2$ (others) to expert actions.

The success criteria for the locomotion tasks are based only on the distance between the agent (or the ball in AntSoccer) and the goal location. In particular, the tasks do not consider joint positions determining success, as in previous works (Fu et al., 2020; Park et al., 2023).

**Manipulation tasks.** We adopt the UR5e robot arm and Robotiq 2F-85 gripper models from MuJoCo Menagerie (Zakka et al., 2022), and the drawer, window, and button box models from Meta-World (Yu et al., 2019). The robot is end-effector controlled with a 5-D action space, where the dimensions correspond to the displacements in the $x$ position, $y$ position, $z$ position, gripper yaw, and gripper opening. In all manipulation tasks, we place invisible walls to prevent objects from moving into an area beyond the robot arm's reach. These invisible walls also prevent the cube objects from moving outside the camera viewpoint in pixel-based manipulation environments. However, since some blind spots still exist in `visual-scene` even with the walls, we further filter out such rare cases in trajectories prevent ambiguous camera observations completely.

In Puzzle, not every button configuration is reachable from the initial state. While the $3 \times 3$, $4 \times 5$, and $4 \times 6$ puzzles do have this property, the $4 \times 4$ puzzle does not. This can be seen by computing the rank of the $nm \times nm$ button effect matrix over $\mathbb{F}_2$ (the field with two elements), where $n$ and $m$ denote the numbers of rows and columns, respectively. In our tasks, we ensure that every test-time goal is solvable. Also, we note that the maximum value of the minimum number of button presses to reach a state from another state in each puzzle is 9 ($3 \times 3$ puzzle), 7 ($4 \times 4$ puzzle), 20 ($4 \times 5$ puzzle), or 24 ($4 \times 6$ puzzle). Each puzzle environment contains at least one evaluation goal that requires the maximum number of presses.

The `play` datasets are collected by open-loop, non-Markovian scripted policies, and the `noisy` datasets are collected by closed-loop, Markovian scripted policies. For the `play` datasets, we add temporally correlated action noise to the expert actions to enhance state coverage. For the `noisy` datasets, we first sample the degree of action noise at the beginning of each episode, and collect a

trajectory with the chosen amount of (time-independent) Gaussian action noise. This ensures high coverage while having a sufficient number of optimal trajectories.

The success criteria for the manipulation tasks are based only on the object configurations; the arm pose is not considered when determining success. For cubes, only the distances between the goal positions and their current positions are considered, and their orientations are ignored.

**Drawing tasks.** We modify the original Powderworld environment (Frans & Isola, 2023) to make it offline and goal-conditioned. We also re-implement Powderworld (which was originally implemented in PyTorch) in NumPy to remove the dependency on PyTorch. We provide three versions of Powderworld tasks: `powderworld-easy` uses two elements (plant and stone), `powderworld-medium` uses five elements (sand, water, fire, plant, and stone), and `powderworld-hard` uses eight elements (sand, water, fire, plant, stone, gas, wood, and ice). An action in Powderworld corresponds to drawing a $4 \times 4$-sized square with a specific element brush on the $32 \times 32$-sized board. Since naïvely implementing this atomic action requires up to 512-dimensional discrete actions, we split it into three *sequential* 8-dimensional actions that correspond to element selection, $x$-coordinate selection, and $y$-coordinate selection. To ensure full observability, we add three additional dimensions that contain information about the currently selected element and $x$ coordinate to the original $32 \times 32 \times 3$-dimensional image, which results in a $32 \times 32 \times 6$-dimensional observation space. When the agent selects an invalid action (which can only happen in `powderworld-{easy, medium}`, which has fewer than 8 elements), the environment instead uses a randomly sampled valid action.

The datasets are collected by a scripted policy that randomly draws squares and lines or fills the entire board with randomly selected brushes. With a probability of $0.5$, it performs a random action (*i.e.*, places a random element on a randomly sampled position).

For the success criterion for evaluation goals, we use the following procedure to allow for some tolerance: For each pixel in the goal image, we check if the current image has a matching pixel that is shifted by up to one pixel in any direction. We then compute the error as the number of pixels that do not match, and consider the task successful if the error is below a certain threshold.

## G.2 SINGLE-TASK VARIANTS

OGBench also supports standard (*i.e.*, non-goal-conditioned) offline RL by providing single-task variants of locomotion and manipulation tasks. To convert a goal-conditioned task into a standard reward-maximizing task, we fix an evaluation goal and relabel the dataset with a semi-sparse reward function. This semi-sparse reward function is defined as the negative of the number of unaccomplished subtasks in the current state, and the episode immediately terminates when the agent completes all subtasks of the target evaluation goal. In locomotion environments, rewards are always $-1$ or $0$, as there are no separate subtasks. In manipulation environments, rewards range between $-n_{\text{task}}$ and $0$, where $n_{\text{task}}$ denotes the number of subtasks (*e.g.*, in `puzzle-4x6`, $n_{\text{task}} = 24$ as there are 24 buttons).

Each locomotion and manipulation task in OGBench provides five single-task variants that correspond to the five evaluation goals (Appendix H), resulting in a total of $410$ single-task tasks. They are named with the suffix "`singletask-task[n]`" (*e.g.*, `scene-play-singletask-task2-v0`), where `[n]` denotes a number between $1$ and $5$ (inclusive). Among the five tasks in each environment, the most representative one is chosen as the "default" task, and is aliased by the suffix "`singletask`" without a task number. For example, in `cube-double`, the second task (standard double pick-and-place; see Figure 6) is set as the default task, and `cube-double-play-singletask-v0` and `cube-double-play-singletask-task2-v0` refer to the same task. Default tasks can be useful in various ways. For instance, one may report performance only on default tasks to reduce the computational burden, or may treat default tasks as a "training" task set for tuning hyperparameters while using the other four tasks as a "validation" task set. We provide the list of default tasks in Table 9.

While we do not provide a separate benchmarking result on the single-task environments, a benchmarking table of several representative offline RL algorithms on 50 tasks can be found in the work by Park et al. (2025).

### G.3 ORACLE REPRESENTATION VARIANTS

OGBench also provides oracle representation variants of locomotion and manipulation tasks, denoted by the suffix "oraclerep" (*e.g.*, antmaze-large-navigate-oraclerep-v0). These tasks provide low-dimensional oracle goal representations that contain only relevant information for fulfilling the goal success criterion. This corresponds to the $x$-$y$ coordinates of the agent (or the ball in antsoccer) in locomotion environments, and the positions of the cubes and the states of the objects in manipulation environments. The oraclerep tasks reduce the burden of goal representation learning, potentially helping diagnose the bottlenecks in goal-conditioned RL algorithms. We do not provide a separate benchmarking result for the oracle representation variants.

### G.4 METHODS

Our implementations of six offline GCRL algorithms (GCBC, GCIVL, GCIQL, QRL, CRL, and HIQL) are based on JAX (Bradbury et al., 2018). In our benchmark, each run typically takes 2-5 hours (state-based tasks) or 5-12 hours (pixel-based tasks) on an A5000 GPU, depending on the task and algorithm.

For benchmarking, we periodically evaluate the performance (goal success rate in percentage) of each agent on each test-time goal with 50 rollouts every 100K steps, and report the average success rate across the last three evaluation epochs (*i.e.*, at 800K, 900K, and 1M steps for state-based tasks and at 300K, 400K, and 500K steps for pixel-based tasks). That is, the performance of each agent is averaged over 750 rollouts (3 evaluation epochs $\times$ 5 test-time goals $\times$ 50 rollouts). While we use a relatively large number of evaluation rollouts for robustness, researchers can adjust the number of evaluation rollouts (*e.g.*, 20 episodes for each test-time goal) to reduce the computational burden.

We provide the full list of common hyperparameters in Table 10. We find that methods are more sensitive to policy extraction hyperparameters (*e.g.*, the BC coefficient in DDPG+BC) (Park et al., 2024a), and report these in a separate table (Table 11). Specifically, for each method, we use the same value learning hyperparameters across the benchmark except for the discount factor $\gamma$ (Table 10), but individually tune the policy extraction hyperparameters (*e.g.*, AWR $\alpha$ and DDPG+BC $\alpha$) for each dataset category (Tables 10 and 11).

We apply layer normalization (Ba et al., 2016) to the value networks, but not to the policy networks. In pixel-based environments, we use a smaller version of the IMPALA encoder (Espeholt et al., 2018). We use random-crop image augmentation (with a probability of 0.5) for pixel-based manipulation tasks, but not for pixel-based locomotion or drawing tasks, as we find it to be helpful mainly on manipulation tasks. In pixel-based environments, we do not apply frame stacking for simplicity, as we find it does not necessarily improve performance on most tasks including Visual AntMaze (although we believe the performance on Visual HumanoidMaze can further be improved with frame stacking). For policies, we parameterize the action distribution with a unit-variance Gaussian distribution. We find that using a fixed standard deviation is especially important for DDPG+BC. During evaluation, we use the deterministic mean of the learned Gaussian policy. However, in Powderworld, which has a discrete action space, we use a stochastic policy with a temperature of 0.3 (*i.e.*, we divide the action logits by 0.3), as this additional stochasticity helps prevent the agent from getting stuck in certain states.

Table 6: **Dataset categories.** We list the dataset categories used to aggregate results in Figure 2. Note that some datasets or tasks (*e.g.*, PointMaze) do not belong to any of these aggregation categories (for being too simple, too special, etc.).

| Category | Datasets |
|---|---|
| Locomotion (states) | antmaze-medium-navigate-v0
antmaze-large-navigate-v0
antmaze-giant-navigate-v0
humanoidmaze-medium-navigate-v0
humanoidmaze-large-navigate-v0
humanoidmaze-giant-navigate-v0
antsoccer-arena-navigate-v0
antsoccer-medium-navigate-v0 |
| Locomotion (pixels) | visual-antmaze-medium-navigate-v0
visual-antmaze-large-navigate-v0
visual-antmaze-giant-navigate-v0
visual-humanoidmaze-medium-navigate-v0
visual-humanoidmaze-large-navigate-v0
visual-humanoidmaze-giant-navigate-v0 |
| Manipulation (states) | cube-single-play-v0
cube-double-play-v0
cube-triple-play-v0
cube-quadruple-play-v0
scene-play-v0
puzzle-3x3-play-v0
puzzle-4x4-play-v0
puzzle-4x5-play-v0
puzzle-4x6-play-v0 |
| Manipulation (pixels) | visual-cube-single-play-v0
visual-cube-double-play-v0
visual-cube-triple-play-v0
visual-cube-quadruple-play-v0
visual-scene-play-v0
visual-puzzle-3x3-play-v0
visual-puzzle-4x4-play-v0
visual-puzzle-4x5-play-v0
visual-puzzle-4x6-play-v0 |
| Drawing (pixels) | powderworld-easy-play-v0
powderworld-medium-play-v0
powderworld-hard-play-v0 |
| Stitching (states) | antmaze-medium-stitch-v0
antmaze-large-stitch-v0
antmaze-giant-stitch-v0
humanoidmaze-medium-stitch-v0
humanoidmaze-large-stitch-v0
humanoidmaze-giant-stitch-v0
antsoccer-arena-stitch-v0
antsoccer-medium-stitch-v0 |
| Stitching (pixels) | visual-antmaze-medium-stitch-v0
visual-antmaze-large-stitch-v0
visual-antmaze-giant-stitch-v0
visual-humanoidmaze-medium-stitch-v0
visual-humanoidmaze-large-stitch-v0
visual-humanoidmaze-giant-stitch-v0 |
| Exploratory (states) | antmaze-medium-explore-v0
antmaze-large-explore-v0 |
| Exploratory (pixels) | visual-antmaze-medium-explore-v0
visual-antmaze-large-explore-v0 |
| Stochastic (states) | antmaze-teleport-navigate-v0 |
| Stochastic (pixels) | visual-antmaze-telpeport-navigate-v0 |

Table 7: **Environment specifications.** See Table 8 for the dataset specifications. Note that the episode lengths of datasets and environments can be different.

| Environment Type | Environment | State Dim. | Action Dim. | Maximum Episode Length |
|---|---|---|---|---|
| pointmaze | pointmaze-medium-v0 | 2 | 2 | 1000 |
| | pointmaze-large-v0 | 2 | 2 | 1000 |
| | pointmaze-giant-v0 | 2 | 2 | 1000 |
| | pointmaze-teleport-v0 | 2 | 2 | 1000 |
| antmaze | antmaze-medium-v0 | 29 | 8 | 1000 |
| | antmaze-large-v0 | 29 | 8 | 1000 |
| | antmaze-giant-v0 | 29 | 8 | 1000 |
| | antmaze-teleport-v0 | 29 | 8 | 1000 |
| humanoidmaze | humanoidmaze-medium-v0 | 69 | 21 | 2000 |
| | humanoidmaze-large-v0 | 69 | 21 | 2000 |
| | humanoidmaze-giant-v0 | 69 | 21 | 4000 |
| antsoccer | antsoccer-arena-v0 | 42 | 8 | 1000 |
| | antsoccer-medium-v0 | 42 | 8 | 1000 |
| visual-antmaze | visual-antmaze-medium-v0 | $64 \times 64 \times 3$ | 8 | 1000 |
| | visual-antmaze-large-v0 | $64 \times 64 \times 3$ | 8 | 1000 |
| | visual-antmaze-giant-v0 | $64 \times 64 \times 3$ | 8 | 1000 |
| | visual-antmaze-teleport-v0 | $64 \times 64 \times 3$ | 8 | 1000 |
| visual-humanoidmaze | visual-humanoidmaze-medium-v0 | $64 \times 64 \times 3$ | 21 | 2000 |
| | visual-humanoidmaze-large-v0 | $64 \times 64 \times 3$ | 21 | 2000 |
| | visual-humanoidmaze-giant-v0 | $64 \times 64 \times 3$ | 21 | 4000 |
| cube | cube-single-v0 | 28 | 5 | 200 |
| | cube-double-v0 | 37 | 5 | 500 |
| | cube-triple-v0 | 46 | 5 | 1000 |
| | cube-quadruple-v0 | 55 | 5 | 1000 |
| scene | scene-v0 | 40 | 5 | 750 |
| puzzle | puzzle-3x3-v0 | 55 | 5 | 500 |
| | puzzle-4x4-v0 | 83 | 5 | 500 |
| | puzzle-4x5-v0 | 99 | 5 | 1000 |
| | puzzle-4x6-v0 | 115 | 5 | 1000 |
| visual-cube | visual-cube-single-v0 | $64 \times 64 \times 3$ | 5 | 200 |
| | visual-cube-double-v0 | $64 \times 64 \times 3$ | 5 | 500 |
| | visual-cube-triple-v0 | $64 \times 64 \times 3$ | 5 | 1000 |
| | visual-cube-quadruple-v0 | $64 \times 64 \times 3$ | 5 | 1000 |
| visual-scene | visual-scene-v0 | $64 \times 64 \times 3$ | 5 | 750 |
| visual-puzzle | visual-puzzle-3x3-v0 | $64 \times 64 \times 3$ | 5 | 500 |
| | visual-puzzle-4x4-v0 | $64 \times 64 \times 3$ | 5 | 500 |
| | visual-puzzle-4x5-v0 | $64 \times 64 \times 3$ | 5 | 1000 |
| | visual-puzzle-4x6-v0 | $64 \times 64 \times 3$ | 5 | 1000 |
| powderworld | powderworld-easy-v0 | $32 \times 32 \times 6$ | 8 (discrete) | 500 |
| | powderworld-medium-v0 | $32 \times 32 \times 6$ | 8 (discrete) | 500 |
| | powderworld-hard-v0 | $32 \times 32 \times 6$ | 8 (discrete) | 500 |

Table 8: **Dataset specifications.** See Table 7 for the environment specifications. Note that the episode lengths of datasets and environments can be different.

| Environment Type | Dataset Type | Dataset | # Transitions | # Episodes | Data Episode Length |
|---|---|---|---|---|---|
| pointmaze | navigate | pointmaze-medium-navigate-v0
pointmaze-large-navigate-v0
pointmaze-giant-navigate-v0
pointmaze-teleport-navigate-v0 | 1M
1M
1M
1M | 1000
1000
500
1000 | 1000
1000
2000
1000 |
| | stitch | pointmaze-medium-stitch-v0
pointmaze-large-stitch-v0
pointmaze-giant-stitch-v0
pointmaze-teleport-stitch-v0 | 1M
1M
1M
1M | 5000
5000
5000
5000 | 200
200
200
200 |
| antmaze | navigate | antmaze-medium-navigate-v0
antmaze-large-navigate-v0
antmaze-giant-navigate-v0
antmaze-teleport-navigate-v0 | 1M
1M
1M
1M | 1000
1000
500
1000 | 1000
1000
2000
1000 |
| | stitch | antmaze-medium-stitch-v0
antmaze-large-stitch-v0
antmaze-giant-stitch-v0
antmaze-teleport-stitch-v0 | 1M
1M
1M
1M | 5000
5000
5000
5000 | 200
200
200
200 |
| | explore | antmaze-medium-explore-v0
antmaze-large-explore-v0
antmaze-teleport-explore-v0 | 5M
5M
5M | 10000
10000
10000 | 500
500
500 |
| humanoidmaze | navigate | humanoidmaze-medium-navigate-v0
humanoidmaze-large-navigate-v0
humanoidmaze-giant-navigate-v0 | 2M
2M
4M | 1000
1000
1000 | 2000
2000
4000 |
| | stitch | humanoidmaze-medium-stitch-v0
humanoidmaze-large-stitch-v0
humanoidmaze-giant-stitch-v0 | 2M
2M
4M | 5000
5000
10000 | 400
400
400 |
| antsoccer | navigate | antsoccer-arena-navigate-v0
antsoccer-medium-navigate-v0 | 1M
4M | 1000
4000 | 1000
1000 |
| | stitch | antsoccer-arena-stitch-v0
antsoccer-medium-stitch-v0 | 1M
4M | 5000
8000 | 200
500 |
| visual-antmaze | navigate | visual-antmaze-medium-navigate-v0
visual-antmaze-large-navigate-v0
visual-antmaze-giant-navigate-v0
visual-antmaze-teleport-navigate-v0 | 1M
1M
1M
1M | 1000
1000
500
1000 | 1000
1000
2000
1000 |
| | stitch | visual-antmaze-medium-stitch-v0
visual-antmaze-large-stitch-v0
visual-antmaze-giant-stitch-v0
visual-antmaze-teleport-stitch-v0 | 1M
1M
1M
1M | 5000
5000
5000
5000 | 200
200
200
200 |
| | explore | visual-antmaze-medium-explore-v0
visual-antmaze-large-explore-v0
visual-antmaze-teleport-explore-v0 | 5M
5M
5M | 10000
10000
10000 | 500
500
500 |
| visual-humanoidmaze | navigate | visual-humanoidmaze-medium-navigate-v0
visual-humanoidmaze-large-navigate-v0
visual-humanoidmaze-giant-navigate-v0 | 2M
2M
4M | 1000
1000
1000 | 2000
2000
4000 |
| | stitch | visual-humanoidmaze-medium-stitch-v0
visual-humanoidmaze-large-stitch-v0
visual-humanoidmaze-giant-stitch-v0 | 2M
2M
4M | 5000
5000
10000 | 400
400
400 |
| cube | play | cube-single-play-v0
cube-double-play-v0
cube-triple-play-v0
cube-quadruple-play-v0 | 1M
1M
3M
5M | 1000
1000
3000
5000 | 1000
1000
1000
1000 |
| | noisy | cube-single-noisy-v0
cube-double-noisy-v0
cube-triple-noisy-v0
cube-quadruple-noisy-v0 | 1M
1M
3M
5M | 1000
1000
3000
5000 | 1000
1000
1000
1000 |
| scene | play | scene-play-v0 | 1M | 1000 | 1000 |
| | noisy | scene-noisy-v0 | 1M | 1000 | 1000 |
| puzzle | play | puzzle-3x3-play-v0
puzzle-4x4-play-v0
puzzle-4x5-play-v0
puzzle-4x6-play-v0 | 1M
1M
3M
5M | 1000
1000
3000
5000 | 1000
1000
1000
1000 |
| | noisy | puzzle-3x3-noisy-v0
puzzle-4x4-noisy-v0
puzzle-4x5-noisy-v0
puzzle-4x6-noisy-v0 | 1M
1M
3M
5M | 1000
1000
3000
5000 | 1000
1000
1000
1000 |
| visual-cube | play | visual-cube-single-play-v0
visual-cube-double-play-v0
visual-cube-triple-play-v0
visual-cube-quadruple-play-v0 | 1M
1M
3M
5M | 1000
1000
3000
5000 | 1000
1000
1000
1000 |
| | noisy | visual-cube-single-noisy-v0
visual-cube-double-noisy-v0
visual-cube-triple-noisy-v0
visual-cube-quadruple-noisy-v0 | 1M
1M
3M
5M | 1000
1000
3000
5000 | 1000
1000
1000
1000 |
| visual-scene | play | visual-scene-play-v0 | 1M | 1000 | 1000 |
| | noisy | visual-scene-noisy-v0 | 1M | 1000 | 1000 |
| visual-puzzle | play | visual-puzzle-3x3-play-v0
visual-puzzle-4x4-play-v0
visual-puzzle-4x5-play-v0
visual-puzzle-4x6-play-v0 | 1M
1M
3M
5M | 1000
1000
3000
5000 | 1000
1000
1000
1000 |
| | noisy | visual-puzzle-3x3-noisy-v0
visual-puzzle-4x4-noisy-v0
visual-puzzle-4x5-noisy-v0
visual-puzzle-4x6-noisy-v0 | 1M
1M
3M
5M | 1000
1000
3000
5000 | 1000
1000
1000
1000 |
| powderworld | play | powderworld-easy-play-v0
powderworld-medium-play-v0
powderworld-hard-play-v0 | 1M
3M
5M | 1000
3000
5000 | 1000
1000
1000 |

Table 9: **Designated default tasks for single-task environments.** For single-task (`singletask`) variants, each environment provides five tasks corresponding to the five evaluation goals, with the most representative one chosen as the default task.

| Environment Type | Environment | Default Task |
|---|---|---|
| pointmaze | pointmaze-medium-v0 | task1 |
| | pointmaze-large-v0 | task1 |
| | pointmaze-giant-v0 | task1 |
| | pointmaze-teleport-v0 | task1 |
| antmaze | antmaze-medium-v0 | task1 |
| | antmaze-large-v0 | task1 |
| | antmaze-giant-v0 | task1 |
| | antmaze-teleport-v0 | task1 |
| humanoidmaze | humanoidmaze-medium-v0 | task1 |
| | humanoidmaze-large-v0 | task1 |
| | humanoidmaze-giant-v0 | task1 |
| antsoccer | antsoccer-arena-v0 | task4 |
| | antsoccer-medium-v0 | task4 |
| visual-antmaze | visual-antmaze-medium-v0 | task1 |
| | visual-antmaze-large-v0 | task1 |
| | visual-antmaze-giant-v0 | task1 |
| | visual-antmaze-teleport-v0 | task1 |
| visual-humanoidmaze | visual-humanoidmaze-medium-v0 | task1 |
| | visual-humanoidmaze-large-v0 | task1 |
| | visual-humanoidmaze-giant-v0 | task1 |
| cube | cube-single-v0 | task2 |
| | cube-double-v0 | task2 |
| | cube-triple-v0 | task2 |
| | cube-quadruple-v0 | task2 |
| scene | scene-v0 | task2 |
| puzzle | puzzle-3x3-v0 | task4 |
| | puzzle-4x4-v0 | task4 |
| | puzzle-4x5-v0 | task2 |
| | puzzle-4x6-v0 | task2 |
| visual-cube | visual-cube-single-v0 | task2 |
| | visual-cube-double-v0 | task2 |
| | visual-cube-triple-v0 | task2 |
| | visual-cube-quadruple-v0 | task2 |
| visual-scene | visual-scene-v0 | task2 |
| visual-puzzle | visual-puzzle-3x3-v0 | task4 |
| | visual-puzzle-4x4-v0 | task4 |
| | visual-puzzle-4x5-v0 | task2 |
| | visual-puzzle-4x6-v0 | task2 |

Table 10: **Common hyperparameters.**

| Hyperparameter | Value |
|---|---|
| Learning rate | 0.0003 |
| Optimizer | Adam (Kingma & Ba, 2015) |
| # gradient steps | 1000000 (states), 500000 (pixels) |
| Minibatch size | 1024 (states), 256 (pixels) |
| MLP dimensions | $(512, 512, 512)$ |
| Nonlinearity | GELU (Hendrycks & Gimpel, 2016) |
| Target smoothing coefficient | 0.005 |
| Discount factor $\gamma$ | 0.995 ({antmaze, pointmaze}-giant, humanoidmaze), 0.99 (others) |
| Image augmentation probability | 0.5 (pixel-based manipulation), 0 (others) |
| GCIVL/GCIQL expectile $\kappa$ | 0.9 |
| GCIQL expectile $\kappa$ | 0.9 |
| QRL quasimetric | IQE (Wang & Isola, 2022) |
| QRL latent dimension | 512 (64 components $\times$ 8-dimensional latents) |
| QRL margin $\epsilon$ | 0.05 |
| CRL latent dimension | 512 |
| HIQL expectile $\kappa$ | 0.7 |
| HIQL subgoal step $k$ | 100 (humanoidmaze), 25 (other locomotion), 10 (others) |
| HIQL subgoal representation dimension | 10 |
| Policy $(p_{\mathrm{cur}}^{\mathcal{D}}, p_{\mathrm{traj}}^{\mathcal{D}}, p_{\mathrm{geom}}^{\mathcal{D}}, p_{\mathrm{rand}}^{\mathcal{D}})$ ratio for $p_{\mathrm{mixed}}^{\mathcal{D}}$ | $(0, 0.5, 0, 0.5)$ (stitch), $(0, 0, 0, 1)$ (explore), $(0, 1, 0, 0)$ (others) |
| Value $(p_{\mathrm{cur}}^{\mathcal{D}}, p_{\mathrm{traj}}^{\mathcal{D}}, p_{\mathrm{geom}}^{\mathcal{D}}, p_{\mathrm{rand}}^{\mathcal{D}})$ ratio for $p_{\mathrm{mixed}}^{\mathcal{D}}$ | $(0.2, 0, 0.5, 0.3)$ |

Table 11: **Hyperparameters for policy extraction.** Each cell indicates the policy extraction method and its $\alpha$ value (*i.e.*, the temperature (AWR) or the BC coefficient (DDPG+BC)).

| Environment Type | Dataset Type | Dataset | GCIVL | GCIQL | QRL | CRL | HIQL |
|---|---|---|---|---|---|---|---|
| pointmaze | navigate | pointmaze-medium-navigate-v0 | AWR 10.0 | DDPG 0.003 | DDPG 0.0003 | DDPG 0.03 | AWR 3.0 |
| | | pointmaze-large-navigate-v0 | AWR 10.0 | DDPG 0.003 | DDPG 0.0003 | DDPG 0.03 | AWR 3.0 |
| | | pointmaze-giant-navigate-v0 | AWR 10.0 | DDPG 0.003 | DDPG 0.0003 | DDPG 0.03 | AWR 3.0 |
| | | pointmaze-teleport-navigate-v0 | AWR 10.0 | DDPG 0.003 | DDPG 0.0003 | DDPG 0.03 | AWR 3.0 |
| | stitch | pointmaze-medium-stitch-v0 | AWR 10.0 | DDPG 0.003 | DDPG 0.0003 | DDPG 0.03 | AWR 3.0 |
| | | pointmaze-large-stitch-v0 | AWR 10.0 | DDPG 0.003 | DDPG 0.0003 | DDPG 0.03 | AWR 3.0 |
| | | pointmaze-giant-stitch-v0 | AWR 10.0 | DDPG 0.003 | DDPG 0.0003 | DDPG 0.03 | AWR 3.0 |
| | | pointmaze-teleport-stitch-v0 | AWR 10.0 | DDPG 0.003 | DDPG 0.0003 | DDPG 0.03 | AWR 3.0 |
| antmaze | navigate | antmaze-medium-navigate-v0 | AWR 10.0 | DDPG 0.3 | DDPG 0.003 | DDPG 0.1 | AWR 3.0 |
| | | antmaze-large-navigate-v0 | AWR 10.0 | DDPG 0.3 | DDPG 0.003 | DDPG 0.1 | AWR 3.0 |
| | | antmaze-giant-navigate-v0 | AWR 10.0 | DDPG 0.3 | DDPG 0.003 | DDPG 0.1 | AWR 3.0 |
| | | antmaze-teleport-navigate-v0 | AWR 10.0 | DDPG 0.3 | DDPG 0.003 | DDPG 0.1 | AWR 3.0 |
| | stitch | antmaze-medium-stitch-v0 | AWR 10.0 | DDPG 0.3 | DDPG 0.003 | DDPG 0.1 | AWR 3.0 |
| | | antmaze-large-stitch-v0 | AWR 10.0 | DDPG 0.3 | DDPG 0.003 | DDPG 0.1 | AWR 3.0 |
| | | antmaze-giant-stitch-v0 | AWR 10.0 | DDPG 0.3 | DDPG 0.003 | DDPG 0.1 | AWR 3.0 |
| | | antmaze-teleport-stitch-v0 | AWR 10.0 | DDPG 0.3 | DDPG 0.003 | DDPG 0.1 | AWR 3.0 |
| | explore | antmaze-medium-explore-v0 | AWR 10.0 | DDPG 0.01 | DDPG 0.001 | DDPG 0.003 | AWR 10.0 |
| | | antmaze-large-explore-v0 | AWR 10.0 | DDPG 0.01 | DDPG 0.001 | DDPG 0.003 | AWR 10.0 |
| | | antmaze-teleport-explore-v0 | AWR 10.0 | DDPG 0.01 | DDPG 0.001 | DDPG 0.003 | AWR 10.0 |
| humanoidmaze | navigate | humanoidmaze-medium-navigate-v0 | AWR 10.0 | DDPG 0.1 | DDPG 0.001 | DDPG 0.1 | AWR 3.0 |
| | | humanoidmaze-large-navigate-v0 | AWR 10.0 | DDPG 0.1 | DDPG 0.001 | DDPG 0.1 | AWR 3.0 |
| | | humanoidmaze-giant-navigate-v0 | AWR 10.0 | DDPG 0.1 | DDPG 0.001 | DDPG 0.1 | AWR 3.0 |
| | stitch | humanoidmaze-medium-stitch-v0 | AWR 10.0 | DDPG 0.1 | DDPG 0.001 | DDPG 0.1 | AWR 3.0 |
| | | humanoidmaze-large-stitch-v0 | AWR 10.0 | DDPG 0.1 | DDPG 0.001 | DDPG 0.1 | AWR 3.0 |
| | | humanoidmaze-giant-stitch-v0 | AWR 10.0 | DDPG 0.1 | DDPG 0.001 | DDPG 0.1 | AWR 3.0 |
| antsoccer | navigate | antsoccer-arena-navigate-v0 | AWR 10.0 | DDPG 0.1 | DDPG 0.003 | DDPG 0.3 | AWR 3.0 |
| | | antsoccer-medium-navigate-v0 | AWR 10.0 | DDPG 0.1 | DDPG 0.003 | DDPG 0.3 | AWR 3.0 |
| | stitch | antsoccer-arena-stitch-v0 | AWR 10.0 | DDPG 0.1 | DDPG 0.003 | DDPG 0.3 | AWR 3.0 |
| | | antsoccer-medium-stitch-v0 | AWR 10.0 | DDPG 0.1 | DDPG 0.003 | DDPG 0.3 | AWR 3.0 |
| visual-antmaze | navigate | visual-antmaze-medium-navigate-v0 | AWR 10.0 | DDPG 0.3 | DDPG 0.003 | DDPG 0.1 | AWR 3.0 |
| | | visual-antmaze-large-navigate-v0 | AWR 10.0 | DDPG 0.3 | DDPG 0.003 | DDPG 0.1 | AWR 3.0 |
| | | visual-antmaze-giant-navigate-v0 | AWR 10.0 | DDPG 0.3 | DDPG 0.003 | DDPG 0.1 | AWR 3.0 |
| | | visual-antmaze-teleport-navigate-v0 | AWR 10.0 | DDPG 0.3 | DDPG 0.003 | DDPG 0.1 | AWR 3.0 |
| | stitch | visual-antmaze-medium-stitch-v0 | AWR 10.0 | DDPG 0.3 | DDPG 0.003 | DDPG 0.1 | AWR 3.0 |
| | | visual-antmaze-large-stitch-v0 | AWR 10.0 | DDPG 0.3 | DDPG 0.003 | DDPG 0.1 | AWR 3.0 |
| | | visual-antmaze-giant-stitch-v0 | AWR 10.0 | DDPG 0.3 | DDPG 0.003 | DDPG 0.1 | AWR 3.0 |
| | | visual-antmaze-teleport-stitch-v0 | AWR 10.0 | DDPG 0.3 | DDPG 0.003 | DDPG 0.1 | AWR 3.0 |
| | explore | visual-antmaze-medium-explore-v0 | AWR 10.0 | DDPG 0.01 | DDPG 0.001 | DDPG 0.003 | AWR 10.0 |
| | | visual-antmaze-large-explore-v0 | AWR 10.0 | DDPG 0.01 | DDPG 0.001 | DDPG 0.003 | AWR 10.0 |
| | | visual-antmaze-teleport-explore-v0 | AWR 10.0 | DDPG 0.01 | DDPG 0.001 | DDPG 0.003 | AWR 10.0 |
| visual-humanoidmaze | navigate | visual-humanoidmaze-medium-navigate-v0 | AWR 10.0 | DDPG 0.1 | DDPG 0.001 | DDPG 0.1 | AWR 3.0 |
| | | visual-humanoidmaze-large-navigate-v0 | AWR 10.0 | DDPG 0.1 | DDPG 0.001 | DDPG 0.1 | AWR 3.0 |
| | | visual-humanoidmaze-giant-navigate-v0 | AWR 10.0 | DDPG 0.1 | DDPG 0.001 | DDPG 0.1 | AWR 3.0 |
| | stitch | visual-humanoidmaze-medium-stitch-v0 | AWR 10.0 | DDPG 0.1 | DDPG 0.001 | DDPG 0.1 | AWR 3.0 |
| | | visual-humanoidmaze-large-stitch-v0 | AWR 10.0 | DDPG 0.1 | DDPG 0.001 | DDPG 0.1 | AWR 3.0 |
| | | visual-humanoidmaze-giant-stitch-v0 | AWR 10.0 | DDPG 0.1 | DDPG 0.001 | DDPG 0.1 | AWR 3.0 |
| cube | play | cube-single-play-v0 | AWR 10.0 | DDPG 1.0 | DDPG 0.3 | DDPG 3.0 | AWR 3.0 |
| | | cube-double-play-v0 | AWR 10.0 | DDPG 1.0 | DDPG 0.3 | DDPG 3.0 | AWR 3.0 |
| | | cube-triple-play-v0 | AWR 10.0 | DDPG 1.0 | DDPG 0.3 | DDPG 3.0 | AWR 3.0 |
| | | cube-quadruple-play-v0 | AWR 10.0 | DDPG 1.0 | DDPG 0.3 | DDPG 3.0 | AWR 3.0 |
| | noisy | cube-single-noisy-v0 | AWR 10.0 | DDPG 0.03 | DDPG 0.03 | DDPG 0.1 | AWR 3.0 |
| | | cube-double-noisy-v0 | AWR 10.0 | DDPG 0.03 | DDPG 0.03 | DDPG 0.1 | AWR 3.0 |
| | | cube-triple-noisy-v0 | AWR 10.0 | DDPG 0.03 | DDPG 0.03 | DDPG 0.1 | AWR 3.0 |
| | | cube-quadruple-noisy-v0 | AWR 10.0 | DDPG 0.03 | DDPG 0.03 | DDPG 0.1 | AWR 3.0 |
| scene | play | scene-play-v0 | AWR 10.0 | DDPG 1.0 | DDPG 0.3 | DDPG 3.0 | AWR 3.0 |
| | noisy | scene-noisy-v0 | AWR 10.0 | DDPG 0.03 | DDPG 0.03 | DDPG 0.1 | AWR 3.0 |
| puzzle | play | puzzle-3x3-play-v0 | AWR 10.0 | DDPG 1.0 | DDPG 0.3 | DDPG 3.0 | AWR 3.0 |
| | | puzzle-4x4-play-v0 | AWR 10.0 | DDPG 1.0 | DDPG 0.3 | DDPG 3.0 | AWR 3.0 |
| | | puzzle-4x5-play-v0 | AWR 10.0 | DDPG 1.0 | DDPG 0.3 | DDPG 3.0 | AWR 3.0 |
| | | puzzle-4x6-play-v0 | AWR 10.0 | DDPG 1.0 | DDPG 0.3 | DDPG 3.0 | AWR 3.0 |
| | noisy | puzzle-3x3-noisy-v0 | AWR 10.0 | DDPG 0.03 | DDPG 0.03 | DDPG 0.1 | AWR 3.0 |
| | | puzzle-4x4-noisy-v0 | AWR 10.0 | DDPG 0.03 | DDPG 0.03 | DDPG 0.1 | AWR 3.0 |
| | | puzzle-4x5-noisy-v0 | AWR 10.0 | DDPG 0.03 | DDPG 0.03 | DDPG 0.1 | AWR 3.0 |
| | | puzzle-4x6-noisy-v0 | AWR 10.0 | DDPG 0.03 | DDPG 0.03 | DDPG 0.1 | AWR 3.0 |
| visual-cube | play | visual-cube-single-play-v0 | AWR 10.0 | DDPG 1.0 | DDPG 0.3 | DDPG 3.0 | AWR 3.0 |
| | | visual-cube-double-play-v0 | AWR 10.0 | DDPG 1.0 | DDPG 0.3 | DDPG 3.0 | AWR 3.0 |
| | | visual-cube-triple-play-v0 | AWR 10.0 | DDPG 1.0 | DDPG 0.3 | DDPG 3.0 | AWR 3.0 |
| | | visual-cube-quadruple-play-v0 | AWR 10.0 | DDPG 1.0 | DDPG 0.3 | DDPG 3.0 | AWR 3.0 |
| | noisy | visual-cube-single-noisy-v0 | AWR 10.0 | DDPG 0.03 | DDPG 0.03 | DDPG 0.1 | AWR 3.0 |
| | | visual-cube-double-noisy-v0 | AWR 10.0 | DDPG 0.03 | DDPG 0.03 | DDPG 0.1 | AWR 3.0 |
| | | visual-cube-triple-noisy-v0 | AWR 10.0 | DDPG 0.03 | DDPG 0.03 | DDPG 0.1 | AWR 3.0 |
| | | visual-cube-quadruple-noisy-v0 | AWR 10.0 | DDPG 0.03 | DDPG 0.03 | DDPG 0.1 | AWR 3.0 |
| visual-scene | play | visual-scene-play-v0 | AWR 10.0 | DDPG 1.0 | DDPG 0.3 | DDPG 3.0 | AWR 3.0 |
| | noisy | visual-scene-noisy-v0 | AWR 10.0 | DDPG 0.03 | DDPG 0.03 | DDPG 0.1 | AWR 3.0 |
| visual-puzzle | play | visual-puzzle-3x3-play-v0 | AWR 10.0 | DDPG 1.0 | DDPG 0.3 | DDPG 3.0 | AWR 3.0 |
| | | visual-puzzle-4x4-play-v0 | AWR 10.0 | DDPG 1.0 | DDPG 0.3 | DDPG 3.0 | AWR 3.0 |
| | | visual-puzzle-4x5-play-v0 | AWR 10.0 | DDPG 1.0 | DDPG 0.3 | DDPG 3.0 | AWR 3.0 |
| | | visual-puzzle-4x6-play-v0 | AWR 10.0 | DDPG 1.0 | DDPG 0.3 | DDPG 3.0 | AWR 3.0 |
| | noisy | visual-puzzle-3x3-noisy-v0 | AWR 10.0 | DDPG 0.03 | DDPG 0.03 | DDPG 0.1 | AWR 3.0 |
| | | visual-puzzle-4x4-noisy-v0 | AWR 10.0 | DDPG 0.03 | DDPG 0.03 | DDPG 0.1 | AWR 3.0 |
| | | visual-puzzle-4x5-noisy-v0 | AWR 10.0 | DDPG 0.03 | DDPG 0.03 | DDPG 0.1 | AWR 3.0 |
| | | visual-puzzle-4x6-noisy-v0 | AWR 10.0 | DDPG 0.03 | DDPG 0.03 | DDPG 0.1 | AWR 3.0 |
| powderworld | play | powderworld-easy-play-v0 | AWR 3.0 | AWR 3.0 | AWR 3.0 | AWR 3.0 | AWR 3.0 |
| | | powderworld-medium-play-v0 | AWR 3.0 | AWR 3.0 | AWR 3.0 | AWR 3.0 | AWR 3.0 |
| | | powderworld-hard-play-v0 | AWR 3.0 | AWR 3.0 | AWR 3.0 | AWR 3.0 | AWR 3.0 |

## H  EVALUATION GOALS AND PER-GOAL BENCHMARKING RESULTS

Each task in OGBench provides five evaluation goals. We provide their full image descriptions in Figures 4 to 10, and the full per-goal evaluation results in Tables 12 to 24, which share the same format as Table 2.

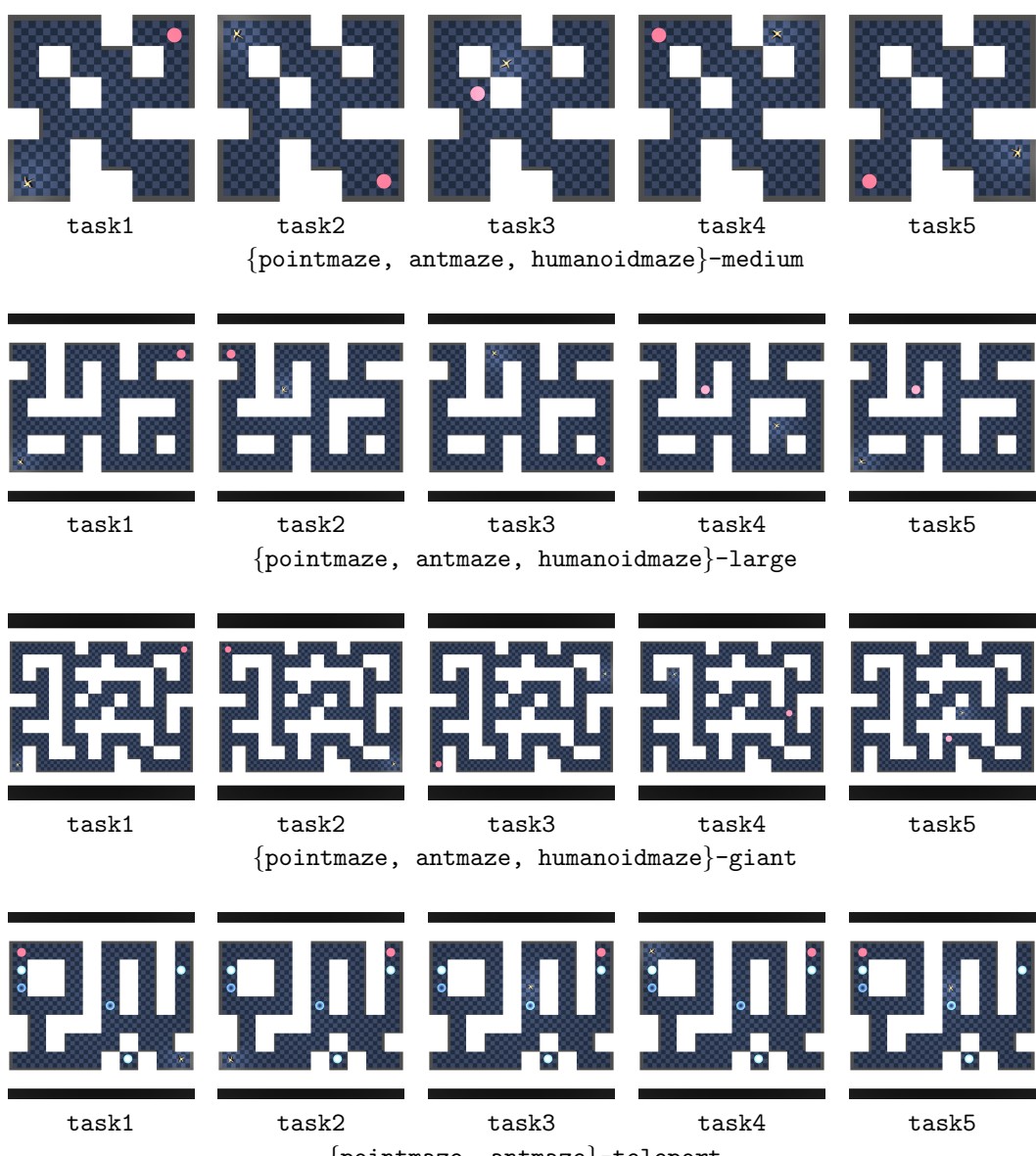

Figure 4: **PointMaze, AntMaze, and HumanoidMaze goals.**

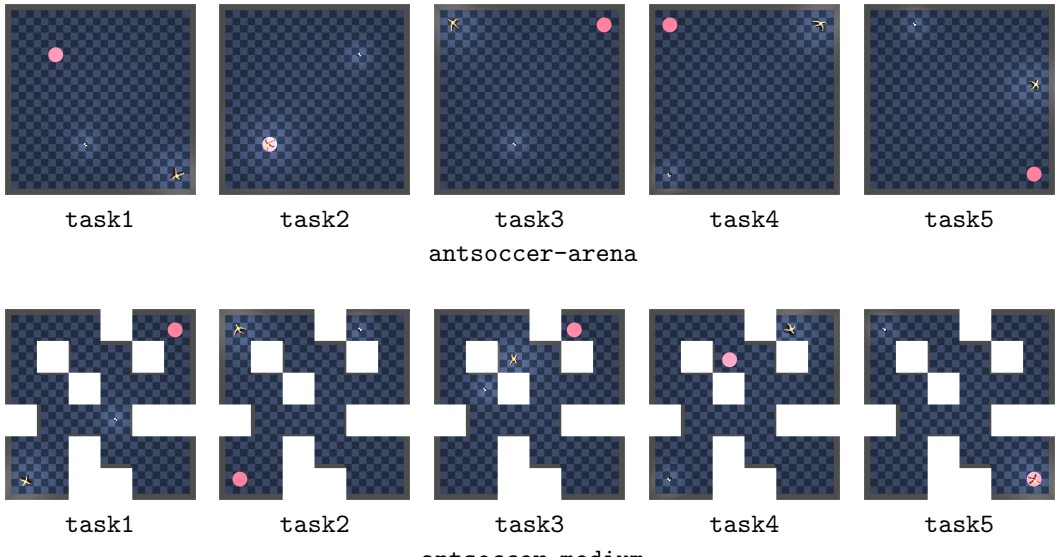

Figure 5: **AntSoccer goals.**

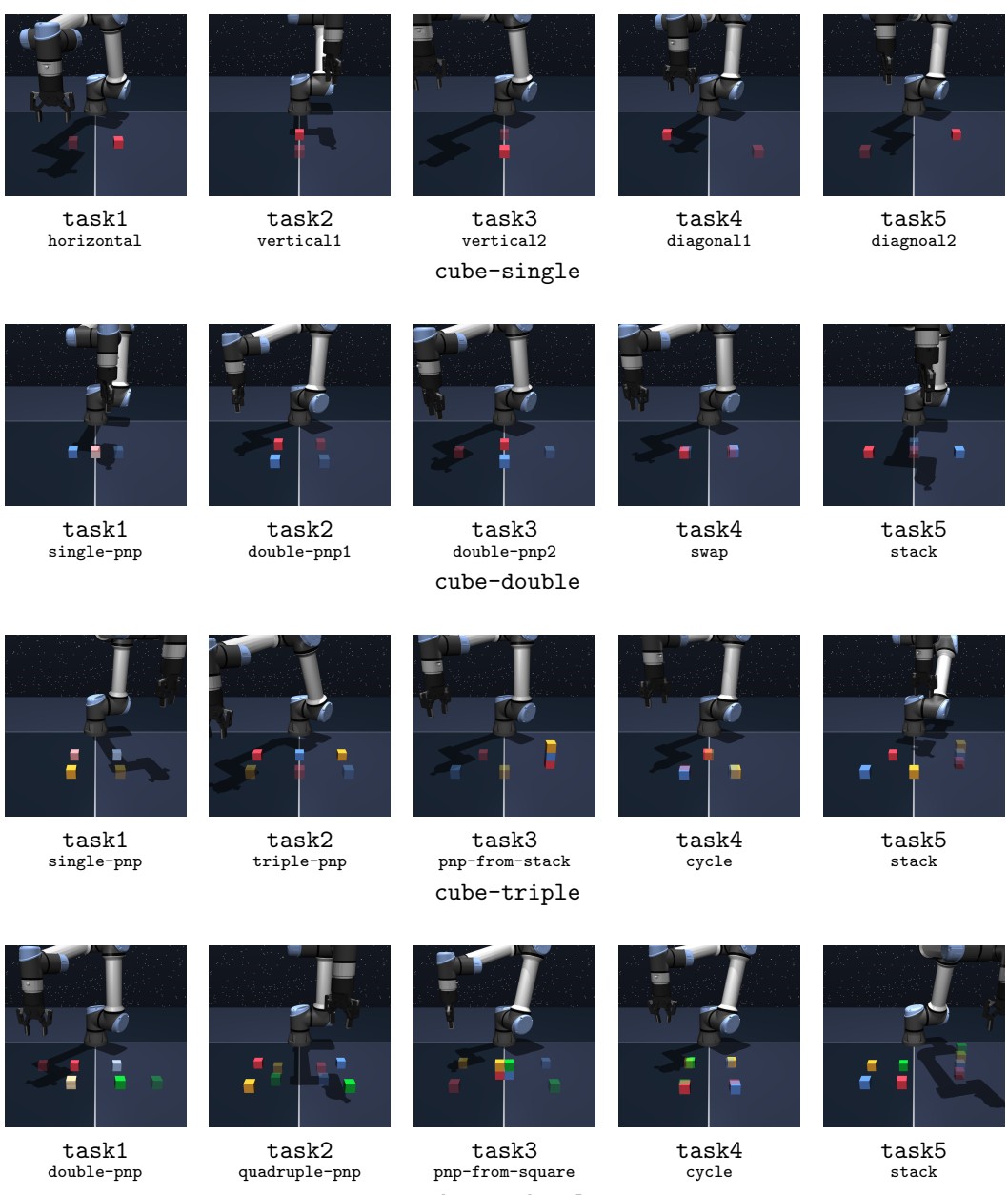

Figure 6: **Cube goals.**

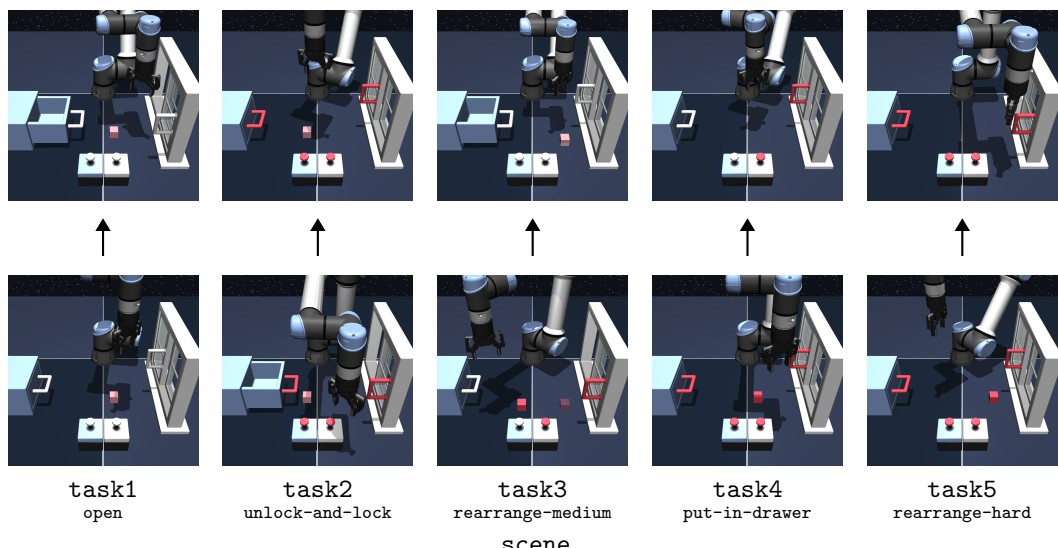

Figure 7: **Scene goals.**

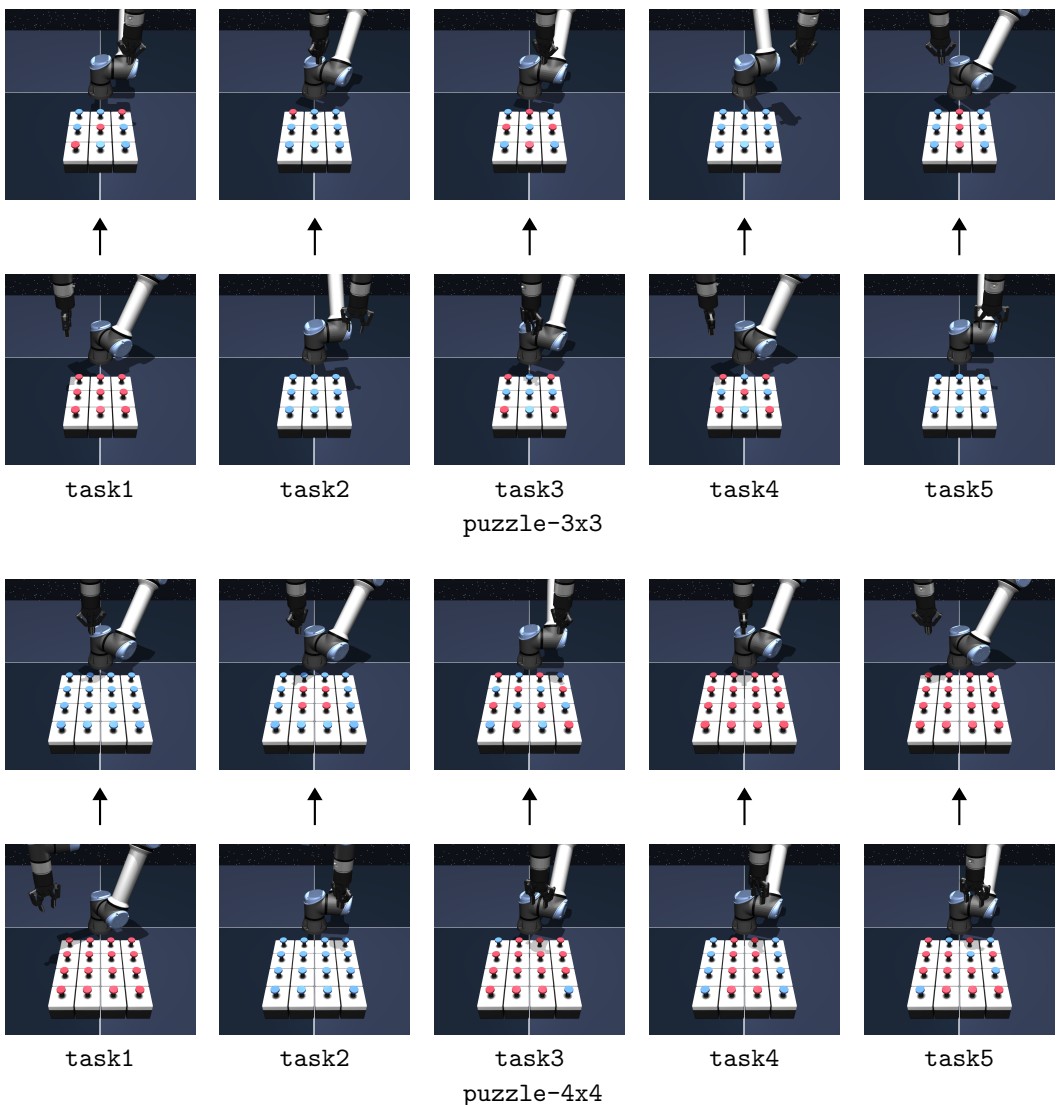

Figure 8: **Puzzle goals.**

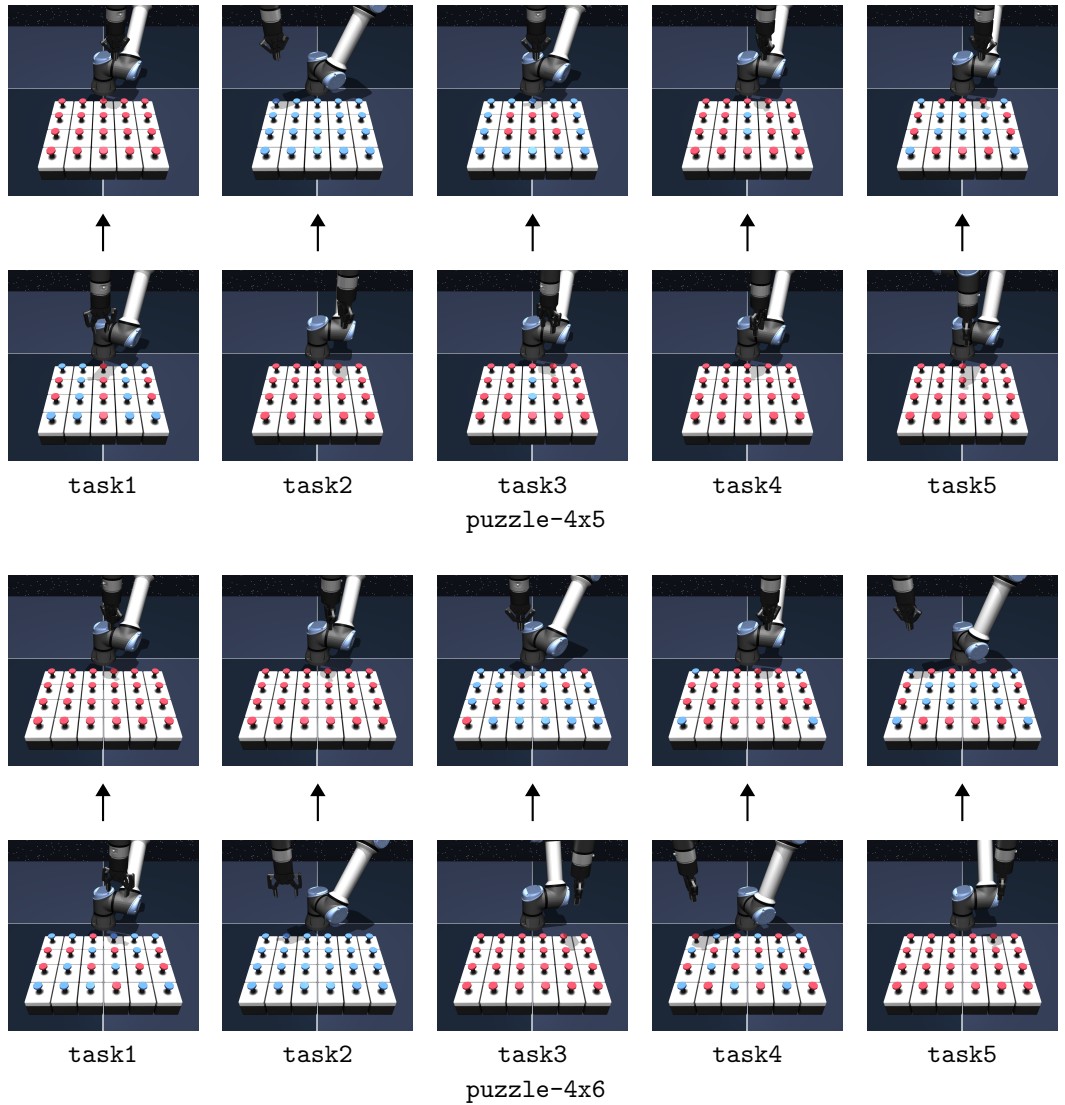

Figure 9: **Puzzle goals.**

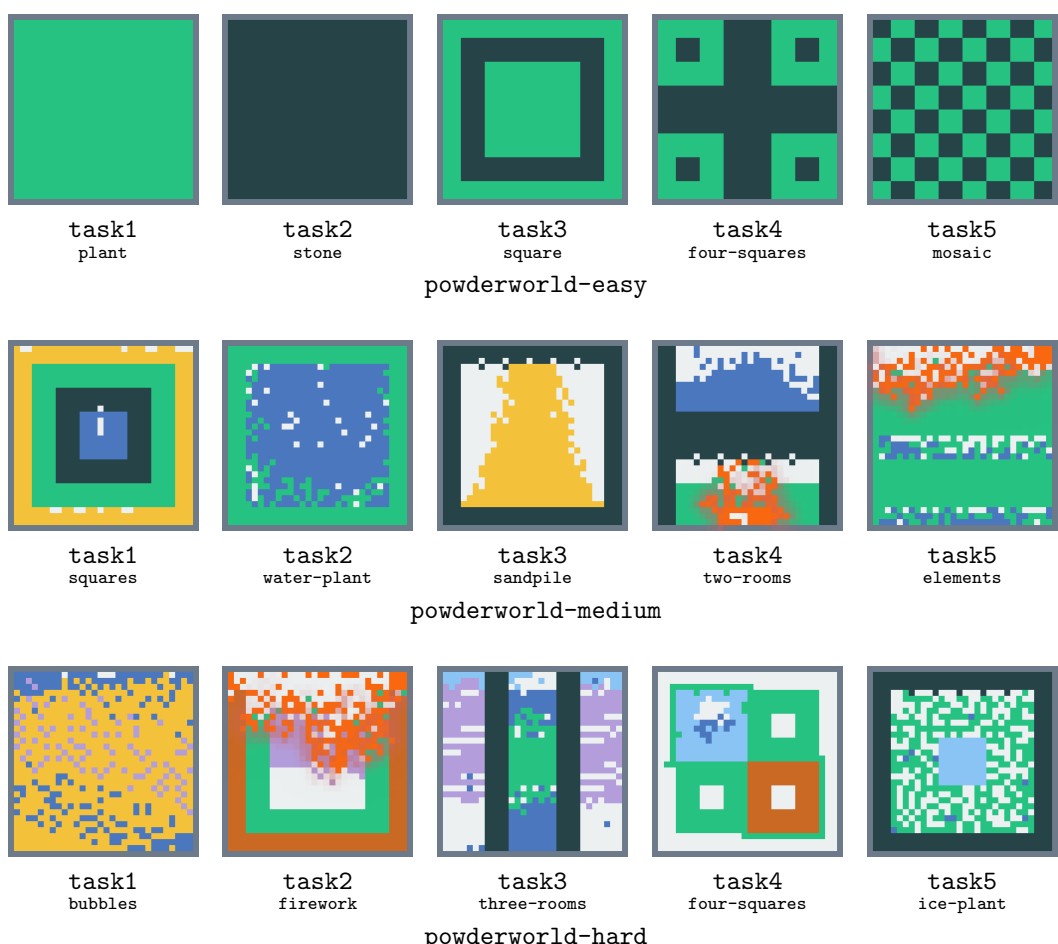

Figure 10: **Powderworld goals.**

Table 12: **Full results on PointMaze.**

| Environment Type | Dataset Type | Dataset | Task | GCBC | GCIVL | GCIQL | QRL | CRL | HIQL |
|---|---|---|---|---|---|---|---|---|---|
| pointmaze | navigate | pointmaze-medium-navigate-v0 | task1 | 30 ±27 | 88 ±16 | **97** ±4 | **100** ±0 | 20 ±6 | **99** ±1 |
| | | | task2 | 3 ±2 | **95** ±10 | 76 ±29 | 94 ±17 | 45 ±25 | 87 ±7 |
| | | | task3 | 5 ±5 | 37 ±28 | 10 ±28 | 23 ±20 | 30 ±4 | **55** ±13 |
| | | | task4 | 0 ±1 | 2 ±2 | 0 ±0 | **94** ±14 | 28 ±29 | 82 ±12 |
| | | | task5 | 4 ±3 | 92 ±7 | 79 ±6 | **97** ±8 | 24 ±13 | 70 ±10 |
| | | | overall | 9 ±6 | 63 ±6 | 53 ±8 | **82** ±5 | 29 ±7 | **79** ±5 |
| | | pointmaze-large-navigate-v0 | task1 | 63 ±11 | 76 ±23 | 86 ±14 | **95** ±8 | 42 ±27 | 83 ±13 |
| | | | task2 | 1 ±2 | 0 ±0 | 0 ±0 | **100** ±0 | 31 ±24 | 2 ±7 |
| | | | task3 | 10 ±7 | **98** ±5 | 83 ±8 | 40 ±50 | 78 ±7 | 88 ±10 |
| | | | task4 | 20 ±18 | 0 ±0 | 0 ±0 | **96** ±7 | 24 ±14 | 72 ±19 |
| | | | task5 | 52 ±17 | 53 ±20 | 0 ±0 | **96** ±7 | 20 ±10 | 46 ±16 |
| | | | overall | 29 ±6 | 45 ±5 | 34 ±3 | **86** ±9 | 39 ±7 | 58 ±5 |
| | | pointmaze-giant-navigate-v0 | task1 | 1 ±3 | 0 ±0 | 0 ±0 | **98** ±7 | 6 ±15 | 0 ±0 |
| | | | task2 | 1 ±4 | 0 ±0 | 0 ±0 | **92** ±16 | 28 ±10 | 72 ±17 |
| | | | task3 | 0 ±0 | 0 ±1 | 0 ±0 | **68** ±27 | 9 ±5 | 32 ±11 |
| | | | task4 | 0 ±0 | 0 ±0 | 0 ±0 | **66** ±20 | 64 ±17 | 60 ±22 |
| | | | task5 | 5 ±12 | 0 ±0 | 0 ±0 | 19 ±32 | 29 ±28 | **66** ±20 |
| | | | overall | 1 ±2 | 0 ±0 | 0 ±0 | **68** ±7 | 27 ±10 | 46 ±9 |
| | | pointmaze-teleport-navigate-v0 | task1 | 1 ±2 | **33** ±12 | 0 ±1 | 0 ±0 | 3 ±3 | 5 ±5 |
| | | | task2 | 4 ±6 | **49** ±2 | 39 ±14 | 8 ±10 | 30 ±23 | 6 ±6 |
| | | | task3 | **50** ±4 | 46 ±5 | 31 ±19 | 2 ±5 | 26 ±6 | 39 ±9 |
| | | | task4 | 33 ±13 | **49** ±4 | 42 ±13 | 12 ±16 | 40 ±11 | 24 ±11 |
| | | | task5 | 38 ±6 | **48** ±4 | 9 ±9 | 1 ±2 | 20 ±15 | 17 ±8 |
| | | | overall | 25 ±3 | **45** ±3 | 24 ±7 | 4 ±4 | 24 ±6 | 18 ±4 |
| | stitch | pointmaze-medium-stitch-v0 | task1 | 21 ±29 | 76 ±14 | 56 ±24 | **94** ±13 | 0 ±0 | 77 ±14 |
| | | | task2 | 32 ±35 | **79** ±23 | 26 ±19 | 81 ±34 | 0 ±0 | 61 ±23 |
| | | | task3 | 33 ±34 | 69 ±16 | 0 ±0 | 66 ±29 | 2 ±3 | **82** ±13 |
| | | | task4 | 0 ±0 | 41 ±37 | 0 ±0 | 68 ±32 | 0 ±0 | **92** ±6 |
| | | | task5 | 29 ±37 | 84 ±11 | 22 ±22 | **92** ±9 | 0 ±0 | 59 ±9 |
| | | | overall | 23 ±18 | 70 ±14 | 21 ±9 | **80** ±12 | 0 ±1 | 74 ±6 |
| | | pointmaze-large-stitch-v0 | task1 | 8 ±13 | 0 ±1 | 56 ±11 | **100** ±1 | 0 ±0 | 3 ±5 |
| | | | task2 | 0 ±0 | 0 ±0 | 0 ±0 | **74** ±37 | 0 ±0 | 0 ±0 |
| | | | task3 | 26 ±28 | 60 ±29 | **98** ±4 | 74 ±23 | 0 ±0 | 59 ±25 |
| | | | task4 | 0 ±0 | 0 ±0 | 0 ±0 | **88** ±32 | 0 ±0 | 1 ±4 |
| | | | task5 | 0 ±0 | 0 ±0 | 0 ±0 | **85** ±22 | 0 ±0 | 0 ±0 |
| | | | overall | 7 ±5 | 12 ±6 | 31 ±2 | **84** ±15 | 0 ±0 | 13 ±6 |
| | | pointmaze-giant-stitch-v0 | task1 | 0 ±0 | 0 ±0 | 0 ±0 | **99** ±2 | 0 ±0 | 0 ±0 |
| | | | task2 | 0 ±0 | 0 ±0 | 0 ±0 | **80** ±27 | 0 ±0 | 0 ±0 |
| | | | task3 | 0 ±0 | 0 ±0 | 0 ±0 | **3** ±5 | 0 ±0 | 0 ±0 |
| | | | task4 | 0 ±0 | 0 ±0 | 0 ±0 | **63** ±23 | 0 ±0 | 0 ±0 |
| | | | task5 | 0 ±0 | 0 ±0 | 0 ±0 | **4** ±8 | 0 ±0 | 0 ±0 |
| | | | overall | 0 ±0 | 0 ±0 | 0 ±0 | **50** ±8 | 0 ±0 | 0 ±0 |
| | | pointmaze-teleport-stitch-v0 | task1 | 28 ±20 | **34** ±14 | 0 ±0 | 0 ±0 | 0 ±0 | 24 ±13 |
| | | | task2 | 13 ±15 | **41** ±8 | 12 ±14 | 7 ±7 | 0 ±0 | 23 ±11 |
| | | | task3 | 48 ±8 | **50** ±5 | 47 ±2 | 15 ±13 | 0 ±0 | 46 ±9 |
| | | | task4 | 40 ±16 | **50** ±6 | 46 ±5 | 19 ±12 | 8 ±8 | 46 ±5 |
| | | | task5 | 29 ±16 | **48** ±5 | 21 ±7 | 1 ±3 | 13 ±13 | 31 ±10 |
| | | | overall | 31 ±9 | **44** ±2 | 25 ±3 | 9 ±5 | 4 ±3 | 34 ±4 |

Table 13: **Full results on AntMaze.**

| Environment Type | Dataset Type | Dataset | Task | GCBC | GCIVL | GCIQL | QRL | CRL | HIQL |
|---|---|---|---|---|---|---|---|---|---|
| antmaze | navigate | antmaze-medium-navigate-v0 | task1 | $35_{\pm9}$ | $81_{\pm10}$ | $63_{\pm9}$ | $93_{\pm2}$ | $\mathbf{97}_{\pm1}$ | $94_{\pm2}$ |
| | | | task2 | $21_{\pm7}$ | $85_{\pm5}$ | $78_{\pm8}$ | $90_{\pm5}$ | $95_{\pm2}$ | $\mathbf{97}_{\pm1}$ |
| | | | task3 | $24_{\pm6}$ | $60_{\pm13}$ | $71_{\pm8}$ | $86_{\pm6}$ | $92_{\pm3}$ | $\mathbf{96}_{\pm2}$ |
| | | | task4 | $28_{\pm7}$ | $42_{\pm25}$ | $59_{\pm12}$ | $83_{\pm4}$ | $94_{\pm5}$ | $\mathbf{96}_{\pm2}$ |
| | | | task5 | $37_{\pm10}$ | $92_{\pm3}$ | $85_{\pm7}$ | $88_{\pm8}$ | $\mathbf{96}_{\pm2}$ | $\mathbf{96}_{\pm2}$ |
| | | | overall | $29_{\pm4}$ | $72_{\pm8}$ | $71_{\pm4}$ | $88_{\pm3}$ | $95_{\pm1}$ | $\mathbf{96}_{\pm1}$ |
| | | antmaze-large-navigate-v0 | task1 | $6_{\pm3}$ | $16_{\pm12}$ | $21_{\pm6}$ | $71_{\pm15}$ | $91_{\pm3}$ | $\mathbf{93}_{\pm3}$ |
| | | | task2 | $16_{\pm4}$ | $5_{\pm6}$ | $25_{\pm7}$ | $\mathbf{77}_{\pm7}$ | $62_{\pm14}$ | $78_{\pm9}$ |
| | | | task3 | $65_{\pm4}$ | $49_{\pm18}$ | $80_{\pm5}$ | $\mathbf{94}_{\pm2}$ | $91_{\pm2}$ | $\mathbf{96}_{\pm2}$ |
| | | | task4 | $14_{\pm3}$ | $2_{\pm2}$ | $19_{\pm6}$ | $64_{\pm8}$ | $85_{\pm11}$ | $\mathbf{94}_{\pm2}$ |
| | | | task5 | $18_{\pm4}$ | $5_{\pm2}$ | $26_{\pm9}$ | $67_{\pm9}$ | $85_{\pm3}$ | $\mathbf{94}_{\pm3}$ |
| | | | overall | $24_{\pm2}$ | $16_{\pm5}$ | $34_{\pm4}$ | $75_{\pm6}$ | $83_{\pm4}$ | $\mathbf{91}_{\pm2}$ |
| | | antmaze-giant-navigate-v0 | task1 | $0_{\pm0}$ | $0_{\pm0}$ | $0_{\pm0}$ | $1_{\pm2}$ | $2_{\pm2}$ | $\mathbf{47}_{\pm10}$ |
| | | | task2 | $0_{\pm0}$ | $0_{\pm0}$ | $0_{\pm0}$ | $17_{\pm5}$ | $21_{\pm10}$ | $\mathbf{74}_{\pm5}$ |
| | | | task3 | $0_{\pm0}$ | $0_{\pm0}$ | $0_{\pm0}$ | $14_{\pm8}$ | $5_{\pm5}$ | $\mathbf{55}_{\pm7}$ |
| | | | task4 | $0_{\pm0}$ | $0_{\pm0}$ | $0_{\pm0}$ | $18_{\pm6}$ | $35_{\pm9}$ | $\mathbf{69}_{\pm5}$ |
| | | | task5 | $1_{\pm1}$ | $1_{\pm1}$ | $1_{\pm1}$ | $18_{\pm5}$ | $16_{\pm10}$ | $\mathbf{82}_{\pm4}$ |
| | | | overall | $0_{\pm0}$ | $0_{\pm0}$ | $0_{\pm0}$ | $14_{\pm3}$ | $16_{\pm3}$ | $\mathbf{65}_{\pm5}$ |
| | | antmaze-teleport-navigate-v0 | task1 | $17_{\pm5}$ | $35_{\pm5}$ | $26_{\pm5}$ | $31_{\pm6}$ | $35_{\pm5}$ | $37_{\pm5}$ |
| | | | task2 | $51_{\pm5}$ | $41_{\pm5}$ | $58_{\pm8}$ | $47_{\pm22}$ | $\mathbf{92}_{\pm3}$ | $66_{\pm8}$ |
| | | | task3 | $22_{\pm3}$ | $36_{\pm8}$ | $31_{\pm5}$ | $35_{\pm6}$ | $\mathbf{47}_{\pm4}$ | $37_{\pm5}$ |
| | | | task4 | $25_{\pm5}$ | $45_{\pm3}$ | $33_{\pm5}$ | $33_{\pm6}$ | $\mathbf{50}_{\pm2}$ | $30_{\pm2}$ |
| | | | task5 | $14_{\pm6}$ | $38_{\pm6}$ | $26_{\pm9}$ | $28_{\pm8}$ | $\mathbf{44}_{\pm3}$ | $41_{\pm8}$ |
| | | | overall | $26_{\pm3}$ | $39_{\pm3}$ | $35_{\pm5}$ | $35_{\pm5}$ | $\mathbf{53}_{\pm2}$ | $42_{\pm3}$ |
| | stitch | antmaze-medium-stitch-v0 | task1 | $70_{\pm33}$ | $76_{\pm13}$ | $17_{\pm12}$ | $43_{\pm20}$ | $43_{\pm10}$ | $\mathbf{92}_{\pm2}$ |
| | | | task2 | $65_{\pm19}$ | $80_{\pm4}$ | $22_{\pm16}$ | $61_{\pm12}$ | $46_{\pm14}$ | $\mathbf{94}_{\pm3}$ |
| | | | task3 | $21_{\pm15}$ | $16_{\pm12}$ | $41_{\pm9}$ | $72_{\pm29}$ | $46_{\pm17}$ | $\mathbf{95}_{\pm2}$ |
| | | | task4 | $1_{\pm2}$ | $0_{\pm0}$ | $32_{\pm9}$ | $80_{\pm9}$ | $53_{\pm19}$ | $\mathbf{93}_{\pm2}$ |
| | | | task5 | $70_{\pm33}$ | $47_{\pm20}$ | $34_{\pm14}$ | $41_{\pm18}$ | $75_{\pm8}$ | $\mathbf{95}_{\pm3}$ |
| | | | overall | $45_{\pm11}$ | $44_{\pm6}$ | $29_{\pm6}$ | $59_{\pm7}$ | $53_{\pm6}$ | $\mathbf{94}_{\pm1}$ |
| | | antmaze-large-stitch-v0 | task1 | $2_{\pm2}$ | $23_{\pm9}$ | $0_{\pm0}$ | $7_{\pm5}$ | $1_{\pm1}$ | $\mathbf{85}_{\pm5}$ |
| | | | task2 | $0_{\pm0}$ | $0_{\pm0}$ | $0_{\pm0}$ | $10_{\pm5}$ | $4_{\pm4}$ | $\mathbf{24}_{\pm16}$ |
| | | | task3 | $15_{\pm14}$ | $69_{\pm6}$ | $37_{\pm10}$ | $73_{\pm8}$ | $43_{\pm11}$ | $\mathbf{94}_{\pm3}$ |
| | | | task4 | $0_{\pm0}$ | $0_{\pm0}$ | $0_{\pm0}$ | $1_{\pm1}$ | $5_{\pm5}$ | $\mathbf{70}_{\pm8}$ |
| | | | task5 | $0_{\pm0}$ | $0_{\pm0}$ | $0_{\pm0}$ | $1_{\pm1}$ | $1_{\pm2}$ | $\mathbf{60}_{\pm9}$ |
| | | | overall | $3_{\pm3}$ | $18_{\pm2}$ | $7_{\pm2}$ | $18_{\pm2}$ | $11_{\pm2}$ | $\mathbf{67}_{\pm5}$ |
| | | antmaze-giant-stitch-v0 | task1 | $\mathbf{0}_{\pm0}$ | $\mathbf{0}_{\pm0}$ | $\mathbf{0}_{\pm0}$ | $\mathbf{0}_{\pm0}$ | $\mathbf{0}_{\pm0}$ | $\mathbf{0}_{\pm1}$ |
| | | | task2 | $0_{\pm0}$ | $0_{\pm0}$ | $0_{\pm0}$ | $0_{\pm0}$ | $0_{\pm0}$ | $\mathbf{5}_{\pm5}$ |
| | | | task3 | $\mathbf{0}_{\pm0}$ | $\mathbf{0}_{\pm0}$ | $\mathbf{0}_{\pm0}$ | $\mathbf{0}_{\pm0}$ | $\mathbf{0}_{\pm0}$ | $\mathbf{0}_{\pm0}$ |
| | | | task4 | $0_{\pm0}$ | $0_{\pm0}$ | $0_{\pm0}$ | $0_{\pm0}$ | $0_{\pm0}$ | $\mathbf{3}_{\pm3}$ |
| | | | task5 | $0_{\pm0}$ | $0_{\pm0}$ | $0_{\pm0}$ | $\mathbf{2}_{\pm2}$ | $0_{\pm0}$ | $0_{\pm1}$ |
| | | | overall | $0_{\pm0}$ | $0_{\pm0}$ | $0_{\pm0}$ | $0_{\pm0}$ | $0_{\pm0}$ | $\mathbf{2}_{\pm2}$ |
| | | antmaze-teleport-stitch-v0 | task1 | $21_{\pm13}$ | $39_{\pm7}$ | $12_{\pm4}$ | $22_{\pm6}$ | $30_{\pm6}$ | $\mathbf{44}_{\pm5}$ |
| | | | task2 | $39_{\pm12}$ | $\mathbf{44}_{\pm6}$ | $18_{\pm7}$ | $22_{\pm6}$ | $30_{\pm4}$ | $42_{\pm3}$ |
| | | | task3 | $34_{\pm12}$ | $\mathbf{36}_{\pm8}$ | $18_{\pm4}$ | $25_{\pm7}$ | $23_{\pm11}$ | $26_{\pm4}$ |
| | | | task4 | $\mathbf{46}_{\pm6}$ | $44_{\pm4}$ | $18_{\pm5}$ | $24_{\pm9}$ | $38_{\pm4}$ | $26_{\pm4}$ |
| | | | task5 | $16_{\pm14}$ | $33_{\pm6}$ | $17_{\pm6}$ | $26_{\pm5}$ | $32_{\pm7}$ | $\mathbf{40}_{\pm6}$ |
| | | | overall | $31_{\pm6}$ | $\mathbf{39}_{\pm3}$ | $17_{\pm2}$ | $24_{\pm5}$ | $31_{\pm4}$ | $36_{\pm2}$ |
| | explore | antmaze-medium-explore-v0 | task1 | $3_{\pm6}$ | $10_{\pm8}$ | $12_{\pm6}$ | $1_{\pm1}$ | $2_{\pm2}$ | $\mathbf{29}_{\pm17}$ |
| | | | task2 | $1_{\pm2}$ | $74_{\pm9}$ | $53_{\pm8}$ | $1_{\pm1}$ | $8_{\pm6}$ | $\mathbf{84}_{\pm10}$ |
| | | | task3 | $1_{\pm2}$ | $0_{\pm0}$ | $0_{\pm0}$ | $3_{\pm5}$ | $4_{\pm6}$ | $\mathbf{18}_{\pm24}$ |
| | | | task4 | $\mathbf{0}_{\pm0}$ | $\mathbf{0}_{\pm0}$ | $\mathbf{0}_{\pm0}$ | $\mathbf{0}_{\pm0}$ | $\mathbf{0}_{\pm0}$ | $\mathbf{0}_{\pm0}$ |
| | | | task5 | $3_{\pm4}$ | $10_{\pm6}$ | $0_{\pm0}$ | $1_{\pm1}$ | $2_{\pm2}$ | $\mathbf{52}_{\pm27}$ |
| | | | overall | $2_{\pm1}$ | $19_{\pm3}$ | $13_{\pm2}$ | $1_{\pm1}$ | $3_{\pm2}$ | $\mathbf{37}_{\pm10}$ |
| | | antmaze-large-explore-v0 | task1 | $0_{\pm0}$ | $\mathbf{37}_{\pm12}$ | $1_{\pm1}$ | $0_{\pm0}$ | $0_{\pm1}$ | $1_{\pm3}$ |
| | | | task2 | $\mathbf{0}_{\pm0}$ | $\mathbf{0}_{\pm0}$ | $\mathbf{0}_{\pm0}$ | $\mathbf{0}_{\pm0}$ | $\mathbf{0}_{\pm0}$ | $\mathbf{0}_{\pm0}$ |
| | | | task3 | $0_{\pm0}$ | $12_{\pm6}$ | $1_{\pm1}$ | $0_{\pm0}$ | $1_{\pm1}$ | $\mathbf{18}_{\pm24}$ |
| | | | task4 | $\mathbf{0}_{\pm0}$ | $\mathbf{0}_{\pm0}$ | $\mathbf{0}_{\pm0}$ | $\mathbf{0}_{\pm0}$ | $\mathbf{0}_{\pm0}$ | $\mathbf{0}_{\pm0}$ |
| | | | task5 | $\mathbf{0}_{\pm0}$ | $\mathbf{0}_{\pm0}$ | $\mathbf{0}_{\pm0}$ | $\mathbf{0}_{\pm0}$ | $\mathbf{0}_{\pm0}$ | $\mathbf{0}_{\pm0}$ |
| | | | overall | $0_{\pm0}$ | $\mathbf{10}_{\pm3}$ | $0_{\pm0}$ | $0_{\pm0}$ | $0_{\pm0}$ | $4_{\pm5}$ |
| | | antmaze-teleport-explore-v0 | task1 | $2_{\pm2}$ | $\mathbf{32}_{\pm4}$ | $0_{\pm1}$ | $0_{\pm0}$ | $2_{\pm1}$ | $\mathbf{32}_{\pm11}$ |
| | | | task2 | $0_{\pm0}$ | $2_{\pm4}$ | $8_{\pm6}$ | $0_{\pm1}$ | $5_{\pm4}$ | $\mathbf{33}_{\pm17}$ |
| | | | task3 | $4_{\pm3}$ | $\mathbf{48}_{\pm3}$ | $13_{\pm8}$ | $4_{\pm4}$ | $47_{\pm6}$ | $34_{\pm16}$ |
| | | | task4 | $2_{\pm2}$ | $\mathbf{47}_{\pm5}$ | $14_{\pm8}$ | $4_{\pm4}$ | $16_{\pm12}$ | $37_{\pm19}$ |
| | | | task5 | $4_{\pm2}$ | $31_{\pm2}$ | $2_{\pm1}$ | $3_{\pm3}$ | $28_{\pm5}$ | $\mathbf{34}_{\pm14}$ |
| | | | overall | $2_{\pm1}$ | $32_{\pm2}$ | $7_{\pm3}$ | $2_{\pm2}$ | $20_{\pm2}$ | $\mathbf{34}_{\pm15}$ |

Table 14: **Full results on HumanoidMaze.**

| Environment Type | Dataset Type | Dataset | Task | GCBC | GCIVL | GCIQL | QRL | CRL | HIQL |
|---|---|---|---|---|---|---|---|---|---|
| humanoidmaze | navigate | humanoidmaze-medium-navigate-v0 | task1 | $4_{\pm1}$ | $22_{\pm5}$ | $23_{\pm6}$ | $12_{\pm7}$ | $84_{\pm3}$ | $\mathbf{95}_{\pm2}$ |
| | | | task2 | $8_{\pm4}$ | $42_{\pm8}$ | $49_{\pm6}$ | $25_{\pm8}$ | $80_{\pm5}$ | $\mathbf{96}_{\pm2}$ |
| | | | task3 | $12_{\pm3}$ | $15_{\pm3}$ | $12_{\pm6}$ | $25_{\pm10}$ | $43_{\pm11}$ | $\mathbf{79}_{\pm6}$ |
| | | | task4 | $2_{\pm1}$ | $0_{\pm0}$ | $1_{\pm0}$ | $16_{\pm7}$ | $5_{\pm5}$ | $\mathbf{75}_{\pm6}$ |
| | | | task5 | $12_{\pm4}$ | $40_{\pm8}$ | $51_{\pm8}$ | $29_{\pm12}$ | $87_{\pm7}$ | $\mathbf{97}_{\pm1}$ |
| | | | overall | $8_{\pm2}$ | $24_{\pm2}$ | $27_{\pm2}$ | $21_{\pm8}$ | $60_{\pm4}$ | $\mathbf{89}_{\pm2}$ |
| | | humanoidmaze-large-navigate-v0 | task1 | $1_{\pm1}$ | $6_{\pm2}$ | $3_{\pm2}$ | $3_{\pm2}$ | $36_{\pm11}$ | $\mathbf{67}_{\pm4}$ |
| | | | task2 | $0_{\pm0}$ | $0_{\pm0}$ | $0_{\pm0}$ | $0_{\pm0}$ | $0_{\pm0}$ | $\mathbf{2}_{\pm3}$ |
| | | | task3 | $3_{\pm1}$ | $6_{\pm2}$ | $5_{\pm2}$ | $17_{\pm6}$ | $54_{\pm17}$ | $\mathbf{88}_{\pm3}$ |
| | | | task4 | $2_{\pm1}$ | $0_{\pm0}$ | $1_{\pm1}$ | $4_{\pm2}$ | $23_{\pm11}$ | $\mathbf{42}_{\pm11}$ |
| | | | task5 | $1_{\pm1}$ | $1_{\pm1}$ | $1_{\pm1}$ | $2_{\pm1}$ | $6_{\pm4}$ | $\mathbf{47}_{\pm10}$ |
| | | | overall | $1_{\pm0}$ | $2_{\pm1}$ | $2_{\pm1}$ | $5_{\pm1}$ | $24_{\pm4}$ | $\mathbf{49}_{\pm4}$ |
| | | humanoidmaze-giant-navigate-v0 | task1 | $0_{\pm0}$ | $0_{\pm0}$ | $0_{\pm0}$ | $0_{\pm0}$ | $1_{\pm1}$ | $\mathbf{13}_{\pm7}$ |
| | | | task2 | $0_{\pm0}$ | $1_{\pm1}$ | $1_{\pm1}$ | $2_{\pm1}$ | $9_{\pm5}$ | $\mathbf{35}_{\pm11}$ |
| | | | task3 | $0_{\pm0}$ | $0_{\pm0}$ | $0_{\pm0}$ | $0_{\pm0}$ | $2_{\pm2}$ | $\mathbf{11}_{\pm4}$ |
| | | | task4 | $0_{\pm0}$ | $0_{\pm0}$ | $0_{\pm0}$ | $0_{\pm0}$ | $\mathbf{3}_{\pm2}$ | $2_{\pm2}$ |
| | | | task5 | $1_{\pm1}$ | $0_{\pm0}$ | $1_{\pm1}$ | $\mathbf{2}_{\pm1}$ | $1_{\pm1}$ | $\mathbf{2}_{\pm2}$ |
| | | | overall | $0_{\pm0}$ | $0_{\pm0}$ | $0_{\pm0}$ | $1_{\pm0}$ | $3_{\pm2}$ | $\mathbf{12}_{\pm4}$ |
| | stitch | humanoidmaze-medium-stitch-v0 | task1 | $20_{\pm7}$ | $13_{\pm3}$ | $12_{\pm3}$ | $6_{\pm5}$ | $27_{\pm7}$ | $\mathbf{84}_{\pm5}$ |
| | | | task2 | $49_{\pm12}$ | $7_{\pm2}$ | $8_{\pm5}$ | $13_{\pm4}$ | $37_{\pm7}$ | $\mathbf{94}_{\pm2}$ |
| | | | task3 | $24_{\pm8}$ | $25_{\pm3}$ | $20_{\pm7}$ | $30_{\pm6}$ | $40_{\pm4}$ | $\mathbf{86}_{\pm4}$ |
| | | | task4 | $3_{\pm2}$ | $1_{\pm1}$ | $2_{\pm2}$ | $18_{\pm5}$ | $28_{\pm7}$ | $\mathbf{86}_{\pm4}$ |
| | | | task5 | $49_{\pm8}$ | $16_{\pm3}$ | $18_{\pm7}$ | $22_{\pm2}$ | $49_{\pm5}$ | $\mathbf{90}_{\pm4}$ |
| | | | overall | $29_{\pm5}$ | $12_{\pm2}$ | $12_{\pm3}$ | $18_{\pm2}$ | $36_{\pm2}$ | $\mathbf{88}_{\pm2}$ |
| | | humanoidmaze-large-stitch-v0 | task1 | $3_{\pm4}$ | $2_{\pm1}$ | $1_{\pm1}$ | $0_{\pm0}$ | $0_{\pm0}$ | $\mathbf{21}_{\pm5}$ |
| | | | task2 | $0_{\pm0}$ | $0_{\pm0}$ | $0_{\pm0}$ | $0_{\pm0}$ | $0_{\pm0}$ | $\mathbf{5}_{\pm2}$ |
| | | | task3 | $20_{\pm11}$ | $3_{\pm2}$ | $1_{\pm1}$ | $16_{\pm7}$ | $13_{\pm3}$ | $\mathbf{84}_{\pm4}$ |
| | | | task4 | $2_{\pm1}$ | $1_{\pm1}$ | $0_{\pm1}$ | $1_{\pm1}$ | $4_{\pm1}$ | $\mathbf{19}_{\pm4}$ |
| | | | task5 | $2_{\pm2}$ | $1_{\pm1}$ | $0_{\pm0}$ | $0_{\pm0}$ | $3_{\pm1}$ | $\mathbf{12}_{\pm2}$ |
| | | | overall | $6_{\pm3}$ | $1_{\pm1}$ | $0_{\pm0}$ | $3_{\pm1}$ | $4_{\pm1}$ | $\mathbf{28}_{\pm3}$ |
| | | humanoidmaze-giant-stitch-v0 | task1 | $0_{\pm0}$ | $0_{\pm0}$ | $0_{\pm0}$ | $0_{\pm0}$ | $0_{\pm0}$ | $\mathbf{1}_{\pm2}$ |
| | | | task2 | $0_{\pm0}$ | $1_{\pm1}$ | $0_{\pm0}$ | $1_{\pm1}$ | $0_{\pm0}$ | $\mathbf{12}_{\pm6}$ |
| | | | task3 | $0_{\pm0}$ | $0_{\pm0}$ | $0_{\pm0}$ | $0_{\pm0}$ | $0_{\pm0}$ | $\mathbf{2}_{\pm2}$ |
| | | | task4 | $0_{\pm0}$ | $0_{\pm0}$ | $0_{\pm0}$ | $0_{\pm0}$ | $0_{\pm0}$ | $\mathbf{1}_{\pm1}$ |
| | | | task5 | $0_{\pm0}$ | $0_{\pm0}$ | $1_{\pm1}$ | $\mathbf{1}_{\pm1}$ | $0_{\pm1}$ | $0_{\pm1}$ |
| | | | overall | $0_{\pm0}$ | $0_{\pm0}$ | $0_{\pm0}$ | $0_{\pm0}$ | $0_{\pm0}$ | $\mathbf{3}_{\pm2}$ |

Table 15: **Full results on AntSoccer.**

| Environment Type | Dataset Type | Dataset | Task | GCBC | GCIVL | GCIQL | QRL | CRL | HIQL |
|---|---|---|---|---|---|---|---|---|---|
| antsoccer | navigate | antsoccer-arena-navigate-v0 | task1 | $7_{\pm3}$ | $61_{\pm4}$ | $60_{\pm6}$ | $12_{\pm2}$ | $32_{\pm5}$ | $\mathbf{67}_{\pm4}$ |
| | | | task2 | $10_{\pm3}$ | $45_{\pm5}$ | $56_{\pm4}$ | $8_{\pm4}$ | $27_{\pm4}$ | $\mathbf{59}_{\pm4}$ |
| | | | task3 | $3_{\pm2}$ | $62_{\pm6}$ | $63_{\pm6}$ | $12_{\pm2}$ | $28_{\pm5}$ | $\mathbf{76}_{\pm4}$ |
| | | | task4 | $3_{\pm1}$ | $23_{\pm5}$ | $28_{\pm6}$ | $3_{\pm2}$ | $11_{\pm2}$ | $\mathbf{30}_{\pm3}$ |
| | | | task5 | $3_{\pm1}$ | $42_{\pm6}$ | $42_{\pm7}$ | $3_{\pm3}$ | $15_{\pm4}$ | $\mathbf{56}_{\pm4}$ |
| | | | overall | $5_{\pm1}$ | $47_{\pm3}$ | $50_{\pm2}$ | $8_{\pm2}$ | $23_{\pm2}$ | $\mathbf{58}_{\pm2}$ |
| | | antsoccer-medium-navigate-v0 | task1 | $9_{\pm3}$ | $17_{\pm5}$ | $29_{\pm5}$ | $8_{\pm8}$ | $13_{\pm4}$ | $\mathbf{45}_{\pm6}$ |
| | | | task2 | $\mathbf{0}_{\pm0}$ | $\mathbf{0}_{\pm0}$ | $\mathbf{0}_{\pm0}$ | $\mathbf{0}_{\pm0}$ | $\mathbf{0}_{\pm0}$ | $\mathbf{0}_{\pm0}$ |
| | | | task3 | $0_{\pm0}$ | $0_{\pm0}$ | $0_{\pm1}$ | $1_{\pm1}$ | $0_{\pm1}$ | $\mathbf{2}_{\pm1}$ |
| | | | task4 | $0_{\pm0}$ | $0_{\pm0}$ | $1_{\pm1}$ | $1_{\pm1}$ | $1_{\pm1}$ | $\mathbf{3}_{\pm1}$ |
| | | | task5 | $1_{\pm1}$ | $3_{\pm1}$ | $4_{\pm3}$ | $1_{\pm2}$ | $2_{\pm1}$ | $\mathbf{13}_{\pm5}$ |
| | | | overall | $2_{\pm0}$ | $4_{\pm1}$ | $7_{\pm1}$ | $2_{\pm2}$ | $3_{\pm1}$ | $\mathbf{13}_{\pm2}$ |
| | stitch | antsoccer-arena-stitch-v0 | task1 | $\mathbf{73}_{\pm5}$ | $37_{\pm4}$ | $6_{\pm2}$ | $2_{\pm1}$ | $2_{\pm2}$ | $24_{\pm2}$ |
| | | | task2 | $\mathbf{36}_{\pm19}$ | $13_{\pm4}$ | $2_{\pm1}$ | $2_{\pm2}$ | $1_{\pm0}$ | $14_{\pm4}$ |
| | | | task3 | $6_{\pm15}$ | $\mathbf{34}_{\pm9}$ | $1_{\pm1}$ | $1_{\pm1}$ | $0_{\pm0}$ | $20_{\pm3}$ |
| | | | task4 | $7_{\pm12}$ | $\mathbf{11}_{\pm3}$ | $0_{\pm0}$ | $0_{\pm0}$ | $0_{\pm0}$ | $7_{\pm2}$ |
| | | | task5 | $0_{\pm0}$ | $\mathbf{12}_{\pm3}$ | $1_{\pm1}$ | $0_{\pm0}$ | $0_{\pm0}$ | $\mathbf{12}_{\pm5}$ |
| | | | overall | $\mathbf{24}_{\pm8}$ | $21_{\pm3}$ | $2_{\pm0}$ | $1_{\pm1}$ | $1_{\pm0}$ | $15_{\pm1}$ |
| | | antsoccer-medium-stitch-v0 | task1 | $10_{\pm7}$ | $4_{\pm2}$ | $0_{\pm0}$ | $0_{\pm0}$ | $0_{\pm0}$ | $\mathbf{21}_{\pm6}$ |
| | | | task2 | $\mathbf{0}_{\pm0}$ | $\mathbf{0}_{\pm0}$ | $\mathbf{0}_{\pm0}$ | $\mathbf{0}_{\pm0}$ | $\mathbf{0}_{\pm0}$ | $\mathbf{0}_{\pm0}$ |
| | | | task3 | $\mathbf{0}_{\pm0}$ | $\mathbf{0}_{\pm0}$ | $\mathbf{0}_{\pm0}$ | $\mathbf{0}_{\pm0}$ | $\mathbf{0}_{\pm0}$ | $\mathbf{0}_{\pm0}$ |
| | | | task4 | $\mathbf{0}_{\pm0}$ | $\mathbf{0}_{\pm0}$ | $\mathbf{0}_{\pm0}$ | $\mathbf{0}_{\pm0}$ | $\mathbf{0}_{\pm0}$ | $\mathbf{0}_{\pm0}$ |
| | | | task5 | $\mathbf{0}_{\pm0}$ | $\mathbf{0}_{\pm0}$ | $\mathbf{0}_{\pm0}$ | $\mathbf{0}_{\pm0}$ | $\mathbf{0}_{\pm0}$ | $\mathbf{0}_{\pm1}$ |
| | | | overall | $2_{\pm1}$ | $1_{\pm0}$ | $0_{\pm0}$ | $0_{\pm0}$ | $0_{\pm0}$ | $\mathbf{4}_{\pm1}$ |

Table 16: **Full results on Visual AntMaze.**

| Environment Type | Dataset Type | Dataset | Task | GCBC | GCIVL | GCIQL | QRL | CRL | HIQL |
|---|---|---|---|---|---|---|---|---|---|
| visual-antmaze | navigate | visual-antmaze-medium-navigate-v0 | task1 | $17_{\pm6}$ | $30_{\pm7}$ | $16_{\pm3}$ | $0_{\pm0}$ | $\mathbf{92}_{\pm2}$ | $90_{\pm4}$ |
| | | | task2 | $8_{\pm2}$ | $21_{\pm6}$ | $7_{\pm2}$ | $0_{\pm0}$ | $\mathbf{94}_{\pm2}$ | $92_{\pm7}$ |
| | | | task3 | $17_{\pm1}$ | $24_{\pm5}$ | $16_{\pm4}$ | $0_{\pm0}$ | $\mathbf{98}_{\pm1}$ | $94_{\pm4}$ |
| | | | task4 | $12_{\pm2}$ | $21_{\pm3}$ | $9_{\pm2}$ | $0_{\pm0}$ | $\mathbf{94}_{\pm2}$ | $94_{\pm2}$ |
| | | | task5 | $4_{\pm2}$ | $16_{\pm5}$ | $6_{\pm2}$ | $0_{\pm0}$ | $\mathbf{94}_{\pm2}$ | $94_{\pm5}$ |
| | | | overall | $11_{\pm2}$ | $22_{\pm2}$ | $11_{\pm1}$ | $0_{\pm0}$ | $\mathbf{94}_{\pm1}$ | $93_{\pm4}$ |
| | | visual-antmaze-large-navigate-v0 | task1 | $3_{\pm1}$ | $7_{\pm2}$ | $4_{\pm3}$ | $0_{\pm0}$ | $\mathbf{78}_{\pm5}$ | $60_{\pm10}$ |
| | | | task2 | $4_{\pm3}$ | $4_{\pm1}$ | $2_{\pm1}$ | $0_{\pm0}$ | $\mathbf{80}_{\pm3}$ | $28_{\pm9}$ |
| | | | task3 | $4_{\pm2}$ | $6_{\pm2}$ | $4_{\pm1}$ | $1_{\pm1}$ | $\mathbf{90}_{\pm3}$ | $85_{\pm10}$ |
| | | | task4 | $4_{\pm2}$ | $5_{\pm3}$ | $6_{\pm1}$ | $0_{\pm1}$ | $\mathbf{88}_{\pm3}$ | $46_{\pm7}$ |
| | | | task5 | $4_{\pm2}$ | $5_{\pm1}$ | $4_{\pm2}$ | $0_{\pm0}$ | $\mathbf{83}_{\pm2}$ | $44_{\pm10}$ |
| | | | overall | $4_{\pm0}$ | $5_{\pm1}$ | $4_{\pm1}$ | $0_{\pm0}$ | $\mathbf{84}_{\pm1}$ | $53_{\pm9}$ |
| | | visual-antmaze-giant-navigate-v0 | task1 | $0_{\pm0}$ | $0_{\pm0}$ | $0_{\pm0}$ | $0_{\pm0}$ | $\mathbf{17}_{\pm2}$ | $2_{\pm1}$ |
| | | | task2 | $1_{\pm1}$ | $2_{\pm1}$ | $1_{\pm1}$ | $0_{\pm0}$ | $\mathbf{73}_{\pm9}$ | $12_{\pm8}$ |
| | | | task3 | $0_{\pm0}$ | $0_{\pm0}$ | $0_{\pm0}$ | $0_{\pm0}$ | $\mathbf{22}_{\pm6}$ | $2_{\pm3}$ |
| | | | task4 | $0_{\pm1}$ | $0_{\pm1}$ | $0_{\pm0}$ | $0_{\pm0}$ | $\mathbf{47}_{\pm5}$ | $4_{\pm2}$ |
| | | | task5 | $1_{\pm1}$ | $2_{\pm3}$ | $1_{\pm1}$ | $0_{\pm1}$ | $\mathbf{77}_{\pm5}$ | $13_{\pm11}$ |
| | | | overall | $0_{\pm0}$ | $1_{\pm1}$ | $0_{\pm0}$ | $0_{\pm0}$ | $\mathbf{47}_{\pm2}$ | $6_{\pm4}$ |
| | | visual-antmaze-teleport-navigate-v0 | task1 | $2_{\pm2}$ | $6_{\pm1}$ | $2_{\pm1}$ | $3_{\pm2}$ | $\mathbf{32}_{\pm3}$ | $32_{\pm5}$ |
| | | | task2 | $6_{\pm3}$ | $9_{\pm3}$ | $9_{\pm2}$ | $6_{\pm4}$ | $\mathbf{73}_{\pm8}$ | $40_{\pm6}$ |
| | | | task3 | $9_{\pm1}$ | $12_{\pm3}$ | $9_{\pm2}$ | $10_{\pm4}$ | $\mathbf{47}_{\pm3}$ | $33_{\pm1}$ |
| | | | task4 | $10_{\pm2}$ | $10_{\pm2}$ | $8_{\pm3}$ | $6_{\pm4}$ | $\mathbf{50}_{\pm4}$ | $44_{\pm5}$ |
| | | | task5 | $1_{\pm1}$ | $3_{\pm1}$ | $3_{\pm1}$ | $4_{\pm2}$ | $\mathbf{36}_{\pm5}$ | $33_{\pm7}$ |
| | | | overall | $5_{\pm1}$ | $8_{\pm1}$ | $6_{\pm1}$ | $6_{\pm3}$ | $\mathbf{48}_{\pm2}$ | $37_{\pm2}$ |
| | stitch | visual-antmaze-medium-stitch-v0 | task1 | $\mathbf{80}_{\pm4}$ | $0_{\pm1}$ | $0_{\pm0}$ | $0_{\pm0}$ | $33_{\pm4}$ | $75_{\pm8}$ |
| | | | task2 | $\mathbf{90}_{\pm4}$ | $1_{\pm2}$ | $0_{\pm0}$ | $0_{\pm0}$ | $69_{\pm5}$ | $85_{\pm7}$ |
| | | | task3 | $69_{\pm18}$ | $15_{\pm6}$ | $8_{\pm1}$ | $0_{\pm0}$ | $88_{\pm1}$ | $\mathbf{92}_{\pm1}$ |
| | | | task4 | $1_{\pm1}$ | $7_{\pm4}$ | $3_{\pm1}$ | $0_{\pm1}$ | $70_{\pm12}$ | $\mathbf{88}_{\pm4}$ |
| | | | task5 | $\mathbf{97}_{\pm1}$ | $6_{\pm3}$ | $1_{\pm1}$ | $0_{\pm0}$ | $85_{\pm5}$ | $93_{\pm1}$ |
| | | | overall | $67_{\pm4}$ | $6_{\pm2}$ | $2_{\pm0}$ | $0_{\pm0}$ | $69_{\pm2}$ | $\mathbf{87}_{\pm2}$ |
| | | visual-antmaze-large-stitch-v0 | task1 | $26_{\pm11}$ | $0_{\pm0}$ | $0_{\pm0}$ | $0_{\pm0}$ | $6_{\pm1}$ | $\mathbf{36}_{\pm5}$ |
| | | | task2 | $0_{\pm0}$ | $0_{\pm0}$ | $0_{\pm0}$ | $0_{\pm0}$ | $2_{\pm1}$ | $\mathbf{3}_{\pm2}$ |
| | | | task3 | $73_{\pm14}$ | $3_{\pm2}$ | $0_{\pm0}$ | $2_{\pm2}$ | $36_{\pm10}$ | $\mathbf{87}_{\pm6}$ |
| | | | task4 | $7_{\pm5}$ | $1_{\pm1}$ | $0_{\pm0}$ | $1_{\pm1}$ | $\mathbf{8}_{\pm1}$ | $7_{\pm4}$ |
| | | | task5 | $\mathbf{11}_{\pm5}$ | $0_{\pm0}$ | $0_{\pm0}$ | $0_{\pm0}$ | $5_{\pm2}$ | $6_{\pm1}$ |
| | | | overall | $24_{\pm3}$ | $1_{\pm1}$ | $0_{\pm0}$ | $1_{\pm1}$ | $11_{\pm3}$ | $\mathbf{28}_{\pm2}$ |
| | | visual-antmaze-giant-stitch-v0 | task1 | $\mathbf{0}_{\pm0}$ | $\mathbf{0}_{\pm0}$ | $\mathbf{0}_{\pm0}$ | $\mathbf{0}_{\pm0}$ | $\mathbf{0}_{\pm0}$ | $\mathbf{0}_{\pm0}$ |
| | | | task2 | $\mathbf{1}_{\pm2}$ | $0_{\pm0}$ | $0_{\pm0}$ | $0_{\pm0}$ | $0_{\pm0}$ | $\mathbf{1}_{\pm1}$ |
| | | | task3 | $\mathbf{0}_{\pm0}$ | $\mathbf{0}_{\pm0}$ | $\mathbf{0}_{\pm0}$ | $\mathbf{0}_{\pm0}$ | $\mathbf{0}_{\pm0}$ | $\mathbf{0}_{\pm0}$ |
| | | | task4 | $\mathbf{0}_{\pm0}$ | $\mathbf{0}_{\pm0}$ | $\mathbf{0}_{\pm0}$ | $\mathbf{0}_{\pm0}$ | $\mathbf{0}_{\pm0}$ | $\mathbf{0}_{\pm0}$ |
| | | | task5 | $\mathbf{0}_{\pm0}$ | $\mathbf{0}_{\pm0}$ | $\mathbf{0}_{\pm0}$ | $\mathbf{0}_{\pm0}$ | $0_{\pm1}$ | $\mathbf{0}_{\pm0}$ |
| | | | overall | $\mathbf{0}_{\pm0}$ | $\mathbf{0}_{\pm0}$ | $\mathbf{0}_{\pm0}$ | $\mathbf{0}_{\pm0}$ | $\mathbf{0}_{\pm0}$ | $\mathbf{0}_{\pm0}$ |
| | | visual-antmaze-teleport-stitch-v0 | task1 | $\mathbf{37}_{\pm4}$ | $2_{\pm2}$ | $1_{\pm1}$ | $0_{\pm0}$ | $20_{\pm5}$ | $\mathbf{36}_{\pm5}$ |
| | | | task2 | $36_{\pm3}$ | $2_{\pm1}$ | $1_{\pm1}$ | $1_{\pm1}$ | $\mathbf{40}_{\pm9}$ | $38_{\pm3}$ |
| | | | task3 | $17_{\pm6}$ | $2_{\pm1}$ | $2_{\pm1}$ | $3_{\pm4}$ | $32_{\pm9}$ | $\mathbf{36}_{\pm5}$ |
| | | | task4 | $39_{\pm9}$ | $1_{\pm1}$ | $0_{\pm0}$ | $2_{\pm3}$ | $\mathbf{45}_{\pm7}$ | $37_{\pm6}$ |
| | | | task5 | $29_{\pm1}$ | $1_{\pm1}$ | $1_{\pm1}$ | $1_{\pm1}$ | $22_{\pm9}$ | $\mathbf{38}_{\pm5}$ |
| | | | overall | $32_{\pm3}$ | $1_{\pm1}$ | $1_{\pm0}$ | $1_{\pm2}$ | $32_{\pm6}$ | $\mathbf{37}_{\pm4}$ |
| | explore | visual-antmaze-medium-explore-v0 | task1 | $\mathbf{0}_{\pm0}$ | $\mathbf{0}_{\pm0}$ | $\mathbf{0}_{\pm0}$ | $\mathbf{0}_{\pm0}$ | $\mathbf{0}_{\pm0}$ | $\mathbf{0}_{\pm0}$ |
| | | | task2 | $\mathbf{0}_{\pm0}$ | $\mathbf{0}_{\pm0}$ | $\mathbf{0}_{\pm0}$ | $\mathbf{0}_{\pm0}$ | $\mathbf{0}_{\pm0}$ | $\mathbf{0}_{\pm0}$ |
| | | | task3 | $\mathbf{0}_{\pm0}$ | $\mathbf{0}_{\pm0}$ | $\mathbf{0}_{\pm0}$ | $\mathbf{0}_{\pm0}$ | $0_{\pm0}$ | $\mathbf{1}_{\pm2}$ |
| | | | task4 | $\mathbf{0}_{\pm0}$ | $\mathbf{0}_{\pm0}$ | $\mathbf{0}_{\pm0}$ | $\mathbf{0}_{\pm0}$ | $\mathbf{0}_{\pm0}$ | $\mathbf{0}_{\pm0}$ |
| | | | task5 | $\mathbf{0}_{\pm0}$ | $\mathbf{0}_{\pm0}$ | $\mathbf{0}_{\pm0}$ | $\mathbf{0}_{\pm0}$ | $\mathbf{0}_{\pm0}$ | $\mathbf{0}_{\pm0}$ |
| | | | overall | $\mathbf{0}_{\pm0}$ | $\mathbf{0}_{\pm0}$ | $\mathbf{0}_{\pm0}$ | $\mathbf{0}_{\pm0}$ | $\mathbf{0}_{\pm0}$ | $\mathbf{0}_{\pm0}$ |
| | | visual-antmaze-large-explore-v0 | task1 | $\mathbf{0}_{\pm0}$ | $\mathbf{0}_{\pm0}$ | $\mathbf{0}_{\pm0}$ | $\mathbf{0}_{\pm0}$ | $\mathbf{0}_{\pm0}$ | $\mathbf{0}_{\pm0}$ |
| | | | task2 | $\mathbf{0}_{\pm0}$ | $\mathbf{0}_{\pm0}$ | $\mathbf{0}_{\pm0}$ | $\mathbf{0}_{\pm0}$ | $\mathbf{0}_{\pm0}$ | $\mathbf{0}_{\pm0}$ |
| | | | task3 | $\mathbf{0}_{\pm0}$ | $\mathbf{0}_{\pm0}$ | $\mathbf{0}_{\pm0}$ | $\mathbf{0}_{\pm0}$ | $\mathbf{0}_{\pm0}$ | $\mathbf{0}_{\pm0}$ |
| | | | task4 | $\mathbf{0}_{\pm0}$ | $\mathbf{0}_{\pm0}$ | $\mathbf{0}_{\pm0}$ | $\mathbf{0}_{\pm0}$ | $\mathbf{0}_{\pm0}$ | $\mathbf{0}_{\pm0}$ |
| | | | task5 | $\mathbf{0}_{\pm0}$ | $\mathbf{0}_{\pm0}$ | $\mathbf{0}_{\pm0}$ | $\mathbf{0}_{\pm0}$ | $\mathbf{0}_{\pm0}$ | $\mathbf{0}_{\pm0}$ |
| | | | overall | $\mathbf{0}_{\pm0}$ | $\mathbf{0}_{\pm0}$ | $\mathbf{0}_{\pm0}$ | $\mathbf{0}_{\pm0}$ | $\mathbf{0}_{\pm0}$ | $\mathbf{0}_{\pm0}$ |
| | | visual-antmaze-teleport-explore-v0 | task1 | $\mathbf{0}_{\pm0}$ | $\mathbf{0}_{\pm0}$ | $\mathbf{0}_{\pm0}$ | $\mathbf{0}_{\pm0}$ | $\mathbf{0}_{\pm0}$ | $\mathbf{0}_{\pm0}$ |
| | | | task2 | $\mathbf{0}_{\pm0}$ | $\mathbf{0}_{\pm0}$ | $\mathbf{0}_{\pm0}$ | $\mathbf{0}_{\pm0}$ | $\mathbf{0}_{\pm0}$ | $\mathbf{0}_{\pm0}$ |
| | | | task3 | $0_{\pm0}$ | $0_{\pm1}$ | $0_{\pm1}$ | $0_{\pm0}$ | $3_{\pm1}$ | $\mathbf{38}_{\pm8}$ |
| | | | task4 | $0_{\pm0}$ | $0_{\pm0}$ | $0_{\pm0}$ | $0_{\pm0}$ | $0_{\pm0}$ | $\mathbf{28}_{\pm13}$ |
| | | | task5 | $0_{\pm0}$ | $0_{\pm0}$ | $0_{\pm0}$ | $0_{\pm0}$ | $2_{\pm2}$ | $\mathbf{27}_{\pm18}$ |
| | | | overall | $0_{\pm0}$ | $0_{\pm0}$ | $0_{\pm0}$ | $0_{\pm0}$ | $1_{\pm0}$ | $\mathbf{19}_{\pm8}$ |

Table 17: **Full results on Visual HumanoidMaze.**

| Environment Type | Dataset Type | Dataset | Task | GCBC | GCIVL | GCIQL | QRL | CRL | HIQL |
|---|---|---|---|---|---|---|---|---|---|
| visual-humanoidmaze | navigate | visual-humanoidmaze-medium-navigate-v0 | task1 | $\mathbf{0}_{\pm 0}$ | $\mathbf{0}_{\pm 0}$ | $\mathbf{0}_{\pm 0}$ | $\mathbf{0}_{\pm 0}$ | $\mathbf{0}_{\pm 0}$ | $\mathbf{0}_{\pm 0}$ |
| | | | task2 | $\mathbf{0}_{\pm 0}$ | $\mathbf{0}_{\pm 0}$ | $\mathbf{0}_{\pm 0}$ | $\mathbf{0}_{\pm 0}$ | $\mathbf{0}_{\pm 0}$ | $\mathbf{0}_{\pm 0}$ |
| | | | task3 | $0_{\pm 0}$ | $0_{\pm 0}$ | $0_{\pm 0}$ | $0_{\pm 0}$ | $\mathbf{2}_{\pm 1}$ | $0_{\pm 1}$ |
| | | | task4 | $\mathbf{0}_{\pm 0}$ | $\mathbf{0}_{\pm 0}$ | $\mathbf{0}_{\pm 0}$ | $\mathbf{0}_{\pm 0}$ | $\mathbf{0}_{\pm 0}$ | $\mathbf{0}_{\pm 0}$ |
| | | | task5 | $0_{\pm 0}$ | $0_{\pm 0}$ | $0_{\pm 0}$ | $0_{\pm 0}$ | $\mathbf{3}_{\pm 2}$ | $0_{\pm 1}$ |
| | | | overall | $0_{\pm 0}$ | $0_{\pm 0}$ | $0_{\pm 0}$ | $0_{\pm 0}$ | $\mathbf{1}_{\pm 0}$ | $0_{\pm 0}$ |
| | | visual-humanoidmaze-large-navigate-v0 | task1 | $\mathbf{0}_{\pm 0}$ | $\mathbf{0}_{\pm 0}$ | $\mathbf{0}_{\pm 0}$ | $\mathbf{0}_{\pm 0}$ | $\mathbf{0}_{\pm 0}$ | $\mathbf{0}_{\pm 0}$ |
| | | | task2 | $\mathbf{0}_{\pm 0}$ | $\mathbf{0}_{\pm 0}$ | $\mathbf{0}_{\pm 0}$ | $\mathbf{0}_{\pm 0}$ | $\mathbf{0}_{\pm 0}$ | $\mathbf{0}_{\pm 0}$ |
| | | | task3 | $\mathbf{0}_{\pm 0}$ | $\mathbf{0}_{\pm 0}$ | $\mathbf{0}_{\pm 0}$ | $\mathbf{0}_{\pm 0}$ | $\mathbf{0}_{\pm 0}$ | $\mathbf{0}_{\pm 0}$ |
| | | | task4 | $\mathbf{0}_{\pm 0}$ | $\mathbf{0}_{\pm 0}$ | $\mathbf{0}_{\pm 0}$ | $\mathbf{0}_{\pm 0}$ | $\mathbf{0}_{\pm 0}$ | $\mathbf{0}_{\pm 0}$ |
| | | | task5 | $\mathbf{0}_{\pm 0}$ | $\mathbf{0}_{\pm 0}$ | $\mathbf{0}_{\pm 0}$ | $\mathbf{0}_{\pm 0}$ | $\mathbf{0}_{\pm 0}$ | $\mathbf{0}_{\pm 0}$ |
| | | | overall | $\mathbf{0}_{\pm 0}$ | $\mathbf{0}_{\pm 0}$ | $\mathbf{0}_{\pm 0}$ | $\mathbf{0}_{\pm 0}$ | $\mathbf{0}_{\pm 0}$ | $\mathbf{0}_{\pm 0}$ |
| | | visual-humanoidmaze-giant-navigate-v0 | task1 | $\mathbf{0}_{\pm 0}$ | $\mathbf{0}_{\pm 0}$ | $\mathbf{0}_{\pm 0}$ | $\mathbf{0}_{\pm 0}$ | $\mathbf{0}_{\pm 0}$ | $\mathbf{0}_{\pm 0}$ |
| | | | task2 | $\mathbf{0}_{\pm 0}$ | $\mathbf{0}_{\pm 0}$ | $\mathbf{0}_{\pm 0}$ | $\mathbf{0}_{\pm 0}$ | $\mathbf{0}_{\pm 0}$ | $\mathbf{0}_{\pm 0}$ |
| | | | task3 | $\mathbf{0}_{\pm 0}$ | $\mathbf{0}_{\pm 0}$ | $\mathbf{0}_{\pm 0}$ | $\mathbf{0}_{\pm 0}$ | $\mathbf{0}_{\pm 0}$ | $\mathbf{0}_{\pm 0}$ |
| | | | task4 | $\mathbf{0}_{\pm 0}$ | $\mathbf{0}_{\pm 0}$ | $\mathbf{0}_{\pm 0}$ | $\mathbf{0}_{\pm 0}$ | $\mathbf{0}_{\pm 0}$ | $\mathbf{0}_{\pm 0}$ |
| | | | task5 | $\mathbf{0}_{\pm 0}$ | $\mathbf{0}_{\pm 0}$ | $\mathbf{0}_{\pm 0}$ | $\mathbf{0}_{\pm 0}$ | $\mathbf{0}_{\pm 0}$ | $\mathbf{0}_{\pm 0}$ |
| | | | overall | $\mathbf{0}_{\pm 0}$ | $\mathbf{0}_{\pm 0}$ | $\mathbf{0}_{\pm 0}$ | $\mathbf{0}_{\pm 0}$ | $\mathbf{0}_{\pm 0}$ | $\mathbf{0}_{\pm 0}$ |
| | stitch | visual-humanoidmaze-medium-stitch-v0 | task1 | $\mathbf{0}_{\pm 0}$ | $\mathbf{0}_{\pm 0}$ | $\mathbf{0}_{\pm 0}$ | $\mathbf{0}_{\pm 0}$ | $\mathbf{0}_{\pm 0}$ | $\mathbf{0}_{\pm 0}$ |
| | | | task2 | $\mathbf{0}_{\pm 0}$ | $\mathbf{0}_{\pm 0}$ | $\mathbf{0}_{\pm 0}$ | $\mathbf{0}_{\pm 0}$ | $\mathbf{0}_{\pm 0}$ | $\mathbf{0}_{\pm 0}$ |
| | | | task3 | $\mathbf{3}_{\pm 1}$ | $0_{\pm 0}$ | $0_{\pm 0}$ | $0_{\pm 0}$ | $\mathbf{3}_{\pm 2}$ | $1_{\pm 2}$ |
| | | | task4 | $\mathbf{0}_{\pm 0}$ | $\mathbf{0}_{\pm 0}$ | $\mathbf{0}_{\pm 0}$ | $\mathbf{0}_{\pm 0}$ | $\mathbf{0}_{\pm 0}$ | $\mathbf{0}_{\pm 0}$ |
| | | | task5 | $\mathbf{1}_{\pm 1}$ | $0_{\pm 0}$ | $0_{\pm 0}$ | $0_{\pm 0}$ | $0_{\pm 0}$ | $0_{\pm 0}$ |
| | | | overall | $\mathbf{1}_{\pm 0}$ | $0_{\pm 0}$ | $0_{\pm 0}$ | $0_{\pm 0}$ | $\mathbf{1}_{\pm 0}$ | $0_{\pm 0}$ |
| | | visual-humanoidmaze-large-stitch-v0 | task1 | $\mathbf{0}_{\pm 0}$ | $\mathbf{0}_{\pm 0}$ | $\mathbf{0}_{\pm 0}$ | $\mathbf{0}_{\pm 0}$ | $\mathbf{0}_{\pm 0}$ | $\mathbf{0}_{\pm 0}$ |
| | | | task2 | $\mathbf{0}_{\pm 0}$ | $\mathbf{0}_{\pm 0}$ | $\mathbf{0}_{\pm 0}$ | $\mathbf{0}_{\pm 0}$ | $\mathbf{0}_{\pm 0}$ | $\mathbf{0}_{\pm 0}$ |
| | | | task3 | $\mathbf{0}_{\pm 0}$ | $\mathbf{0}_{\pm 0}$ | $\mathbf{0}_{\pm 0}$ | $\mathbf{0}_{\pm 0}$ | $\mathbf{0}_{\pm 0}$ | $\mathbf{0}_{\pm 0}$ |
| | | | task4 | $\mathbf{0}_{\pm 0}$ | $\mathbf{0}_{\pm 0}$ | $\mathbf{0}_{\pm 0}$ | $\mathbf{0}_{\pm 0}$ | $\mathbf{0}_{\pm 0}$ | $\mathbf{0}_{\pm 0}$ |
| | | | task5 | $\mathbf{0}_{\pm 0}$ | $\mathbf{0}_{\pm 0}$ | $\mathbf{0}_{\pm 0}$ | $\mathbf{0}_{\pm 0}$ | $\mathbf{0}_{\pm 0}$ | $\mathbf{0}_{\pm 0}$ |
| | | | overall | $\mathbf{0}_{\pm 0}$ | $\mathbf{0}_{\pm 0}$ | $\mathbf{0}_{\pm 0}$ | $\mathbf{0}_{\pm 0}$ | $\mathbf{0}_{\pm 0}$ | $\mathbf{0}_{\pm 0}$ |
| | | visual-humanoidmaze-giant-stitch-v0 | task1 | $\mathbf{0}_{\pm 0}$ | $\mathbf{0}_{\pm 0}$ | $\mathbf{0}_{\pm 0}$ | $\mathbf{0}_{\pm 0}$ | $\mathbf{0}_{\pm 0}$ | $\mathbf{0}_{\pm 0}$ |
| | | | task2 | $\mathbf{0}_{\pm 0}$ | $\mathbf{0}_{\pm 0}$ | $\mathbf{0}_{\pm 0}$ | $\mathbf{0}_{\pm 0}$ | $\mathbf{0}_{\pm 0}$ | $\mathbf{0}_{\pm 0}$ |
| | | | task3 | $\mathbf{0}_{\pm 0}$ | $\mathbf{0}_{\pm 0}$ | $\mathbf{0}_{\pm 0}$ | $\mathbf{0}_{\pm 0}$ | $\mathbf{0}_{\pm 0}$ | $\mathbf{0}_{\pm 0}$ |
| | | | task4 | $\mathbf{0}_{\pm 0}$ | $\mathbf{0}_{\pm 0}$ | $\mathbf{0}_{\pm 0}$ | $\mathbf{0}_{\pm 0}$ | $\mathbf{0}_{\pm 0}$ | $\mathbf{0}_{\pm 0}$ |
| | | | task5 | $\mathbf{0}_{\pm 0}$ | $\mathbf{0}_{\pm 0}$ | $\mathbf{0}_{\pm 0}$ | $\mathbf{0}_{\pm 0}$ | $\mathbf{0}_{\pm 0}$ | $\mathbf{0}_{\pm 0}$ |
| | | | overall | $\mathbf{0}_{\pm 0}$ | $\mathbf{0}_{\pm 0}$ | $\mathbf{0}_{\pm 0}$ | $\mathbf{0}_{\pm 0}$ | $\mathbf{0}_{\pm 0}$ | $\mathbf{0}_{\pm 0}$ |

Table 18: **Full results on Cube.**

| Environment Type | Dataset Type | Dataset | Task | GCBC | GCIVL | GCIQL | QRL | CRL | HIQL |
|---|---|---|---|---|---|---|---|---|---|
| cube | play | cube-single-play-v0 | task1 | $7_{\pm 3}$ | $57_{\pm 6}$ | $\mathbf{71}_{\pm 9}$ | $6_{\pm 2}$ | $20_{\pm 6}$ | $15_{\pm 5}$ |
| | | | task2 | $5_{\pm 2}$ | $51_{\pm 6}$ | $\mathbf{71}_{\pm 6}$ | $5_{\pm 2}$ | $20_{\pm 4}$ | $16_{\pm 5}$ |
| | | | task3 | $7_{\pm 3}$ | $55_{\pm 6}$ | $\mathbf{70}_{\pm 6}$ | $4_{\pm 1}$ | $21_{\pm 6}$ | $16_{\pm 3}$ |
| | | | task4 | $4_{\pm 2}$ | $50_{\pm 4}$ | $\mathbf{61}_{\pm 8}$ | $4_{\pm 2}$ | $16_{\pm 3}$ | $14_{\pm 5}$ |
| | | | task5 | $4_{\pm 2}$ | $52_{\pm 6}$ | $\mathbf{67}_{\pm 7}$ | $4_{\pm 3}$ | $15_{\pm 3}$ | $13_{\pm 4}$ |
| | | | overall | $6_{\pm 2}$ | $53_{\pm 4}$ | $\mathbf{68}_{\pm 6}$ | $5_{\pm 1}$ | $19_{\pm 2}$ | $15_{\pm 3}$ |
| | | cube-double-play-v0 | task1 | $6_{\pm 3}$ | $58_{\pm 5}$ | $\mathbf{74}_{\pm 8}$ | $6_{\pm 3}$ | $30_{\pm 7}$ | $22_{\pm 6}$ |
| | | | task2 | $0_{\pm 0}$ | $51_{\pm 6}$ | $\mathbf{55}_{\pm 11}$ | $0_{\pm 0}$ | $9_{\pm 2}$ | $4_{\pm 3}$ |
| | | | task3 | $0_{\pm 0}$ | $42_{\pm 7}$ | $\mathbf{45}_{\pm 7}$ | $0_{\pm 0}$ | $6_{\pm 1}$ | $3_{\pm 2}$ |
| | | | task4 | $0_{\pm 0}$ | $\mathbf{7}_{\pm 2}$ | $4_{\pm 3}$ | $0_{\pm 0}$ | $0_{\pm 0}$ | $1_{\pm 1}$ |
| | | | task5 | $0_{\pm 0}$ | $21_{\pm 1}$ | $\mathbf{23}_{\pm 6}$ | $0_{\pm 0}$ | $3_{\pm 1}$ | $2_{\pm 1}$ |
| | | | overall | $1_{\pm 1}$ | $36_{\pm 3}$ | $\mathbf{40}_{\pm 5}$ | $1_{\pm 0}$ | $10_{\pm 2}$ | $6_{\pm 2}$ |
| | | cube-triple-play-v0 | task1 | $5_{\pm 4}$ | $3_{\pm 1}$ | $13_{\pm 3}$ | $1_{\pm 1}$ | $\mathbf{19}_{\pm 5}$ | $12_{\pm 6}$ |
| | | | task2 | $0_{\pm 0}$ | $0_{\pm 0}$ | $0_{\pm 0}$ | $0_{\pm 0}$ | $\mathbf{2}_{\pm 1}$ | $0_{\pm 0}$ |
| | | | task3 | $0_{\pm 0}$ | $0_{\pm 0}$ | $0_{\pm 0}$ | $0_{\pm 0}$ | $\mathbf{1}_{\pm 1}$ | $0_{\pm 0}$ |
| | | | task4 | $\mathbf{0}_{\pm 0}$ | $\mathbf{0}_{\pm 0}$ | $\mathbf{0}_{\pm 0}$ | $\mathbf{0}_{\pm 0}$ | $\mathbf{0}_{\pm 0}$ | $\mathbf{0}_{\pm 0}$ |
| | | | task5 | $\mathbf{0}_{\pm 0}$ | $\mathbf{0}_{\pm 0}$ | $\mathbf{0}_{\pm 0}$ | $\mathbf{0}_{\pm 0}$ | $\mathbf{0}_{\pm 0}$ | $\mathbf{0}_{\pm 0}$ |
| | | | overall | $1_{\pm 1}$ | $1_{\pm 0}$ | $3_{\pm 1}$ | $0_{\pm 0}$ | $\mathbf{4}_{\pm 1}$ | $3_{\pm 1}$ |
| | | cube-quadruple-play-v0 | task1 | $\mathbf{0}_{\pm 0}$ | $\mathbf{0}_{\pm 0}$ | $\mathbf{0}_{\pm 0}$ | $\mathbf{0}_{\pm 0}$ | $\mathbf{0}_{\pm 0}$ | $\mathbf{0}_{\pm 0}$ |
| | | | task2 | $\mathbf{0}_{\pm 0}$ | $\mathbf{0}_{\pm 0}$ | $\mathbf{0}_{\pm 0}$ | $\mathbf{0}_{\pm 0}$ | $\mathbf{0}_{\pm 0}$ | $\mathbf{0}_{\pm 0}$ |
| | | | task3 | $\mathbf{0}_{\pm 0}$ | $\mathbf{0}_{\pm 0}$ | $\mathbf{0}_{\pm 0}$ | $\mathbf{0}_{\pm 0}$ | $\mathbf{0}_{\pm 0}$ | $\mathbf{0}_{\pm 0}$ |
| | | | task4 | $\mathbf{0}_{\pm 0}$ | $\mathbf{0}_{\pm 0}$ | $\mathbf{0}_{\pm 0}$ | $\mathbf{0}_{\pm 0}$ | $\mathbf{0}_{\pm 0}$ | $\mathbf{0}_{\pm 0}$ |
| | | | task5 | $\mathbf{0}_{\pm 0}$ | $\mathbf{0}_{\pm 0}$ | $\mathbf{0}_{\pm 0}$ | $\mathbf{0}_{\pm 0}$ | $\mathbf{0}_{\pm 0}$ | $\mathbf{0}_{\pm 0}$ |
| | | | overall | $\mathbf{0}_{\pm 0}$ | $\mathbf{0}_{\pm 0}$ | $\mathbf{0}_{\pm 0}$ | $\mathbf{0}_{\pm 0}$ | $\mathbf{0}_{\pm 0}$ | $\mathbf{0}_{\pm 0}$ |
| | noisy | cube-single-noisy-v0 | task1 | $5_{\pm 3}$ | $71_{\pm 12}$ | $\mathbf{100}_{\pm 1}$ | $17_{\pm 12}$ | $39_{\pm 4}$ | $48_{\pm 6}$ |
| | | | task2 | $7_{\pm 5}$ | $70_{\pm 11}$ | $\mathbf{100}_{\pm 0}$ | $23_{\pm 8}$ | $39_{\pm 7}$ | $39_{\pm 7}$ |
| | | | task3 | $1_{\pm 1}$ | $67_{\pm 10}$ | $\mathbf{99}_{\pm 1}$ | $4_{\pm 3}$ | $36_{\pm 7}$ | $41_{\pm 7}$ |
| | | | task4 | $16_{\pm 5}$ | $76_{\pm 10}$ | $\mathbf{98}_{\pm 2}$ | $47_{\pm 14}$ | $36_{\pm 4}$ | $36_{\pm 6}$ |
| | | | task5 | $12_{\pm 5}$ | $70_{\pm 12}$ | $\mathbf{100}_{\pm 0}$ | $37_{\pm 9}$ | $42_{\pm 5}$ | $44_{\pm 9}$ |
| | | | overall | $8_{\pm 3}$ | $71_{\pm 9}$ | $\mathbf{99}_{\pm 1}$ | $25_{\pm 6}$ | $38_{\pm 2}$ | $41_{\pm 6}$ |
| | | cube-double-noisy-v0 | task1 | $7_{\pm 3}$ | $53_{\pm 11}$ | $\mathbf{64}_{\pm 8}$ | $16_{\pm 5}$ | $9_{\pm 5}$ | $10_{\pm 3}$ |
| | | | task2 | $0_{\pm 0}$ | $10_{\pm 4}$ | $\mathbf{16}_{\pm 4}$ | $0_{\pm 0}$ | $0_{\pm 0}$ | $1_{\pm 0}$ |
| | | | task3 | $0_{\pm 0}$ | $1_{\pm 1}$ | $\mathbf{6}_{\pm 4}$ | $0_{\pm 0}$ | $0_{\pm 0}$ | $1_{\pm 1}$ |
| | | | task4 | $0_{\pm 0}$ | $4_{\pm 2}$ | $\mathbf{11}_{\pm 3}$ | $0_{\pm 1}$ | $0_{\pm 0}$ | $0_{\pm 1}$ |
| | | | task5 | $0_{\pm 0}$ | $4_{\pm 2}$ | $\mathbf{20}_{\pm 4}$ | $0_{\pm 0}$ | $0_{\pm 0}$ | $0_{\pm 1}$ |
| | | | overall | $1_{\pm 1}$ | $14_{\pm 3}$ | $\mathbf{23}_{\pm 3}$ | $3_{\pm 1}$ | $2_{\pm 1}$ | $2_{\pm 1}$ |
| | | cube-triple-noisy-v0 | task1 | $6_{\pm 3}$ | $\mathbf{44}_{\pm 7}$ | $8_{\pm 2}$ | $5_{\pm 2}$ | $13_{\pm 6}$ | $8_{\pm 3}$ |
| | | | task2 | $0_{\pm 0}$ | $0_{\pm 0}$ | $\mathbf{1}_{\pm 1}$ | $0_{\pm 0}$ | $0_{\pm 0}$ | $0_{\pm 0}$ |
| | | | task3 | $0_{\pm 0}$ | $\mathbf{2}_{\pm 1}$ | $0_{\pm 0}$ | $0_{\pm 0}$ | $0_{\pm 0}$ | $0_{\pm 0}$ |
| | | | task4 | $\mathbf{0}_{\pm 0}$ | $\mathbf{0}_{\pm 0}$ | $\mathbf{0}_{\pm 0}$ | $\mathbf{0}_{\pm 0}$ | $\mathbf{0}_{\pm 0}$ | $\mathbf{0}_{\pm 0}$ |
| | | | task5 | $\mathbf{0}_{\pm 0}$ | $\mathbf{0}_{\pm 0}$ | $\mathbf{0}_{\pm 0}$ | $\mathbf{0}_{\pm 0}$ | $\mathbf{0}_{\pm 0}$ | $\mathbf{0}_{\pm 0}$ |
| | | | overall | $1_{\pm 1}$ | $\mathbf{9}_{\pm 1}$ | $2_{\pm 1}$ | $1_{\pm 0}$ | $3_{\pm 1}$ | $2_{\pm 1}$ |
| | | cube-quadruple-noisy-v0 | task1 | $\mathbf{0}_{\pm 0}$ | $\mathbf{0}_{\pm 0}$ | $\mathbf{0}_{\pm 0}$ | $\mathbf{0}_{\pm 0}$ | $\mathbf{0}_{\pm 0}$ | $\mathbf{0}_{\pm 0}$ |
| | | | task2 | $\mathbf{0}_{\pm 0}$ | $\mathbf{0}_{\pm 0}$ | $\mathbf{0}_{\pm 0}$ | $\mathbf{0}_{\pm 0}$ | $\mathbf{0}_{\pm 0}$ | $\mathbf{0}_{\pm 0}$ |
| | | | task3 | $\mathbf{0}_{\pm 0}$ | $\mathbf{0}_{\pm 0}$ | $\mathbf{0}_{\pm 0}$ | $\mathbf{0}_{\pm 0}$ | $\mathbf{0}_{\pm 0}$ | $\mathbf{0}_{\pm 0}$ |
| | | | task4 | $\mathbf{0}_{\pm 0}$ | $\mathbf{0}_{\pm 0}$ | $\mathbf{0}_{\pm 0}$ | $\mathbf{0}_{\pm 0}$ | $\mathbf{0}_{\pm 0}$ | $\mathbf{0}_{\pm 0}$ |
| | | | task5 | $\mathbf{0}_{\pm 0}$ | $\mathbf{0}_{\pm 0}$ | $\mathbf{0}_{\pm 0}$ | $\mathbf{0}_{\pm 0}$ | $\mathbf{0}_{\pm 0}$ | $\mathbf{0}_{\pm 0}$ |
| | | | overall | $\mathbf{0}_{\pm 0}$ | $\mathbf{0}_{\pm 0}$ | $\mathbf{0}_{\pm 0}$ | $\mathbf{0}_{\pm 0}$ | $\mathbf{0}_{\pm 0}$ | $\mathbf{0}_{\pm 0}$ |

Table 19: **Full results on Scene.**

| Environment Type | Dataset Type | Dataset | Task | GCBC | GCIVL | GCIQL | QRL | CRL | HIQL |
|---|---|---|---|---|---|---|---|---|---|
| scene | play | scene-play-v0 | task1 | $18_{\pm7}$ | $75_{\pm5}$ | $\mathbf{93}_{\pm4}$ | $19_{\pm4}$ | $49_{\pm7}$ | $40_{\pm4}$ |
| | | | task2 | $1_{\pm1}$ | $62_{\pm8}$ | $\mathbf{82}_{\pm8}$ | $1_{\pm1}$ | $12_{\pm4}$ | $40_{\pm9}$ |
| | | | task3 | $2_{\pm1}$ | $64_{\pm7}$ | $\mathbf{72}_{\pm10}$ | $1_{\pm1}$ | $26_{\pm8}$ | $36_{\pm5}$ |
| | | | task4 | $3_{\pm2}$ | $7_{\pm4}$ | $8_{\pm3}$ | $5_{\pm2}$ | $5_{\pm2}$ | $\mathbf{55}_{\pm5}$ |
| | | | task5 | $0_{\pm0}$ | $2_{\pm1}$ | $1_{\pm1}$ | $0_{\pm1}$ | $1_{\pm1}$ | $\mathbf{20}_{\pm5}$ |
| | | | overall | $5_{\pm1}$ | $42_{\pm4}$ | $\mathbf{51}_{\pm4}$ | $5_{\pm1}$ | $19_{\pm2}$ | $38_{\pm3}$ |
| | noisy | scene-noisy-v0 | task1 | $6_{\pm3}$ | $60_{\pm11}$ | $50_{\pm5}$ | $39_{\pm10}$ | $5_{\pm4}$ | $\mathbf{68}_{\pm5}$ |
| | | | task2 | $0_{\pm0}$ | $42_{\pm11}$ | $\mathbf{52}_{\pm13}$ | $2_{\pm1}$ | $0_{\pm0}$ | $29_{\pm6}$ |
| | | | task3 | $0_{\pm0}$ | $27_{\pm6}$ | $\mathbf{28}_{\pm5}$ | $1_{\pm1}$ | $0_{\pm1}$ | $17_{\pm6}$ |
| | | | task4 | $0_{\pm0}$ | $3_{\pm3}$ | $0_{\pm0}$ | $3_{\pm3}$ | $0_{\pm0}$ | $\mathbf{10}_{\pm8}$ |
| | | | task5 | $0_{\pm0}$ | $0_{\pm0}$ | $0_{\pm0}$ | $0_{\pm0}$ | $0_{\pm0}$ | $\mathbf{2}_{\pm2}$ |
| | | | overall | $1_{\pm1}$ | $\mathbf{26}_{\pm5}$ | $\mathbf{26}_{\pm2}$ | $9_{\pm2}$ | $1_{\pm1}$ | $25_{\pm4}$ |

Table 20: **Full results on Puzzle.**

| Environment Type | Dataset Type | Dataset | Task | GCBC | GCIVL | GCIQL | QRL | CRL | HIQL |
|---|---|---|---|---|---|---|---|---|---|
| puzzle | play | puzzle-3x3-play-v0 | task1 | $5_{\pm1}$ | $17_{\pm4}$ | $\mathbf{99}_{\pm2}$ | $3_{\pm2}$ | $11_{\pm3}$ | $29_{\pm4}$ |
| | | | task2 | $2_{\pm1}$ | $4_{\pm2}$ | $\mathbf{96}_{\pm3}$ | $0_{\pm0}$ | $2_{\pm1}$ | $11_{\pm3}$ |
| | | | task3 | $1_{\pm1}$ | $3_{\pm1}$ | $\mathbf{95}_{\pm1}$ | $0_{\pm0}$ | $1_{\pm1}$ | $7_{\pm3}$ |
| | | | task4 | $1_{\pm1}$ | $3_{\pm1}$ | $\mathbf{91}_{\pm3}$ | $0_{\pm0}$ | $2_{\pm1}$ | $5_{\pm1}$ |
| | | | task5 | $1_{\pm0}$ | $2_{\pm1}$ | $\mathbf{94}_{\pm2}$ | $0_{\pm0}$ | $2_{\pm1}$ | $8_{\pm3}$ |
| | | | overall | $2_{\pm0}$ | $6_{\pm1}$ | $\mathbf{95}_{\pm1}$ | $1_{\pm0}$ | $3_{\pm1}$ | $12_{\pm2}$ |
| | | puzzle-4x4-play-v0 | task1 | $0_{\pm0}$ | $17_{\pm4}$ | $\mathbf{42}_{\pm7}$ | $1_{\pm1}$ | $1_{\pm1}$ | $10_{\pm3}$ |
| | | | task2 | $0_{\pm0}$ | $\mathbf{13}_{\pm5}$ | $2_{\pm1}$ | $0_{\pm0}$ | $0_{\pm0}$ | $9_{\pm4}$ |
| | | | task3 | $0_{\pm0}$ | $12_{\pm3}$ | $\mathbf{40}_{\pm5}$ | $0_{\pm0}$ | $0_{\pm0}$ | $7_{\pm3}$ |
| | | | task4 | $0_{\pm0}$ | $11_{\pm3}$ | $\mathbf{23}_{\pm5}$ | $0_{\pm0}$ | $0_{\pm1}$ | $6_{\pm2}$ |
| | | | task5 | $0_{\pm0}$ | $10_{\pm4}$ | $\mathbf{23}_{\pm5}$ | $0_{\pm0}$ | $0_{\pm0}$ | $5_{\pm2}$ |
| | | | overall | $0_{\pm0}$ | $13_{\pm2}$ | $\mathbf{26}_{\pm3}$ | $0_{\pm0}$ | $0_{\pm0}$ | $7_{\pm2}$ |
| | | puzzle-4x5-play-v0 | task1 | $1_{\pm1}$ | $33_{\pm6}$ | $\mathbf{71}_{\pm5}$ | $0_{\pm1}$ | $6_{\pm2}$ | $17_{\pm5}$ |
| | | | task2 | $0_{\pm0}$ | $0_{\pm0}$ | $0_{\pm0}$ | $0_{\pm0}$ | $0_{\pm0}$ | $\mathbf{1}_{\pm0}$ |
| | | | task3 | $\mathbf{0}_{\pm0}$ | $\mathbf{0}_{\pm0}$ | $\mathbf{0}_{\pm0}$ | $\mathbf{0}_{\pm0}$ | $\mathbf{0}_{\pm0}$ | $\mathbf{0}_{\pm0}$ |
| | | | task4 | $\mathbf{0}_{\pm0}$ | $\mathbf{0}_{\pm0}$ | $\mathbf{0}_{\pm0}$ | $\mathbf{0}_{\pm0}$ | $\mathbf{0}_{\pm0}$ | $\mathbf{0}_{\pm0}$ |
| | | | task5 | $\mathbf{0}_{\pm0}$ | $\mathbf{0}_{\pm0}$ | $\mathbf{0}_{\pm0}$ | $\mathbf{0}_{\pm0}$ | $\mathbf{0}_{\pm0}$ | $\mathbf{0}_{\pm0}$ |
| | | | overall | $0_{\pm0}$ | $7_{\pm1}$ | $\mathbf{14}_{\pm1}$ | $0_{\pm0}$ | $1_{\pm0}$ | $4_{\pm1}$ |
| | | puzzle-4x6-play-v0 | task1 | $0_{\pm0}$ | $43_{\pm8}$ | $\mathbf{52}_{\pm6}$ | $0_{\pm0}$ | $12_{\pm5}$ | $12_{\pm5}$ |
| | | | task2 | $0_{\pm0}$ | $\mathbf{7}_{\pm4}$ | $5_{\pm3}$ | $0_{\pm0}$ | $\mathbf{7}_{\pm3}$ | $2_{\pm1}$ |
| | | | task3 | $\mathbf{0}_{\pm0}$ | $\mathbf{0}_{\pm0}$ | $\mathbf{0}_{\pm0}$ | $\mathbf{0}_{\pm0}$ | $\mathbf{0}_{\pm0}$ | $\mathbf{0}_{\pm0}$ |
| | | | task4 | $\mathbf{0}_{\pm0}$ | $\mathbf{0}_{\pm0}$ | $\mathbf{0}_{\pm0}$ | $\mathbf{0}_{\pm0}$ | $\mathbf{0}_{\pm0}$ | $\mathbf{0}_{\pm0}$ |
| | | | task5 | $\mathbf{0}_{\pm0}$ | $\mathbf{0}_{\pm0}$ | $\mathbf{0}_{\pm0}$ | $\mathbf{0}_{\pm0}$ | $\mathbf{0}_{\pm0}$ | $\mathbf{0}_{\pm0}$ |
| | | | overall | $0_{\pm0}$ | $10_{\pm2}$ | $\mathbf{12}_{\pm1}$ | $0_{\pm0}$ | $4_{\pm1}$ | $3_{\pm1}$ |
| | noisy | puzzle-3x3-noisy-v0 | task1 | $4_{\pm2}$ | $89_{\pm8}$ | $\mathbf{100}_{\pm0}$ | $1_{\pm1}$ | $76_{\pm8}$ | $67_{\pm10}$ |
| | | | task2 | $0_{\pm0}$ | $42_{\pm29}$ | $\mathbf{88}_{\pm9}$ | $0_{\pm0}$ | $26_{\pm9}$ | $54_{\pm11}$ |
| | | | task3 | $0_{\pm0}$ | $26_{\pm20}$ | $\mathbf{99}_{\pm1}$ | $0_{\pm0}$ | $15_{\pm9}$ | $43_{\pm12}$ |
| | | | task4 | $0_{\pm0}$ | $23_{\pm20}$ | $\mathbf{94}_{\pm3}$ | $0_{\pm0}$ | $12_{\pm9}$ | $41_{\pm13}$ |
| | | | task5 | $0_{\pm0}$ | $31_{\pm19}$ | $\mathbf{88}_{\pm5}$ | $0_{\pm0}$ | $18_{\pm6}$ | $47_{\pm15}$ |
| | | | overall | $1_{\pm0}$ | $42_{\pm19}$ | $\mathbf{94}_{\pm3}$ | $0_{\pm0}$ | $30_{\pm6}$ | $51_{\pm11}$ |
| | | puzzle-4x4-noisy-v0 | task1 | $0_{\pm0}$ | $\mathbf{51}_{\pm10}$ | $49_{\pm9}$ | $0_{\pm0}$ | $0_{\pm0}$ | $19_{\pm5}$ |
| | | | task2 | $0_{\pm0}$ | $0_{\pm0}$ | $0_{\pm0}$ | $0_{\pm0}$ | $0_{\pm0}$ | $\mathbf{16}_{\pm5}$ |
| | | | task3 | $0_{\pm0}$ | $34_{\pm4}$ | $\mathbf{61}_{\pm14}$ | $0_{\pm0}$ | $0_{\pm0}$ | $17_{\pm6}$ |
| | | | task4 | $0_{\pm0}$ | $9_{\pm3}$ | $\mathbf{23}_{\pm10}$ | $0_{\pm0}$ | $0_{\pm0}$ | $14_{\pm5}$ |
| | | | task5 | $0_{\pm0}$ | $8_{\pm4}$ | $\mathbf{14}_{\pm9}$ | $0_{\pm0}$ | $0_{\pm0}$ | $12_{\pm4}$ |
| | | | overall | $0_{\pm0}$ | $20_{\pm3}$ | $\mathbf{29}_{\pm7}$ | $0_{\pm0}$ | $0_{\pm0}$ | $16_{\pm4}$ |
| | | puzzle-4x5-noisy-v0 | task1 | $0_{\pm0}$ | $\mathbf{97}_{\pm1}$ | $\mathbf{97}_{\pm2}$ | $0_{\pm0}$ | $16_{\pm9}$ | $21_{\pm5}$ |
| | | | task2 | $0_{\pm0}$ | $0_{\pm0}$ | $0_{\pm0}$ | $0_{\pm0}$ | $0_{\pm0}$ | $\mathbf{1}_{\pm1}$ |
| | | | task3 | $0_{\pm0}$ | $0_{\pm0}$ | $0_{\pm0}$ | $0_{\pm0}$ | $0_{\pm0}$ | $\mathbf{1}_{\pm1}$ |
| | | | task4 | $\mathbf{0}_{\pm0}$ | $\mathbf{0}_{\pm0}$ | $\mathbf{0}_{\pm0}$ | $\mathbf{0}_{\pm0}$ | $\mathbf{0}_{\pm0}$ | $\mathbf{0}_{\pm0}$ |
| | | | task5 | $\mathbf{0}_{\pm0}$ | $\mathbf{0}_{\pm0}$ | $\mathbf{0}_{\pm0}$ | $\mathbf{0}_{\pm0}$ | $\mathbf{0}_{\pm0}$ | $\mathbf{0}_{\pm0}$ |
| | | | overall | $0_{\pm0}$ | $\mathbf{19}_{\pm0}$ | $\mathbf{19}_{\pm0}$ | $0_{\pm0}$ | $3_{\pm2}$ | $5_{\pm1}$ |
| | | puzzle-4x6-noisy-v0 | task1 | $0_{\pm0}$ | $80_{\pm8}$ | $\mathbf{86}_{\pm7}$ | $0_{\pm0}$ | $28_{\pm13}$ | $8_{\pm4}$ |
| | | | task2 | $0_{\pm0}$ | $\mathbf{3}_{\pm2}$ | $1_{\pm1}$ | $0_{\pm0}$ | $1_{\pm1}$ | $1_{\pm1}$ |
| | | | task3 | $\mathbf{0}_{\pm0}$ | $\mathbf{0}_{\pm0}$ | $\mathbf{0}_{\pm0}$ | $\mathbf{0}_{\pm0}$ | $\mathbf{0}_{\pm0}$ | $\mathbf{0}_{\pm0}$ |
| | | | task4 | $\mathbf{0}_{\pm0}$ | $\mathbf{0}_{\pm0}$ | $\mathbf{0}_{\pm0}$ | $\mathbf{0}_{\pm0}$ | $\mathbf{0}_{\pm0}$ | $\mathbf{0}_{\pm0}$ |
| | | | task5 | $\mathbf{0}_{\pm0}$ | $\mathbf{0}_{\pm0}$ | $\mathbf{0}_{\pm0}$ | $\mathbf{0}_{\pm0}$ | $\mathbf{0}_{\pm0}$ | $\mathbf{0}_{\pm0}$ |
| | | | overall | $0_{\pm0}$ | $17_{\pm2}$ | $\mathbf{18}_{\pm2}$ | $0_{\pm0}$ | $6_{\pm3}$ | $2_{\pm1}$ |

Table 21: **Full results on Visual Cube.**

| Environment Type | Dataset Type | Dataset | Task | GCBC | GCIVL | GCIQL | QRL | CRL | HIQL |
|---|---|---|---|---|---|---|---|---|---|
| visual-cube | play | visual-cube-single-play-v0 | task1 | $12_{\pm4}$ | $70_{\pm3}$ | $42_{\pm12}$ | $68_{\pm9}$ | $47_{\pm20}$ | $\mathbf{93}_{\pm1}$ |
| | | | task2 | $4_{\pm4}$ | $65_{\pm9}$ | $44_{\pm11}$ | $35_{\pm34}$ | $40_{\pm20}$ | $\mathbf{93}_{\pm3}$ |
| | | | task3 | $6_{\pm5}$ | $49_{\pm2}$ | $24_{\pm9}$ | $41_{\pm27}$ | $33_{\pm17}$ | $\mathbf{84}_{\pm5}$ |
| | | | task4 | $0_{\pm0}$ | $60_{\pm13}$ | $21_{\pm6}$ | $32_{\pm10}$ | $18_{\pm16}$ | $\mathbf{84}_{\pm5}$ |
| | | | task5 | $0_{\pm0}$ | $55_{\pm6}$ | $20_{\pm7}$ | $30_{\pm8}$ | $16_{\pm10}$ | $\mathbf{88}_{\pm2}$ |
| | | | overall | $5_{\pm1}$ | $60_{\pm5}$ | $30_{\pm5}$ | $41_{\pm15}$ | $31_{\pm15}$ | $\mathbf{89}_{\pm0}$ |
| | | visual-cube-double-play-v0 | task1 | $4_{\pm3}$ | $44_{\pm8}$ | $6_{\pm4}$ | $20_{\pm3}$ | $7_{\pm4}$ | $\mathbf{91}_{\pm1}$ |
| | | | task2 | $0_{\pm0}$ | $0_{\pm1}$ | $0_{\pm0}$ | $2_{\pm2}$ | $0_{\pm0}$ | $\mathbf{54}_{\pm5}$ |
| | | | task3 | $0_{\pm1}$ | $0_{\pm0}$ | $0_{\pm0}$ | $2_{\pm0}$ | $0_{\pm0}$ | $\mathbf{40}_{\pm6}$ |
| | | | task4 | $\mathbf{0}_{\pm0}$ | $\mathbf{0}_{\pm0}$ | $\mathbf{0}_{\pm0}$ | $\mathbf{0}_{\pm0}$ | $\mathbf{0}_{\pm0}$ | $\mathbf{0}_{\pm0}$ |
| | | | task5 | $0_{\pm0}$ | $4_{\pm2}$ | $0_{\pm0}$ | $0_{\pm0}$ | $0_{\pm0}$ | $\mathbf{11}_{\pm4}$ |
| | | | overall | $1_{\pm1}$ | $10_{\pm2}$ | $1_{\pm1}$ | $5_{\pm0}$ | $2_{\pm1}$ | $\mathbf{39}_{\pm2}$ |
| | | visual-cube-triple-play-v0 | task1 | $73_{\pm8}$ | $68_{\pm8}$ | $76_{\pm6}$ | $81_{\pm3}$ | $85_{\pm12}$ | $\mathbf{98}_{\pm1}$ |
| | | | task2 | $0_{\pm0}$ | $0_{\pm0}$ | $0_{\pm0}$ | $0_{\pm0}$ | $0_{\pm0}$ | $\mathbf{1}_{\pm1}$ |
| | | | task3 | $0_{\pm0}$ | $0_{\pm0}$ | $0_{\pm0}$ | $0_{\pm0}$ | $0_{\pm0}$ | $\mathbf{7}_{\pm3}$ |
| | | | task4 | $\mathbf{0}_{\pm0}$ | $\mathbf{0}_{\pm0}$ | $\mathbf{0}_{\pm0}$ | $\mathbf{0}_{\pm0}$ | $\mathbf{0}_{\pm0}$ | $\mathbf{0}_{\pm0}$ |
| | | | task5 | $\mathbf{0}_{\pm0}$ | $\mathbf{0}_{\pm0}$ | $\mathbf{0}_{\pm0}$ | $\mathbf{0}_{\pm0}$ | $\mathbf{0}_{\pm0}$ | $\mathbf{0}_{\pm0}$ |
| | | | overall | $15_{\pm2}$ | $14_{\pm2}$ | $15_{\pm1}$ | $16_{\pm1}$ | $17_{\pm2}$ | $\mathbf{21}_{\pm0}$ |
| | | visual-cube-quadruple-play-v0 | task1 | $42_{\pm4}$ | $1_{\pm2}$ | $36_{\pm7}$ | $23_{\pm5}$ | $20_{\pm4}$ | $\mathbf{66}_{\pm7}$ |
| | | | task2 | $\mathbf{0}_{\pm0}$ | $\mathbf{0}_{\pm0}$ | $\mathbf{0}_{\pm0}$ | $\mathbf{0}_{\pm0}$ | $\mathbf{0}_{\pm0}$ | $\mathbf{0}_{\pm0}$ |
| | | | task3 | $0_{\pm0}$ | $0_{\pm0}$ | $0_{\pm0}$ | $0_{\pm0}$ | $0_{\pm0}$ | $\mathbf{1}_{\pm1}$ |
| | | | task4 | $\mathbf{0}_{\pm0}$ | $\mathbf{0}_{\pm0}$ | $\mathbf{0}_{\pm0}$ | $\mathbf{0}_{\pm0}$ | $\mathbf{0}_{\pm0}$ | $\mathbf{0}_{\pm0}$ |
| | | | task5 | $0_{\pm0}$ | $0_{\pm0}$ | $0_{\pm0}$ | $0_{\pm0}$ | $0_{\pm0}$ | $\mathbf{1}_{\pm1}$ |
| | | | overall | $8_{\pm1}$ | $0_{\pm0}$ | $7_{\pm1}$ | $5_{\pm1}$ | $4_{\pm1}$ | $\mathbf{14}_{\pm1}$ |
| | noisy | visual-cube-single-noisy-v0 | task1 | $12_{\pm4}$ | $89_{\pm4}$ | $81_{\pm6}$ | $18_{\pm29}$ | $43_{\pm41}$ | $\mathbf{100}_{\pm1}$ |
| | | | task2 | $17_{\pm6}$ | $48_{\pm14}$ | $0_{\pm0}$ | $0_{\pm0}$ | $28_{\pm13}$ | $\mathbf{100}_{\pm1}$ |
| | | | task3 | $6_{\pm2}$ | $77_{\pm4}$ | $90_{\pm3}$ | $28_{\pm27}$ | $42_{\pm41}$ | $\mathbf{99}_{\pm1}$ |
| | | | task4 | $14_{\pm2}$ | $82_{\pm2}$ | $35_{\pm11}$ | $2_{\pm1}$ | $43_{\pm35}$ | $\mathbf{100}_{\pm1}$ |
| | | | task5 | $20_{\pm10}$ | $77_{\pm2}$ | $33_{\pm7}$ | $2_{\pm1}$ | $37_{\pm27}$ | $\mathbf{99}_{\pm1}$ |
| | | | overall | $14_{\pm3}$ | $75_{\pm3}$ | $48_{\pm3}$ | $10_{\pm5}$ | $39_{\pm30}$ | $\mathbf{99}_{\pm0}$ |
| | | visual-cube-double-noisy-v0 | task1 | $20_{\pm5}$ | $70_{\pm8}$ | $70_{\pm5}$ | $27_{\pm9}$ | $24_{\pm11}$ | $\mathbf{98}_{\pm2}$ |
| | | | task2 | $2_{\pm2}$ | $5_{\pm3}$ | $14_{\pm2}$ | $2_{\pm4}$ | $3_{\pm3}$ | $\mathbf{87}_{\pm9}$ |
| | | | task3 | $2_{\pm1}$ | $5_{\pm5}$ | $16_{\pm6}$ | $0_{\pm0}$ | $2_{\pm2}$ | $\mathbf{68}_{\pm10}$ |
| | | | task4 | $1_{\pm1}$ | $3_{\pm1}$ | $0_{\pm0}$ | $0_{\pm0}$ | $0_{\pm0}$ | $\mathbf{13}_{\pm9}$ |
| | | | task5 | $0_{\pm0}$ | $0_{\pm0}$ | $7_{\pm2}$ | $1_{\pm1}$ | $0_{\pm0}$ | $\mathbf{30}_{\pm4}$ |
| | | | overall | $5_{\pm1}$ | $17_{\pm4}$ | $22_{\pm2}$ | $6_{\pm2}$ | $6_{\pm3}$ | $\mathbf{59}_{\pm3}$ |
| | | visual-cube-triple-noisy-v0 | task1 | $80_{\pm7}$ | $90_{\pm5}$ | $62_{\pm6}$ | $44_{\pm21}$ | $78_{\pm7}$ | $\mathbf{99}_{\pm1}$ |
| | | | task2 | $0_{\pm1}$ | $0_{\pm0}$ | $0_{\pm0}$ | $0_{\pm0}$ | $0_{\pm0}$ | $\mathbf{2}_{\pm2}$ |
| | | | task3 | $0_{\pm1}$ | $0_{\pm0}$ | $0_{\pm0}$ | $0_{\pm0}$ | $0_{\pm0}$ | $\mathbf{14}_{\pm11}$ |
| | | | task4 | $\mathbf{0}_{\pm0}$ | $\mathbf{0}_{\pm0}$ | $\mathbf{0}_{\pm0}$ | $\mathbf{0}_{\pm0}$ | $\mathbf{0}_{\pm0}$ | $\mathbf{0}_{\pm0}$ |
| | | | task5 | $\mathbf{0}_{\pm1}$ | $\mathbf{0}_{\pm0}$ | $\mathbf{0}_{\pm0}$ | $\mathbf{0}_{\pm0}$ | $\mathbf{0}_{\pm0}$ | $\mathbf{0}_{\pm0}$ |
| | | | overall | $16_{\pm1}$ | $18_{\pm1}$ | $12_{\pm1}$ | $9_{\pm4}$ | $16_{\pm1}$ | $\mathbf{23}_{\pm2}$ |
| | | visual-cube-quadruple-noisy-v0 | task1 | $46_{\pm2}$ | $2_{\pm1}$ | $10_{\pm9}$ | $2_{\pm2}$ | $39_{\pm10}$ | $\mathbf{60}_{\pm41}$ |
| | | | task2 | $\mathbf{0}_{\pm0}$ | $\mathbf{0}_{\pm0}$ | $\mathbf{0}_{\pm0}$ | $\mathbf{0}_{\pm0}$ | $\mathbf{0}_{\pm0}$ | $\mathbf{0}_{\pm0}$ |
| | | | task3 | $0_{\pm0}$ | $0_{\pm0}$ | $0_{\pm0}$ | $0_{\pm0}$ | $0_{\pm0}$ | $\mathbf{2}_{\pm2}$ |
| | | | task4 | $\mathbf{0}_{\pm0}$ | $\mathbf{0}_{\pm0}$ | $\mathbf{0}_{\pm0}$ | $\mathbf{0}_{\pm0}$ | $\mathbf{0}_{\pm0}$ | $\mathbf{0}_{\pm0}$ |
| | | | task5 | $\mathbf{0}_{\pm0}$ | $\mathbf{0}_{\pm0}$ | $\mathbf{0}_{\pm0}$ | $\mathbf{0}_{\pm0}$ | $\mathbf{0}_{\pm0}$ | $\mathbf{0}_{\pm0}$ |
| | | | overall | $9_{\pm0}$ | $0_{\pm0}$ | $2_{\pm2}$ | $0_{\pm0}$ | $8_{\pm2}$ | $\mathbf{12}_{\pm8}$ |

Table 22: **Full results on Visual Scene.**

| Environment Type | Dataset Type | Dataset | Task | GCBC | GCIVL | GCIQL | QRL | CRL | HIQL |
|---|---|---|---|---|---|---|---|---|---|
| visual-scene | play | visual-scene-play-v0 | task1 | $59_{\pm7}$ | $\mathbf{84}_{\pm4}$ | $56_{\pm4}$ | $44_{\pm6}$ | $52_{\pm6}$ | $80_{\pm6}$ |
| | | | task2 | $0_{\pm0}$ | $24_{\pm8}$ | $1_{\pm1}$ | $2_{\pm2}$ | $1_{\pm1}$ | $\mathbf{81}_{\pm7}$ |
| | | | task3 | $0_{\pm0}$ | $16_{\pm8}$ | $0_{\pm0}$ | $0_{\pm0}$ | $0_{\pm0}$ | $\mathbf{61}_{\pm11}$ |
| | | | task4 | $2_{\pm1}$ | $0_{\pm0}$ | $3_{\pm4}$ | $2_{\pm1}$ | $1_{\pm1}$ | $\mathbf{20}_{\pm8}$ |
| | | | task5 | $0_{\pm0}$ | $0_{\pm0}$ | $0_{\pm0}$ | $0_{\pm0}$ | $0_{\pm0}$ | $\mathbf{3}_{\pm2}$ |
| | | | overall | $12_{\pm2}$ | $25_{\pm3}$ | $12_{\pm2}$ | $10_{\pm1}$ | $11_{\pm2}$ | $\mathbf{49}_{\pm4}$ |
| | noisy | visual-scene-noisy-v0 | task1 | $64_{\pm9}$ | $76_{\pm6}$ | $49_{\pm22}$ | $8_{\pm2}$ | $70_{\pm9}$ | $\mathbf{91}_{\pm4}$ |
| | | | task2 | $0_{\pm0}$ | $14_{\pm7}$ | $2_{\pm2}$ | $0_{\pm0}$ | $0_{\pm0}$ | $\mathbf{69}_{\pm5}$ |
| | | | task3 | $0_{\pm0}$ | $24_{\pm5}$ | $7_{\pm2}$ | $0_{\pm0}$ | $2_{\pm2}$ | $\mathbf{82}_{\pm6}$ |
| | | | task4 | $0_{\pm1}$ | $0_{\pm0}$ | $0_{\pm0}$ | $0_{\pm0}$ | $1_{\pm1}$ | $\mathbf{8}_{\pm5}$ |
| | | | task5 | $\mathbf{0}_{\pm0}$ | $\mathbf{0}_{\pm0}$ | $\mathbf{0}_{\pm0}$ | $\mathbf{0}_{\pm0}$ | $\mathbf{0}_{\pm0}$ | $\mathbf{0}_{\pm0}$ |
| | | | overall | $13_{\pm2}$ | $23_{\pm2}$ | $12_{\pm4}$ | $2_{\pm0}$ | $15_{\pm2}$ | $\mathbf{50}_{\pm1}$ |

Table 23: **Full results on Visual Puzzle.**

| Environment Type | Dataset Type | Dataset | Task | GCBC | GCIVL | GCIQL | QRL | CRL | HIQL |
|---|---|---|---|---|---|---|---|---|---|
| visual-puzzle | play | visual-puzzle-3x3-play-v0 | task1 | 1 ±1 | **97** ±2 | 5 ±9 | 3 ±3 | 2 ±1 | **98** ±1 |
| | | | task2 | 0 ±0 | 1 ±1 | 0 ±0 | 0 ±0 | 0 ±0 | **73** ±10 |
| | | | task3 | 0 ±0 | 0 ±1 | 0 ±0 | 0 ±0 | 0 ±0 | **64** ±9 |
| | | | task4 | 0 ±0 | 2 ±3 | 0 ±0 | 0 ±0 | 0 ±0 | **63** ±9 |
| | | | task5 | 0 ±0 | 3 ±2 | 0 ±0 | 0 ±0 | 0 ±0 | **66** ±12 |
| | | | overall | 0 ±0 | 21 ±1 | 1 ±2 | 1 ±1 | 0 ±0 | **73** ±8 |
| | | visual-puzzle-4x4-play-v0 | task1 | 11 ±3 | **86** ±7 | 18 ±5 | 0 ±0 | 10 ±7 | 70 ±47 |
| | | | task2 | 19 ±5 | 8 ±5 | 28 ±12 | 0 ±0 | 19 ±13 | **38** ±32 |
| | | | task3 | 8 ±3 | **80** ±4 | 15 ±4 | 0 ±0 | 8 ±7 | 66 ±44 |
| | | | task4 | 6 ±2 | **65** ±9 | 10 ±5 | 0 ±0 | 7 ±5 | **66** ±44 |
| | | | task5 | 6 ±2 | **61** ±8 | 9 ±3 | 0 ±0 | 4 ±4 | 60 ±41 |
| | | | overall | 10 ±1 | **60** ±5 | 16 ±4 | 0 ±0 | 10 ±6 | **60** ±41 |
| | | visual-puzzle-4x5-play-v0 | task1 | 22 ±8 | **86** ±4 | 31 ±8 | 0 ±0 | 31 ±6 | 66 ±44 |
| | | | task2 | 1 ±1 | 0 ±0 | 1 ±1 | 0 ±0 | 0 ±1 | **0** ±0 |
| | | | task3 | **0** ±0 | **0** ±0 | **0** ±0 | **0** ±0 | **0** ±0 | **0** ±0 |
| | | | task4 | **0** ±0 | **0** ±0 | **0** ±0 | **0** ±0 | **0** ±0 | **0** ±0 |
| | | | task5 | **0** ±0 | **0** ±0 | **0** ±0 | **0** ±0 | **0** ±0 | **0** ±0 |
| | | | overall | 5 ±2 | **17** ±1 | 7 ±2 | 0 ±0 | 6 ±1 | 13 ±9 |
| | | visual-puzzle-4x6-play-v0 | task1 | 12 ±4 | **66** ±3 | 10 ±3 | 0 ±0 | 12 ±5 | 34 ±23 |
| | | | task2 | 0 ±1 | **8** ±2 | 2 ±2 | 0 ±0 | 2 ±1 | **8** ±6 |
| | | | task3 | **0** ±0 | **0** ±0 | **0** ±0 | **0** ±0 | **0** ±0 | **0** ±0 |
| | | | task4 | **0** ±0 | **0** ±0 | **0** ±0 | **0** ±0 | **0** ±0 | **0** ±0 |
| | | | task5 | **0** ±0 | **0** ±0 | **0** ±0 | **0** ±0 | **0** ±0 | **0** ±0 |
| | | | overall | 2 ±1 | **15** ±1 | 2 ±1 | 0 ±0 | 3 ±1 | 9 ±6 |
| | noisy | visual-puzzle-3x3-noisy-v0 | task1 | 4 ±6 | **100** ±1 | 98 ±2 | 0 ±0 | 7 ±7 | **100** ±0 |
| | | | task2 | 0 ±0 | 0 ±1 | 5 ±7 | 0 ±0 | 0 ±0 | **64** ±11 |
| | | | task3 | 0 ±0 | 0 ±0 | 2 ±2 | 0 ±0 | 0 ±0 | **55** ±3 |
| | | | task4 | 0 ±0 | 0 ±0 | 5 ±4 | 0 ±0 | 0 ±0 | **61** ±8 |
| | | | task5 | 0 ±0 | 0 ±0 | 19 ±10 | 0 ±0 | 0 ±0 | **71** ±13 |
| | | | overall | 1 ±1 | 20 ±0 | 26 ±4 | 0 ±0 | 1 ±1 | **70** ±6 |
| | | visual-puzzle-4x4-noisy-v0 | task1 | 6 ±2 | 90 ±9 | 85 ±3 | 0 ±0 | 4 ±3 | **98** ±2 |
| | | | task2 | 16 ±7 | 1 ±1 | 1 ±1 | 0 ±0 | 14 ±11 | **44** ±20 |
| | | | task3 | 4 ±3 | 88 ±4 | 77 ±6 | 0 ±0 | 4 ±4 | **95** ±2 |
| | | | task4 | 2 ±1 | 36 ±10 | 47 ±17 | 0 ±0 | 2 ±1 | **91** ±2 |
| | | | task5 | 4 ±2 | 20 ±9 | 34 ±13 | 0 ±0 | 5 ±4 | **94** ±1 |
| | | | overall | 7 ±3 | 47 ±3 | 49 ±7 | 0 ±0 | 6 ±2 | **84** ±4 |
| | | visual-puzzle-4x5-noisy-v0 | task1 | 30 ±6 | 72 ±48 | **96** ±2 | 0 ±0 | 33 ±5 | 72 ±48 |
| | | | task2 | **0** ±0 | **0** ±0 | **0** ±0 | **0** ±0 | **0** ±0 | **0** ±0 |
| | | | task3 | **0** ±0 | **0** ±0 | **0** ±0 | **0** ±0 | **0** ±0 | **0** ±0 |
| | | | task4 | **0** ±0 | **0** ±0 | **0** ±0 | **0** ±0 | **0** ±0 | **0** ±0 |
| | | | task5 | **0** ±0 | **0** ±0 | **0** ±0 | **0** ±0 | **0** ±0 | **0** ±0 |
| | | | overall | 6 ±1 | 14 ±10 | **19** ±0 | 0 ±0 | 7 ±1 | 14 ±10 |
| | | visual-puzzle-4x6-noisy-v0 | task1 | 10 ±2 | 61 ±41 | **82** ±4 | 0 ±0 | 9 ±7 | 56 ±7 |
| | | | task2 | 0 ±0 | 1 ±1 | 2 ±1 | 0 ±0 | 0 ±0 | **12** ±10 |
| | | | task3 | **0** ±0 | **0** ±0 | **0** ±0 | **0** ±0 | **0** ±0 | **0** ±0 |
| | | | task4 | **0** ±0 | **0** ±0 | **0** ±0 | **0** ±0 | **0** ±0 | **0** ±0 |
| | | | task5 | **0** ±0 | **0** ±0 | **0** ±0 | **0** ±0 | **0** ±0 | **0** ±0 |
| | | | overall | 2 ±1 | 12 ±8 | **17** ±1 | 0 ±0 | 2 ±1 | 14 ±2 |

Table 24: **Full results on Powderworld.**

| Environment Type | Dataset Type | Dataset | Task | GCBC | GCIVL | GCIQL | QRL | CRL | HIQL |
|---|---|---|---|---|---|---|---|---|---|
| powderworld | play | powderworld-easy-play-v0 | task1 | 1 ±1 | **99** ±1 | 95 ±4 | 25 ±8 | 40 ±6 | 62 ±5 |
| | | | task2 | 0 ±0 | **96** ±4 | 93 ±3 | 12 ±4 | 43 ±10 | 54 ±10 |
| | | | task3 | 0 ±0 | **100** ±0 | 94 ±8 | 8 ±7 | 15 ±5 | 41 ±26 |
| | | | task4 | 0 ±0 | **100** ±0 | 92 ±8 | 8 ±5 | 10 ±3 | 8 ±6 |
| | | | task5 | 0 ±0 | **100** ±1 | 93 ±5 | 7 ±3 | 2 ±3 | 2 ±2 |
| | | | overall | 0 ±0 | **99** ±1 | 93 ±5 | 12 ±2 | 22 ±5 | 33 ±9 |
| | | powderworld-medium-play-v0 | task1 | 0 ±0 | **81** ±12 | 7 ±14 | 2 ±3 | 0 ±1 | 26 ±20 |
| | | | task2 | 3 ±3 | **28** ±14 | 0 ±0 | 9 ±6 | 2 ±3 | 16 ±12 |
| | | | task3 | 2 ±3 | **99** ±2 | 72 ±14 | 4 ±5 | 2 ±2 | 60 ±37 |
| | | | task4 | 0 ±0 | **39** ±17 | 0 ±1 | 0 ±0 | 0 ±0 | 5 ±4 |
| | | | task5 | 0 ±0 | **2** ±2 | 0 ±0 | 0 ±0 | 0 ±0 | 1 ±2 |
| | | | overall | 1 ±1 | **50** ±4 | 16 ±5 | 3 ±1 | 1 ±1 | 22 ±14 |
| | | powderworld-hard-play-v0 | task1 | **0** ±0 | **0** ±0 | **0** ±0 | **0** ±0 | **0** ±0 | **0** ±0 |
| | | | task2 | 0 ±0 | 4 ±3 | 0 ±0 | 0 ±0 | 0 ±0 | 1 ±2 |
| | | | task3 | 0 ±0 | 4 ±4 | 0 ±0 | 0 ±0 | 0 ±0 | 2 ±2 |
| | | | task4 | 0 ±0 | 12 ±14 | 0 ±0 | 0 ±0 | 0 ±0 | 0 ±0 |
| | | | task5 | **0** ±0 | **0** ±0 | **0** ±0 | **0** ±0 | **0** ±0 | **0** ±1 |
| | | | overall | 0 ±0 | 4 ±3 | 0 ±0 | 0 ±0 | 0 ±0 | 1 ±1 |

