# OpenReview forum: "OGBench: Benchmarking Offline Goal-Conditioned RL"
_ICLR.cc/2025/Conference — ICLR 2025 Poster_

### Official Review · Reviewer_BGac · 2024-10-16

**Soundness:** 3
**Presentation:** 3
**Contribution:** 3
**Rating:** 6
**Confidence:** 4

**Summary:**

This paper contributes to offline GCRL by introducing a suite of high-quality benchmarks. These benchmarks systematically assess offline GCRL algorithms' performance in learning from suboptimal data, goal stitching, long-horizon reasoning, and handling stochasticity, providing a useful tool for advancing offline GCRL research.

**Strengths:**

1. They are very useful benchmarks to evaluate offline GCRL methods.
2. The authors evaluate $6$ standard offline GCRL methods on the proposed benchmarks, establishing a solid baseline for comparison and further development.

**Weaknesses:**

The positioning of this paper is very similar to a previous paper - D4RL (Fu et al., 2021), which was well-known but rejected by ICLR 2021 (https://openreview.net/forum?id=px0-N3_KjA).

This paper makes a clear contribution by providing valuable benchmarks, but it does not introduce novel ideas. However, given its potential importance to the field, I lean towards assigning a slightly positive score.

**Questions:**

1. It would be better to retain one or two decimal places in Table 2. In benchmarks such as D4RL (Fu et al., 2021) and offline robotics (Yang et al., 2022), the success rate/score provided by decimal places is sometimes critical for comparisons.
2. Could the authors clarify in the paper whether a rollout is considered successful if the goal is achieved at the last state, or if any state within the rollout reaches the goal?
3. The authors should include a mention of GOPlan (Wang et al., 2024), which applies model-based planning with a generative adversarial network as the prior policy, in the related work. Additionally, GOPlan and GoFar (Ma et al., 2022) evaluate algorithm robustness in stochastic offline GCRL settings and should be discussed in Section 5.4.


**References**

Fu, J., Kumar, A., Nachum, O., Tucker, G., \& Levine, S. (2021). D4RL: Datasets for Deep Data-Driven Reinforcement Learning.

Ma, J. Y., Yan, J., Jayaraman, D., \& Bastani, O. (2022). Offline Goal-Conditioned Reinforcement Learning via f-Advantage Regression. NeurIPS.

Wang, M., Yang, R., Chen, X., Sun, H., Fang, M., \& Montana, G. (2024). GOPlan: Goal-conditioned Offline Reinforcement Learning by Planning with Learned Models. TMLR.

Yang, R., Lu, Y., Li, W., Sun, H., Fang, M., Du, Y., … Zhang, C. (2022). Rethinking Goal-conditioned Supervised Learning and Its Connection to Offline RL. ICLR.

---

> ### Author Response · Authors · 2024-11-16
> **Author Response**
>
> Thank you for the detailed review and constructive feedback about this work. We especially appreciate the reviewer's clarification question about our goal success criterion. Please find our response below.
>
> * **“This paper makes a clear contribution by providing valuable benchmarks, but it does not introduce novel ideas.”**
>
> Thanks for the positive comments! Indeed, our goal in this work is not to propose a novel method but to provide a new benchmark. Just for clarification, we would like to note that “datasets and benchmarks” is one of the ICLR subject areas ([ICLR 2025 Call for Papers](https://iclr.cc/Conferences/2025/CallForPapers)).
>
> * **Similarity to D4RL**
>
> While D4RL primarily focuses on single-task RL, our benchmark aims to complement D4RL to provide a thorough suite of tasks and algorithms for studying offline goal-conditioned RL. Our benchmark also addresses some key limitations of D4RL (e.g., task performances are saturated [1], it only supports single-goal evaluation, and most tasks do not support pixel-based observations). Unlike D4RL, OGBench provides much more challenging tasks and datasets specifically designed for offline goal-conditioned RL (e.g., HumanoidMaze, AntSoccer, Scene, Puzzle, Powderworld, etc.), while posing diverse challenges for goal-conditioned RL, including long-horizon reasoning, pixel-based control, goal stitching, and stochastic control.
>
> * **“It would be better to retain one or two decimal places in Table 2.”**
>
> Thanks for the suggestion! Following the suggestion, we have revised Tables 2 and 10-16 to show one decimal place; please find them in the revised PDF.
>
> * **How is a rollout considered successful at goal-reaching?**
>
> Thanks for asking this clarification question. During evaluation, the trajectory ends immediately after the agent reaches the goal, meaning that we consider a trajectory to be successful if it contains at least one state (not necessarily the final one) that achieves the goal. It seems we were not entirely clear about our evaluation protocol in the initial draft, and we have clarified this point by making a separate paragraph (titled “Evaluation”) at the beginning of Section 7 of the revised PDF.
>
> * **Missing related works: GOPlan and GoFAR**
>
> Thanks for pointing out these relevant works! We have revised the draft to mention GOPlan and GoFAR (please refer to Section 3 and Appendix C of the revised PDF).
>
> ---
>
> We would like to thank the reviewer again for raising important questions about OGBench. We believe the added clarification about the success criterion as well as the discussions about further related works have improved the quality of the paper. **Have we sufficiently addressed the reviewer’s main concerns?** Please feel free to let us know if there are additional concerns or questions.
>
> [1] Tarasov et al., Revisiting the minimalist approach to offline reinforcement learning, NeurIPS 2023.

---

> > ### Comment · Reviewer_BGac · 2024-11-21
> >
> > Thank you for your reply. This could be an influential work for the offline GCRL community. I maintain my current positive rating.

---

### Official Review · Reviewer_sCze · 2024-10-26

**Soundness:** 3
**Presentation:** 3
**Contribution:** 4
**Rating:** 6
**Confidence:** 3

**Summary:**

This paper introduces OGBench, a comprehensive benchmark for the standardized evaluation of offline GCRL. OGBench provides a diverse range of environments and datasets, covering areas such as locomotion, manipulation, and drawing, to assess key capabilities like trajectory stitching, long-horizon reasoning, image-input handling, and managing stochasticity. The authors evaluate six offline GCRL algorithms using this benchmark, designed to highlight each algorithm's unique strengths and weaknesses across different tasks. OGBench is optimized for computational efficiency and user-friendliness, allowing researchers to easily test and refine their ideas, thereby advancing the field of offline GCRL.

**Strengths:**

* Offline GCRL is an important research direction, and this work addresses the current lack of a challenging and comprehensive benchmark.

* This benchmark evaluates GCRL algorithms across diverse aspects, including stitching, stochastic environments, high-dimensional control, and long-horizon control.

* The benchmark includes popular baseline results, highlighting some unsolved tasks and areas for improvement.

**Weaknesses:**

Some benchmark designs need further discussion or improvement.

* The evaluation seems limited to five tasks. Does this mean there are only five evaluation goals? If so, this number is restricted compared to several prior goal-conditioned tasks such as the Fetch environments.

* The design of the transparent arm in pixel-based manipulation tasks seems unrealistic. It would be better to let users choose between a transparent or solid option. Additionally, demonstrating how transparency affects performance could be helpful.

* The goal information is missing in Table 6. Is the goal in image-based tasks another image provided to the agent, or is it just a transparent bot as shown in the manipulation task figures?

* Can the benchmark support language instructions as a special type of "goal"? This would be interesting given the current research focus on LLMs.

* The stochastic task appears confined to the teleport domain. More stochastic settings that might be relevant to real-world scenarios should be explored, such as random observation perturbation during both data collection and evaluation.

* Current baselines still use MLP networks as backbones. Could the authors consider implementing more advanced architectures, such as transformers? Are there any limitations in the current tasks that more sophisticated network architectures might address?

**Questions:**

See the weakness part.

---

> ### Author Response · Authors · 2024-11-16
> **Author Response (1/2)**
>
> Thank you for the detailed review and constructive feedback about this work. We especially appreciate the reviewer's feedback about evaluation goals, the use of the transparent arm, and several clarification questions. Following the reviewer's suggestion, we have added an option to reproduce datasets with an opaque arm. Please find our response below.
>
> * **Evaluation is limited to five goals. Does this mean there are only five evaluation goals?**
>
> The reviewer’s understanding is correct — we provide five evaluation goals for each task in the benchmark, although their positions, poses, color orders, etc., are slightly randomized each time, and we perform multiple ($50$) rollouts for each evaluation goal. We would first like to note that this type of evaluation (i.e., having a fixed set of evaluation tasks) is quite commonly used in prior works in offline goal/language-conditioned RL and robotics [1, 2, 3]. There are two main reasons for this choice: (1) using five pre-defined goals reduces computational cost for evaluation (which sometimes takes even longer than training in long-horizon environments!), as evaluation with fully randomized goals typically requires many more evaluation rollouts to derive statistically significant results and (2) providing diverse *types* of pre-defined goals enables the practitioner to better analyze their algorithm’s strengths and weaknesses. When designing the tasks, we put substantial effort into choosing a small number of *representative* evaluation goals to maximize research signals while minimizing computational cost. For example, in $\texttt{cube-double}$ tasks, we curated five evaluation goals to cover single pick-and-place, double pick-and-place, swapping, stacking, etc. (Figure 6). The results in Table 14 show that some methods are good at single pick-and-place but struggle with tasks involving double pick-and-place, which can guide practitioners on areas for improvement in their methods.
>
> * **The use of the transparent arm**
>
> Thanks for asking this question! First of all, following the suggestion, we have added an option (`pixel_transparent_arm={False, True}` in `manipspace_env.py`) for the manipulation environments to make the transparency of the arm configurable. Since we have provided the exact commands to reproduce all the datasets, users who wish to use a non-transparent arm can easily generate and use opaque versions of the visual manipulation datasets.
>
> As for why we chose the transparent arm as the default, we also debated this point; there are pros and cons. A solid arm looks more natural, but due to occlusion in some tasks (e.g., Puzzle), it requires the use of a memory component (e.g., RNNs, Transformers) or multiple viewpoints, which can significantly increase computational burdens. On the other hand, a transparent arm looks less natural, but it enables using a single $64 \times 64 \times 3$ image and does not require any additional memory components. Among these two choices, we decided to choose the latter, following our design principle (design principle 4 in Section 6): **minimize unnecessary computation and focus on algorithmic challenges**. After all, OGBench is a benchmark mainly for *algorithms research*, so we prioritized minimizing other orthogonal challenges so that researchers can more quickly iterate on their algorithmic ideas. To further clarify this, we have mentioned this trade-off as a limitation in the newly added limitation section (Appendix A) of the revised PDF. Nevertheless, we believe the new option to configure the transparency of the arm provides additional flexibility for users.
>
> * **“The goal information is missing in Table 6.” / How is a goal given for image-based tasks?**
>
> Thanks for asking the clarification question. In all tasks in our benchmark, the goal space is the same as the state space (L100); in other words, a goal is simply yet another state. As such, in image-based tasks, the agent receives as a goal $g$ the rendering of the desired state (with the transparent arm by default). It seems we were not entirely clear about our evaluation protocol in the initial draft, and we have clarified this point by making a separate paragraph (titled “Evaluation”) at the beginning of Section 7 of the revised PDF.

---

> ### Author Response · Authors · 2024-11-16
> **Author Response (2/2)**
>
> * **Can the benchmark support language instructions as a special type of "goal"?**
>
> Thanks for the suggestion. While language-conditioned RL/control is an important problem, we consider it to be beyond the scope of this benchmark. However, we believe it may not be very difficult to convert many manipulation tasks into language-conditioned tasks (e.g., we can describe evaluation tasks by “swap the cubes”, “turn all buttons into blue”, “open the drawer and window”, etc.). We leave the integration of language instructions and OGBench tasks as an interesting potential extension of this work.
>
> * **About stochastic tasks / “The stochastic task appears confined to the teleport domain.”**
>
> In OGBench, we provide two different types of stochastic environments: (1) the “teleport” maze in navigation environments and (2) Powderworld, whose transition dynamics are highly stochastic and unpredictable. As the reviewer mentioned, our manipulation environments have deterministic dynamics (though the poses/colors of objects are slightly randomized/permuted each time), as in many previous simulated tasks in robotic manipulation (e.g., Fetch [4], Kitchen [5], CALVIN [6], etc.). This is mainly because it is not entirely straightforward to design realistic manipulation tasks with truly stochastic dynamics. That said, we believe the new option for choosing an opaque arm (please see our response above) can provide an additional challenge regarding stochasticity (via partial observability) in manipulation domains.
>
> * **Current baselines still use MLP networks as backbones.**
>
> As the reviewer pointed out, our baselines are based on MLPs and CNNs. While we were unable to add and tune Transformer-based baselines within this short discussion period due to our limited computing resources, we believe complex tasks in OGBench (e.g., $\texttt{visual-humanoidmaze-giant}$, $\texttt{puzzle-4x6}$, $\texttt{cube-quadruple}$) do require scalable and expressive networks and thus provide a ground for future research involving modern architectures. We leave this extension as an exciting future research opportunity.
>
> ---
>
> We would like to thank the reviewer again for raising important questions about OGBench. We believe the new clarifications and the additional features have substantially improved the quality of this work. Have we sufficiently addressed the reviewer’s main concerns? Please feel free to let us know if there are additional concerns or questions.
>
> ---
>
> [1] Fang et al., Planning to Practice: Efficient Online Fine-Tuning by Composing Goals in Latent Space, IROS 2022. \
> [2] Zheng et al., Stabilizing Contrastive RL: Techniques for Robotic Goal Reaching from Offline Data, ICLR 2024. \
> [3] Ghugare et al., Closing the Gap between TD Learning and Supervised Learning -- A Generalisation Point of View, ICLR 2024. \
> [4] Plappert et al., Multi-Goal Reinforcement Learning: Challenging Robotics Environments and Request for Research, 2018. \
> [5] Gupta et al., Relay Policy Learning: Solving Long-Horizon Tasks via Imitation and Reinforcement Learning, CoRL 2019. \
> [6] Mees et al., CALVIN - A benchmark for Language-Conditioned Policy Learning for Long-Horizon Robot Manipulation Tasks, RA-L 2022.

---

> > ### Comment · Reviewer_sCze · 2024-11-21
> >
> > Thank you for your response.Overall, I think this work is valuable to the RL community, and I will maintain my current positive rating.

---

### Official Review · Reviewer_t6ud · 2024-10-30

**Soundness:** 3
**Presentation:** 4
**Contribution:** 3
**Rating:** 8
**Confidence:** 4

**Summary:**

The paper proposes a new suite of tasks for benchmarking goal-conditioned offline RL algorithms. The proposed benchmarks are built with consideration specific to the goal-conditioned navigation tasks, with an emphasis on evaluating relevant task capabilities such as stitching, long-horizon reasoning, dealing with stochastic and visual input. The tasks cover a range of difficulty levels and types, including ant and humanoid locomotion, multi-step manipulation and a painting task with an high-dimension state space entailing combinatorial search. Offline policy data are also collected with various policies to emulate the data sub optimality. The benchmark is shown to be challenging for state-of-the-art and helpful in analysing strengths and limitations on investigated task capabilities. Code and policy scripted are provided and with a minimum dependencies on open-sourced mujoco and PyTorch libraries.

**Strengths:**

* Well-motivated research efforts and a solid execution with many considerations customised to goal-conditioned offline RL.

* A good coverage of task variations and challenges to state-of-the-art algorithms based on different principles.

* Proper task difficulties demonstrated from the results. Could be promising to spur new algorithmic research.

* Decent writing with clear presentation and well organised flow of narratives.

**Weaknesses:**

* The benchmark is motivated for the generality of goal-conditioned navigation tasks, aiming at learning transferrable representations for down-stream tasks. However, this is not reflected in the task design and experimental results. It would be better to include some functionalities to facilitate analysing and transferring the learned latent representations, with a report on a few SotA algorithms' performance on this aspect.

* The tasks seem to only challenge policies with different initial states but not the shift of transition dynamics.

**Questions:**

* The expert data are collected from policies trained by RL. How can we assume the policies provide sufficient optimality? Since the all the tasks are essentially planning towards a goal, would running motion planning/search algorithms with certain guarantees give better data quality?

* Can the benchmark add a study on the representation and the transferability to assess the idea of using goal-conditioned tasks as a general-purpose representation learning?

---

> ### Author Response · Authors · 2024-11-16
> **Author Response**
>
> Thank you for the detailed review and positive and constructive feedback about this work. We especially appreciate the reviewer's questions about representation learning and the optimality of datasets. Please find our response below.
>
> * **“Can the benchmark add a study on the representation and the transferability to assess the idea of using goal-conditioned tasks as a general-purpose representation learning?”**
>
> Thanks for the suggestion! We completely agree that evaluating the transferability of representations to different types of downstream tasks is an important problem. In fact, we are currently working on this very problem (i.e., how to pre-train and fine-tune GCRL representations trained on diverse, unlabeled data for downstream tasks) using the OGBench locomotion/manipulation datasets. However, instead of adding our initial results on representation transfer to the final version of this paper, we hope to separate it into a different study to have more comprehensive analyses and experiments, given the difference between the two problem settings (offline goal-conditioned RL vs. representation-based downstream adaptation). Nevertheless, we would like to highlight that OGBench can indeed facilitate studies on representation learning by providing a variety of unlabeled datasets, and we hope to provide extensive analyses and results (as a separate work) as a more thorough answer to this question in the near future.
>
> * **“The tasks seem to only challenge policies with different initial states but not the shift of transition dynamics.”**
>
> As the reviewer mentioned, we do not specifically address challenges with distributional shifts in transition dynamics in this benchmark. While this is an important issue, we consider it beyond the scope of this work, and would like to refer to other great benchmarks specialized in off-dynamics learning, such as ODRL [1]. We have mentioned this as a limitation in the newly added limitation section (Appendix A) of the revised PDF.
>
> * **“How can we assume the policies provide sufficient optimality?”**
>
> Thanks for the question. In locomotion environments, we trained a low-level directional policy (using RL) to move as far as possible in a commanded direction, and combined it with an oracle BFS-based high-level planner to generate diverse maze-navigation trajectories. Since the objective of the low-level policy is to simply maximize traveled distances, we can easily estimate its performance (optimality). When creating the datasets and tasks, we confirmed that the trained low-level policies were reasonably optimal and that the evaluation tasks were solvable by the expert policy and oracle planner within the maximum episode length. For manipulation tasks, we used manually scripted policies to collect datasets (rather than RL), which we found sufficient for generating datasets.
>
> We would like to thank the reviewer again for raising important questions about OGBench. We hope that our response has addressed the reviewer's questions, and please feel free to let us know if there are any additional concerns or questions.
>
> [1] Lyu et al., ODRL: A Benchmark for Off-Dynamics Reinforcement Learning, NeurIPS 2024.

---

> > ### Comment · Reviewer_t6ud · 2024-11-18
> >
> > Thank you for the discussion and texts for addressing my comments. It is good to know the literature on the off-dynamics benchmark. I maintain my opinion that this is a good and timely work piece and the original score.

---

### Official Review · Reviewer_jjTu · 2024-10-31

**Soundness:** 4
**Presentation:** 4
**Contribution:** 4
**Rating:** 8
**Confidence:** 4

**Summary:**

In this work the authors present OGBench, a novel benchmark for offline goal-conditioned reinforcement learning (GCRL). The authors motivate the need for such benchmark, due to the lack of a standardized evaluation suite for works in offline GCRL and the increasing interest from the research community on the topic. The authors describe the main challenges tackled with OGBench: learning from sub-optimal data, goal stitching, long-horizon reasoning and stochastic environments. To tackle these challenges, the authors describe the design principles of the benchmark and consider a wide range tasks (locomotion, manipulation and drawing) across multiple environments. Additionally, the authors collect several different datasets per environment, considering different sub-optimality conditions of the behavior policy. Finally, the authors evaluate several literature-standard algorithms in OGBench, contributing their implementation as well, and discuss their performance and other findings.

**Strengths:**

- **Originality**:
  - Recently, there has been an extensive push towards the development of benchmarks that focus on different aspects of reinforcement learning (e.g., for offline RL [1], [2]). However, as noted by the authors, there existed no centralized evaluation suite for algorithms in offline GCRL and the community often resorted to different set of tasks (not tailored for, making difficult to access the true performance of the proposed methods. As such, the development of a benchmark that tackles specific research issues in offline GCRL is most welcomed and, to the best of my knowledge, novel.

- **Quality**:
   - The work presented here is of high quality: the proposed benchmark contains a set of diverse tasks that address specific problems in offline GCRL research and high-quality, fine-tuned implementations of recently proposed algorithms for offline GCRL (that often surpass the performance of the original versions). Furthermore, the authors collect a wide range of datasets for each task, with different levels of sub-optimality.

- **Clarity**:
   - The current version of the work is well-written, with no major typos (that I could detect). Furthermore, the document is easy to read, with self-explanatory figures and tables. I would, however, recommend some refrain when using adjectives such as "cool" (Section 6, page 4) or "exciting" (Section 6, page 4), due to their intrinsic subjectivity.

- **Significance**:
   - This work proposes a novel benchmark for offline GCRL. Given the relevance of the topic for RL research, the work can have substantial impact by providing a standardized suite for evaluation of novel algorithms.


**References**:

- [1] Seno, Takuma, and Michita Imai. "d3rlpy: An offline deep reinforcement learning library." Journal of Machine Learning Research 23.315 (2022): 1-20.
- [2] Fu, Justin, et al. "D4rl: Datasets for deep data-driven reinforcement learning." arXiv preprint arXiv:2004.07219 (2020).

**Weaknesses:**

My concerns with the current version of the work are the following (none of them particularly major):
- Currently, it is challenging to assess the varying levels of novelty present in the environments available in OGBench. Throughout Section 7, I found myself questioning whether each environment was accessible elsewhere (in another benchmark or repository) or if it was unique to this benchmark. I do understand that some tasks (e.g., AntMaze) are heavily inspired by/available in other benchmarks. It would strengthen the originality of the benchmark if the authors clarified the level of novelty in each environment.
- Currently the paper lacks any discussion about the limitations of the current benchmark and plans for future extension (for example, one potential extension could tackle robustness to data corruption as explored in [1], or integrating scenarios with multiple modalities).
- Is not clear how the five evaluation goals per task were defined or what criteria was used for their selection. Additionally, why only 5 goals? Since GC policies should be able to reach any (viable) state from any state, it is unclear why the authors selected such a set of small goals.


**References**

- [1] Yang, Rui, et al. "Towards Robust Offline Reinforcement Learning under Diverse Data Corruption." The Twelfth International Conference on Learning Representations.

**Questions:**

Please, also refer to the questions brought up in the "Weaknesses" section.

- **1** - Why not also collect an "exploratory" dataset for the AntSoccer task? Is it due to the lack of coverage of states where the agent is dribbling the ball?
- **2** - In the locomotion and manipulation tasks, how exactly is the goal state provided to the agent? Does it also include the final pose of the agent/robot? In that case, the task goal becomes a convolution of an environmental change (for example, a set of objects in a specific location in the Scene task) and a final pose of the agent?
- **3** - I found it quite interesting the comment at the end of Section 8.2 ("this suggests that we may need to prioritize coverage much more than optimality when collecting datasets for offline GCRL in the real world") as this seems to go against the current trends in large scale collection of interaction data in robotics, where the data are collected across multiple environments/embodiments but always by expert-level demonstrators (e.g., [1]). Can the authors make any parallelism between the case of offline GCRL and other large-scale data collection efforts?


**Reference**:
- [1] Collaboration, Open-X. Embodiment, et al. "Open X-Embodiment: Robotic learning datasets and RT-X models." arXiv preprint arXiv:2310.08864 1.2 (2023).

---

> ### Author Response · Authors · 2024-11-16
> **Author Response (1/2)**
>
> Thank you for the detailed review and positive and constructive feedback about this work. We especially appreciate the reviewer’s questions about limitations and the importance of coverage in data collection. Please find our response below.
>
> * **How new are the OGBench tasks?**
>
> We provide $7$ types of tasks in this benchmark: AntMaze, HumanoidMaze, AntSoccer, Cube, Scene, Puzzle, and Powderworld. OGBench AntMaze is a direct extension of D4RL AntMaze, where we provide more types of datasets, mazes, and observation modalities (states/pixels) to pose further challenges. Powderworld is an existing environment for online RL [1], but we designed evaluation tasks and datasets to make it offline and goal-conditioned.
>
> The other five types of tasks (HumanoidMaze, AntSoccer, Cube, Scene, Puzzle) were designed by us; especially, the manipulation tasks were entirely built from scratch using publicly available MuJoCo assets for objects and robots. That said, some tasks were conceptually inspired by existing online RL tasks (e.g., DMC quadruped-fetch for AntSoccer), as mentioned in the paper (L328).
>
> * **Limitations of OGBench**
>
> Thanks for pointing this out. Following the suggestion, we have added a limitation section in the revised version of the paper (Appendix A):
>
> > **Limitations.** While OGBench covers a number of challenges in offline goal-conditioned RL, such as long-horizon reasoning, goal stitching, and stochastic control, there exist other challenges that our benchmark does not address. For example, all OGBench tasks assume that the environment dynamics remain the same between the training and evaluation environments. Also, although several OGBench tasks (e.g., Cube, Puzzle, and Powderworld) require unseen goal generalization to some degree, our tasks do not specifically test visual generalization to entirely new objects. Finally, we have made several trade-offs to reduce computational cost and to focus the benchmark on algorithms research at the expense of sacrificing realism to some degree (e.g., the use of the transparent arm in manipulation environments, the use of synthetic (yet fully controllable) datasets, etc.). Nonetheless, we believe OGBench can spur the development of performant offline GCRL *algorithms*, which can then help researchers develop scalable data-driven unsupervised RL pre-training methods for real-world tasks.
>
> * **How are the evaluation goals selected? / Evaluation limited to five goals**
>
> Thanks for the question. For each task, we curated a small number (5) of *representative* evaluation goals that cover diverse goal-reaching behaviors of varying difficulty. For example, in $\texttt{cube-double}$, we curated five evaluation goals to cover single pick-and-place, double pick-and-place, swapping, stacking, etc., and in $\texttt{puzzle-4x6}$, we chose five goals with increasing levels of difficulty (from $6$ to $24$, in terms of the minimum number of button presses). On each of the five goals, we perform $50$ rollouts with slightly randomized poses/color orders/initial states to obtain the results.
>
> As the reviewer pointed out, it is indeed possible to evaluate a goal-conditioned policy with any arbitrary goals. However, we chose to restrict to five evaluation goals in this benchmark (similarly to several prior works in offline goal/language-conditioned RL and robotics [2, 3, 4]). There are two main reasons for this choice: (1) using five pre-defined goals reduces computational cost for evaluation (which sometimes takes even longer than training in long-horizon environments!), as evaluation with fully randomized goals typically requires many more evaluation rollouts to derive statistically significant results, and (2) providing diverse *types* of pre-defined goals enables the practitioner to better analyze their algorithm’s strengths and weaknesses. For example, the results in Table 14 show that some methods are good at single pick-and-place but struggle with tasks involving double pick-and-place, which can guide practitioners on areas for improvement in their methods.

---

> ### Author Response · Authors · 2024-11-16
> **Author Response (2/2)**
>
> * **Why are there no exploratory datasets for AntSoccer?**
>
> We found that $\texttt{explore}$ datasets are already very challenging for AntMaze (Table 2), so we decided not to provide separate $\texttt{explore}$ datasets for AntSoccer, which is much harder than AntMaze. That being said, since we provide the exact commands to reproduce all the datasets, we believe it wouldn’t be difficult for a user to generate custom $\texttt{explore}$ datasets for AntSoccer with the provided data-generation scripts.
>
> * **How exactly is the goal state provided to the agent in locomotion/manipulation tasks?**
>
> In all tasks in our benchmark, the goal space is the same as the state space (L100); in other words, a goal is simply yet another state. As such, a goal state contains both the proprioceptive states (e.g., joint angles) and object positions (if they exist). However, when measuring success during evaluation, we consider only the $x$-$y$ position (in locomotion environments) or object positions (in manipulation environments) (though the goal state still contains the full information), following prior practices in offline goal-conditioned RL [3, 5, 6]. We have clarified this point by making a separate paragraph at the beginning of Section 7, and have described the detailed task success criteria in Appendix E.1.
>
> * **Regarding dataset optimality vs. coverage for offline RL/GCRL**
>
> Thanks for asking this question! We were also a bit surprised by the results in Figure 3, which show that insufficient state coverage can lead to a *complete* failure of otherwise solvable tasks. While the current trend in robotics seems to favor collecting large, expert datasets (as the reviewer mentioned), there has also recently been an increasing number of studies investigating the role of dataset coverage for offline RL [7, 8, 9], which have made similar/relevant observations to ours. For instance, Park et al. [7] show that offline RL is mainly bottlenecked by test-time generalization rather than data quality, highlighting the importance of sufficient noise/coverage. In robotics, data coverage aspects are often considered in the form of having “recovery” transitions from suboptimal states in the dataset [8, 9], which can teach the agent how to recover from mistakes during evaluation. For example, the recent $\pi_0$ generalist robot policy from Physical Intelligence (a robotics startup) uses diverse yet suboptimal data that contains recovery behaviors for post-training [9], and its technical report states that “training on only this high-quality data results in a brittle model” [9], which directly aligns with our observation in Figure 3. We hope that our controllable and reproducible data-generation scripts enable systematic and scientific studies of diverse aspects (e.g., coverage, suboptimality, scalability, diversity, etc.) of datasets for offline RL/GCRL.
>
> We would like to thank the reviewer again for raising important questions about OGBench, and we believe the added limitation section as well as the clarifications have improved the paper substantially. Please let us know if there are any additional concerns or questions.
>
> [1] Frans et al., Powderworld: A Platform for Understanding Generalization via Rich Task Distributions, ICLR 2023. \
> [2] Fang et al., Planning to Practice: Efficient Online Fine-Tuning by Composing Goals in Latent Space, IROS 2022. \
> [3] Zheng et al., Stabilizing Contrastive RL: Techniques for Robotic Goal Reaching from Offline Data, ICLR 2024. \
> [4] Ghugare et al., Closing the Gap between TD Learning and Supervised Learning -- A Generalisation Point of View, ICLR 2024. \
> [5] Park et al., HIQL: Offline Goal-Conditioned RL with Latent States as Actions, NeurIPS 2023. \
> [6] Myers et al., Learning Temporal Distances: Contrastive Successor Features Can Provide a Metric Structure for Decision-Making, ICML 2024. \
> [7] Park et al., Is Value Learning Really the Main Bottleneck in Offline RL?, NeurIPS 2024. \
> [8] Ke et al., CCIL: Continuity-based Data Augmentation for Corrective Imitation Learning, ICLR 2024. \
> [9] Physical Intelligence, π0: A Vision-Language-Action Flow Model for General Robot Control, 2024, https://www.physicalintelligence.company/download/pi0.pdf

---

> > ### Comment · Reviewer_jjTu · 2024-11-18
> >
> > Thank you for the insightfull rebuttal. In line with my previous comments, I think the authors did a great job with this work and, as such, I maintain my score.

---

### Meta-Review · Area_Chair_H2h5 · 2024-12-20

**Metareview:**

This paper introduces a dataset for goal-conditioned RL. Reviewers generally agree that it is an important problem and that the dataset could be useful. Reviewers also found the paper to be well-written. I agree with this sentiment and think that the dataset can complement popular RL datasets like D4RL. One concern raised was that only simple model classes (e.g., MLP and CNN) were raised. I think this is okay for a benchmark paper although it would have been great to see more common transformer-style architectures. I would urge authors to consider adding this for revision.

Overall, I recommend acceptance.

**Additional Comments On Reviewer Discussion:**

Reviewers raised the following main concerns:

1. Question about novelty, relation to d4rl (reviewer BGac):

I think the dataset is a nice complement to D4RL although the AntMaze tasks is an extension of the one in D4RL. The environment also has image features although there are extensions of D4RL like Visual D4RL (https://github.com/conglu1997/v-d4rl).

2. The dataset has unrealistic setting like a transparent robotic arm (reviewer sCze)

The authors provided a feature to control this behavior

3. Missing details on goals (reviewer sCze), more decimal places (reviewer BGac), and limitations (reviewer jjTu)

The authors provided these extra details.

4. Use of simple models like MLP and CNN

The authors were unable to run transformer-based models. It will be great to have these results as these are the most common model architectures.

---

### Decision · Program_Chairs · 2025-01-22

Accept (Poster)